# Sensitivity of cirrus and contrail radiative effect on cloud microphysical and environmental parameters.

Kevin Wolf[1], Nicolas Bellouin[1,2], and Olivier Boucher[1]

[1]Institut Pierre-Simon Laplace, Sorbonne Université / CNRS, Paris, France
[2]Department of Meteorology, University of Reading, Reading, United Kingdom

**Correspondence:** Kevin Wolf (kevin.wolf@ipsl.fr)

**Abstract.** Natural cirrus clouds and contrails cover about 30% of the Earth's mid-latitudes and up to 70% of the Tropics. Due to their widespread occurrence, cirrus have a considerable impact on the Earth energy budget, which, on average, leads to a warming net radiative effect (solar + thermal-infrared). However, whether the instantaneous radiative effect (RE), which in some cases corresponds to a radiative forcing, of natural cirrus or contrails is positive or negative depends on their microphysical, macrophysical, and optical properties, as well as the radiative properties of the environment. This is further complicated by the fact that the actual ice crystal shape is often unknown and, thus, ice clouds remain one of the components that are least understood in the Earth's radiative budget.

The present study aims to investigate the dependency of the effect on cirrus RE on eight parameters: solar zenith angle, ice water content, ice crystal effective radius, cirrus temperature, surface albedo, surface temperature, cloud optical thickness of an underlying liquid water cloud, and three ice crystal shapes. In total, 283,500 plane-parallel radiative transfer simulations have been performed, not including three-dimensional scattering effects. Parameter ranges are selected that are typically associated with natural cirrus and contrails. In addition, the effect of variations in the relative humidity profile and the ice cloud geometric thickness have been investigated for a sub-set of the simulations. The multi-dimensionality and complexity of the 8-dimensional parameter space makes it impractical to discuss all potential configurations in detail. Therefore, specific cases are selected and discussed.

For a given parameter combination, the largest impact on solar, thermal-infrared (TIR), and net RE is related to the ice crystal effective radius. The second most important parameter is ice water content, which equally impacts the solar and terrestrial RE. Solar RE of cirrus is also determined by solar zenith angle, surface albedo, liquid cloud optical thickness, and ice crystal shape in descending priority. RE in the TIR spectrum is dominated by surface temperature, ice cloud temperature, liquid water cloud optical thickness, and ice crystal shape. Net RE is controlled by surface albedo, solar zenith angle, and surface temperature in decreasing importance. The relative importance of the studied parameters differs depending on the ambient conditions. Furthermore, and during nighttime the net RE is equal to the TIR RE.

The data set generated in this work is publicly available. It can be used as a look-up-table to extract the RE of cirrus clouds, contrails, and contrail cirrus instead of full radiative transfer calculations.

# 1  Introduction

Cirrus clouds cover large areas of the Earth, with cloud cover estimates of 30 % in the mid-latitudes and up to 70 % in the tropics (Liou, 1986; Wylie and Menzel, 1999; Chen et al., 2000; Sassen et al., 2008; Nazaryan et al., 2008). Due to their widespread occurrence, cirrus can have a considerable impact on the global energy budget. In addition to cirrus, air traffic leads to the formation of condensation trails, also termed contrails, which are geometrically and optically thin clouds with similar radiative effects as thin natural cirrus (Liou, 1986). For the sake of simplicity, the term cirrus is used interchangeably for natural cirrus, contrail-induced cirrus, and contrails throughout this article.

Depending on ambient conditions, contrails are short lived (t < 10 min) but can persist up to a day, when the surrounding air mass is sufficiently cold and moist (Schumann, 1996; Haywood et al., 2009; Schumann and Heymsfield, 2017; Kärcher, 2018). In such conditions, persistent contrails transition from line-shaped clouds to larger cloud fields (Unterstrasser and Stephan, 2020). Modeling and satellite studies have estimated that contrail and contrail-induced cirrus cloud cover can reach up to 6 to 10 % over Europe (Burkhardt and Kärcher, 2011; Quaas et al., 2021) and significantly contribute to high-level cloudiness over Europe (Schumann et al., 2015, 2021).

Under most circumstances cirrus have a cooling effect in the solar wavelength range (0.2–3.5 $\mu$m, sometimes called short-wave) and a heating effect in the thermal-infrared (TIR) wavelength range (3.5–75 $\mu$m, sometimes also termed longwave or terrestrial). The net radiative effect (solar cooling + TIR warming) is often a warming as the TIR effect dominates (Chen et al., 2000). By combining satellite observations and radiative transfer (RT) simulations, Chen et al. (2000) estimated a global annual mean cirrus cloud radiative effect (RE) of $-25.3$ W m$^{-2}$ in the solar wavelength range and 30.7 W m$^{-2}$ in the TIR wavelength range, leading to a positive net effect of 5.4 W m$^{-2}$. However, whether the instantaneous RE of natural cirrus or contrails is positive or negative depends on their microphysical, macrophysical, and optical, as well as radiative properties of the environment. The cloud properties relevant to the RE of the cloud are primarily cloud altitude, cloud temperature, ice water content, ice crystal shape (also called crystal habit), and the orientation of the ice crystals (Fu and Liou, 1993; Stephens et al., 2004; Campbell et al., 2016). Furthermore, the underlying surface properties, i.e., surface albedo and surface temperature, as well as gaseous absorption and additional underlying cloud layers, also have an effect on the cirrus RE. Dynamical processes in the atmosphere have a strong influence on those parameters, for example lifting of air masses along warm conveyor belts or cloud anvils, that lead to a variety of ice crystal shapes and crystal surface roughness (Freudenthaler et al., 1996; Wendisch et al., 2007; Yang et al., 2010; Krämer et al., 2016; Luebke et al., 2016). As a result, the actual distribution of crystal shapes within a cirrus and the related RE are often unclear. Thus ice clouds remain one of the components that is least understood in the Earth's radiative budget (Stevens and Bony, 2013; Bauer et al., 2015; Bickel et al., 2020) and this lack of understanding contributes to uncertainties in the climate impact of aviation (Lee et al., 2021).

To estimate the radiative impact of a cloud as well as related potential uncertainties and sensitivities, RT simulations represent a helpful tool. While the atmospheric RT in liquid water clouds composed of spherical cloud droplets can rely on geometric optics or Mie-scattering theory (Mie, 1908; van de Hulst, 1981), RT in ice clouds is complicated by the non-spherical crystal shape and the interaction with incoming radiation, i.e., through their single-scattering phase function. The single-scattering

phase function, for example, has to be determined by computationally-expensive methods, like ray tracing (Bi et al., 2014),

Monte Carlo simulations (Macke et al., 1996a, b), or the T-matrix method (Mishchenko, 2020). Due to the computational burden of such accurate simulations, parameterizations of ice crystal properties are often developed and validated against the more precise calculations (Takano and Liou, 1989; Fu, 1996; Yang et al., 2000, 2013). More recent ice crystal parameterizations by Yang et al. (2000), Baum et al. (2005a, b), Baum et al. (2007), and Yang et al. (2013) in combination with the latest RT models allow to determine the radiative impact of cirrus clouds with acceptable computational cost and accuracy. By varying

the microphysical and macrophysical properties of the cirrus, as well as the surface properties in the RT model, the natural range of cirrus and their environment can be represented and the RE can be estimated. Furthermore, uncertainties due to the insufficiently known crystal shape can be assessed.

Multiple studies that aimed to investigate the impact of a certain parameter on cloud RE have been performed in the past. Fu and Liou (1993) as well as Yang et al. (2010) focused on the effects of the selected ice crystal habit and ice water path. The effect

of the ice crystal size distribution was analyzed, for example, by Zhang et al. (1999) or Mitchell et al. (2011). A comprehensive study of cirrus radiative effects was conducted by Schumann (2012), who aimed to derive a parameterization to estimate the cloud RE. While those studies are valuable, none of them investigate the effect of multiple factors, like relevant cloud and environmental input parameters. These studies have identified parameters that affect cirrus RE, but all these parameters need to be considered together, including both cloud and environmental parameters. This article is intended as a parametric sensitivity

study that aims to compare the effects of major parameters. Furthermore, we identify the driving parameters of RE by sampling the input parameter range, restricted to values that are typically associated with ice clouds. Finally, we provide an open-access data set, which allows the user to extract cloud REs for user-specific combinations of the input parameters. The look-up-table could in fact be coupled with models of any complexity, as long as they simulate the dimensions of the data set, namely: solar zenith angle, ice cloud temperature, surface albedo, ice water content, surface temperature, ice crystal effective radius, and

liquid water cloud optical thickness.

The study is structured in the following way. Section 2 introduces the selected parameter space, the RT model, and outlines basic definitions as well as methods used in the paper. Subsequently, Section 3 presents the results from the RT simulations. Because our simulations assume plane-parallel atmosphere and homogeneous clouds, Section 4 discusses 3-dimensional RT. That is followed by the summary in Sec. 5.

## 2  Methods and Definitions

### 2.1  Definition of radiative effect and albedo

The radiative impact of a perturbation, e.g., clouds, is quantified by the concept of the radiative effect (RE). The RE is defined as the net difference in downward and upward irradiance ($F^{\downarrow} - F^{\uparrow}$) between the perturbed and unperturbed condition. In the case of clouds, the cloud radiative effect (CRE, denoted here as $\Delta F$) is the difference in fluxes between the cloud ($F_c$) and

cloud-free ($F_{cf}$) atmosphere at a given altitude $z$ (Ramanathan et al., 1989; Stapf et al., 2021; Luebke et al., 2022):

$$\Delta F(z) = F_c(z) - F_{cf}(z) = \left[F^{\downarrow}(z) - F^{\uparrow}(z)\right]_c$$
$$- \left[F^{\downarrow}(z) - F^{\uparrow}(z)\right]_{cf}, \tag{1}$$

where the upward and downward, cloudy and cloud-free irradiances are all counted positive. The net RE is given by:

$$\Delta F_{net}(z) = \Delta F_{sol}(z) + \Delta F_{TIR}(z), \tag{2}$$

which can be split into a solar and a thermal-infrared component. Within this study, the CRE is calculated for the top of atmosphere (TOA), which is set in the radiative transfer calculations to an altitude of 120 km, unless stated otherwise.

In addition to the RE, the albedo $\alpha$ describes the interaction of a cloudy scene or a surface with the solar, incident radiation. The scene albedo $\alpha_{sol}(z = TOA)$ at the TOA is defined as the ratio of the reflected, upward irradiance $F_{sol}^{\uparrow}$ at TOA in relation to the incident, downward irradiance $F_{sol}^{\downarrow}$ at TOA and is given by:

$$\alpha_{sol,TOA} = \frac{F_{sol}^{\uparrow}(z = TOA)}{F_{sol}^{\downarrow}(z = TOA)}. \tag{3}$$

Similarly, the surface albedo $\alpha_{sol,srf}$ is calculated with $F_{sol,srf}^{\uparrow}$ and $F_{sol,srf}^{\downarrow}$ the respective irradiances at the surface ($z = 0$ km).

## 2.2 Radiative transfer simulation set-up

Upward and downward irradiances $F^{\uparrow}$ / $F^{\downarrow}$ were simulated with the library for Radiative transfer (libRadtran, Emde et al., 2016). The solar irradiances $F_{sol}$ cover a wavelength range from 0.3 to 3.5 $\mu$m, which represents 97.7 % of the total incoming solar radiation (0–10 $\mu$m) calculated from the spectrum provided by Kurucz (1992). The thermal infrared (TIR) irradiances include wavelengths from 3.5 to 75 $\mu$m, representing 99.3 % of the integrated blackbody radiation (3.5 to 100 $\mu$m) at 285 K (12°C).

The RT simulations are performed with the one–dimensional (1D) solver DISORT (Stamnes et al., 1988; Buras et al., 2011), which is part of libRadtran. Clouds are assumed to be horizontally uniform and lateral photon transport between columns is neglected, which is called the independent pixel approximation (IPA, Stephens et al., 1991; Cahalan et al., 1994). As the main objective of this study is to map the basic dependencies of $\Delta F$ on the driving parameters, we neglect any variability in the spatial ice water content (IWC) distribution that exists in cirrus (Minnis et al., 1999). We also restrict the simulations to fully cloud covered scenes. The required number of streams was iteratively determined and set to 16 streams, which provides sufficient accuracy while limiting computational time. The trade-off between accuracy and computational time is detailed in Appendix C. The spectral TOA solar irradiance is provided by Kurucz (1992). The RT simulations consider molecular absorption using the 'coarse' resolution REPTRAN parameterization from Gasteiger et al. (2014). Section C in the appendix provides an uncertainty estimation related to the REPTRAN resolution. Absorption by water vapor, carbon dioxide, ozone, nitrous oxide, carbon monoxide, methane, oxygen, and nitrogen and nitrogen dioxide is included in the simulations (Anderson et al., 1986; Emde et al., 2016).

The sensitivity of solar, TIR, and net cloud RE $\Delta F$ is estimated by varying eight parameters. The parameter ranges were chosen to represent commonly observed cirrus and contrail cirrus properties, as well as environmental parameters.

- The daily course of the Sun position is represented by solar zenith angles $\theta$ ranging from $0°$ and $85°$. Larger $\theta$ values are omitted to avoid numerical instability that would require more streams in the calculation. Furthermore, RT simulations with the DISORT solver for $\theta > 85°$ have to be interpreted with caution as DISORT does not consider the sphericity of the Earth and treats atmospheric layers as plane-parallel (Stamnes et al., 1988; Buras et al., 2011). In addition, differences between 1D and three–dimensional (3D) RT simulations increase significantly with values of up to $40\,\%$ (Gounou and Hogan, 2007; Forster et al., 2012).

- The Earth's surface albedo, $\alpha_{\mathrm{srf}}$ ranges from 0 to 1, which represents the full possible range. In general, $\alpha_{\mathrm{srf}}$ varies spectrally but here is kept constant for all solar wavelength. It is varied between 0 and 1 to include surface conditions ranging from open ocean to full sea ice or snow (Baldridge et al., 2009; Gardner and Sharp, 2010; Meerdink et al., 2019; Gueymard et al., 2019). Values of $\alpha_{\mathrm{srf}}$ are given in Table 4. In the TIR wavelength range $\alpha_{\mathrm{srf}}$ is assumed to be 0, which leads to an emissivity $\epsilon = 1$ with the Earth's surface thus acting as a blackbody (Wilber, 1999).

- Three atmospheric profiles (AP) are selected to represent subarctic, mid-latitude, and tropical conditions. The simulations are based on the subarctic winter (`afglsw`), the US standard (`afglus`), and the tropical (`afglt`) profiles after Anderson et al. (1986). Surface temperatures $T_{\mathrm{srf}}$ of $-15.95°C$ (subarctic winter), $14.85°C$ (US standard), and $26.55°C$ (tropical) are defined in libRadtran by the lower most temperature in the APs. The profile of relative humidity is linked to the AP via the Clausius–Clapeyron-equation (Corti and Peter, 2009). Variations in the water vapor (WV) profile primarily impact the RT in the TIR wavelength range, particularly in WV absorption bands, while RT in the solar wavelength range is less affected (Liou, 1992). The cirrus cloud top temperatures $T_{\mathrm{cld,ice}}$ are selected to span the temperature range in which contrails and cirrus typically form (Krämer et al., 2020). Here we cover a range from 219 to 243 K. The resulting ice cloud top altitudes $z_{\mathrm{ice,CT}}$ are set to the altitude, where the temperature in the APs equals the desired $T_{\mathrm{cld,ice}}$. $z_{\mathrm{ice,CT}}$ is found by linear interpolation between the altitude and temperature levels. Cirrus temperatures and related $z_{\mathrm{ice,CT}}$ are listed in Table 1. Within the simulations, the ice cloud geometric thickness $\mathrm{d}z$ is set to 1000 m for all simulations, which represents an average for observed contrails as well as natural cirrus (Freudenthaler et al., 1995; Sassen and Campbell, 2001; Noël and Haeffelin, 2007; Iwabuchi et al., 2012).

- Three different ice crystal shapes, namely: i) moderately rough aggregates of 8-element columns (called 'aggregates' thereafter), agglomerations of 8–columnar ice crystals; ii) 'droxtals', almost spherical ice crystals; and iii) 'plates' are used. These three shapes are selected to represent different stages in the temporal evolution of contrails. Several airborne in situ measurement campaigns that targeted cirrus and contrails imply that aggregates are the dominating ice crystal habit (Liu et al., 2014; Holz et al., 2016; Järvinen et al., 2018). For example, Järvinen et al. (2018) found that 61 to 81 % of the sampled ice crystals had complex shapes. They further noted that severely roughened column aggregates resemble their observations best. Such ice crystals are also assumed in current remote sensing applications of ice cloud, e.g., in

**Table 1.** Surface temperature, cloud top temperature, cloud top altitude, and cloud top pressure level of the liquid water, and ice water cloud depending on the atmosphere profile.

| | Profiles | | |
|---|---|---|---|
| | US Standard (`afglus`) | Tropical (`afglt`) | Subarctic winter (`afglsw`) |
| | Surface temperature | | |
| | 288.2 K (14.85°C) | 299.7 K (26.55°C) | 257.2 K (-15.95°C) |
| Cirrus temperature | Cirrus altitude (km) / pressure (hPa) | | |
| 219 K ($-54°$C) | 10.7 / 240 | 12.7 / 191 | 8.5 / 308 |
| 225 K ($-48°$C) | 9.7 / 276 | 11.8 / 220 | 7.3 / 367 |
| 231 K ($-42°$C) | 8.8 / 318 | 10.9 / 252 | 6.5 / 419 |
| 237 K ($-36°$C) | 7.9 / 363 | 10.0 / 286 | 5.6 / 476 |
| 243 K ($-30°$C) | 7.0 / 414 | 9.1 / 325 | 4.7 / 540 |
| | Cloud top temperature for liquid cloud at 1.5 km (K / °C) | | |
| | 278.5 K / 5.35°C | 290.7 K / 17.55°C | 257.5 K / $-15.65$°C |

the re-defined ice optical properties used by the Moderate Resolution Imaging Spectroradiometer (MODIS) Collection 6 product (Yang et al., 2013; Holz et al., 2016; Platnick et al., 2017; Forster and Mayer, 2022). Furthermore, Forster and Mayer (2022) found mixtures of severely roughened ( 60 %) and smooth ( 40 %) 8-column aggregates to best match observations of (thin) cirrus. As a compromise, we selected moderately rough 8–column–aggregates as the primary ice crystal habit. The second most observed habit are plate-like ice crystals (Holz et al., 2016; Forster et al., 2017; Järvinen et al., 2018), which are included in the simulations as a second shape. The 'droxtal' parameterization is selected to estimate $\Delta F$ of young contrails, which primarily consist of near-spherical ice crystals (Goodman et al., 1998; Lawson et al., 1998; Gayet et al., 2012). We emphasize that contrails can be comprised of other ice crystal shapes, like single columns, hollow columns, 3D bullet rosettes, or mixtures of these (Lawson et al., 1998; Baum et al., 2005a), but the simulated shapes cover the majority of observed cirrus situations. The utilized ice optical properties of the three selected shapes are based on the parameterization from Yang et al. (2013) that assume randomly oriented ice crystals with a 'moderate' surface roughness.

– Within libRadtran clouds are defined by their geometric thickness d$z$, effective radius $r_{\text{eff}}$, and IWC. Typical IWC of contrails and in situ cirrus can range from $10^{-5}$ to 0.2 g m$^{-3}$ as found during the Mid–Latitude Cirrus campaign (Luebke et al., 2016; Krämer et al., 2016, 2020). For our simulations, we span a similar range of IWC from $7 \cdot 10^{-7}$ to 0.1 g m$^{-3}$.

– Aircraft in situ observations of young ($t < 120$ s) contrails showed that these consist of ice crystals with diameters up to a few micrometers (Petzold et al., 1997; Sassen, 1997; Lynch et al., 2002). Shortly thereafter these ice crystals grow in size and reach ice crystal radius $r_{\text{eff}}$ between 2 and 5 $\mu$m (Jeßberger et al., 2013; Bräuer et al., 2021). The majority of

ice crystals in older ($t > 120$ s) contrails and cirrus have $r_{\text{eff}}$ between 10 and 150 $\mu$m (Krämer et al., 2020), while mature cirrus can be composed of ice crystals with diameters larger than 150 $\mu$m (Schröder et al., 2000). The selected ice optical properties allow for simulations between 5 to 85 $\mu$m and thus cover the lower and mid range of the natural crystal size spectrum.

Within libRadtran the bulk-scattering properties of ice clouds are obtained by integrating the single-scattering properties over the entire ice crystal / particle size distribution (PSD). The PSD of an ice cloud can be approximated by a gamma distribution (Hansen and Travis, 1974; Evans, 1998; Heymsfield et al., 2002; Baum et al., 2005a, b), which is given by:

$$n(r_{\text{e}}) = N \cdot r_{\text{e}}^{\mu} \cdot \exp\left(-\Lambda \cdot r_{\text{e}}\right), \tag{4}$$

with $n(r_{\text{e}})\mathrm{d}r$ the number of ice crystals with radii in the range of $r_{\text{e}}$ and $r_{\text{e}} + \mathrm{d}r$. $N$ is a normalization constant such that the integral over the PSD yields the number of crystals in a unit volume (Emde et al., 2016). $N$ itself results from the choice of the parameters in Eq. 4 that are given by the slope $\Lambda = \frac{1}{a \cdot b}$ and dispersion $\mu = \frac{1-3b}{b}$. Inserting $a$ and $b$ into Eq. 4 leads to:

$$n(r_{\text{e}}) = N \cdot r_{\text{e}}^{\left(\frac{1}{b}-3\right)} \cdot \exp\left(-\frac{r}{ab}\right), \tag{5}$$

Parameter $b$ corresponds to the effective variance $\nu_{\text{eff}}$ (unitless), with typical values between 0.1 and 0.5 (Evans, 1998; Heymsfield et al., 2002). In libRadtran $\nu_{\text{eff}}$ is set to 0.25 (Emde et al., 2016). Parameter $a$ corresponds to the targeted effective radius $r_{\text{eff}}$ of the PSD. Multiple definitions for $r_{\text{eff}}$ exist in the case of non-spherical crystals. Here we follow the definition from Yang et al. (2000), Key et al. (2002), Baum et al. (2005b), Baum et al. (2007), and Schumann et al. (2011), which describe the diameter $D_{\text{e}}$ and radius $r_{\text{e}}$ of a non-spherical ice crystal as:

$$D_{\text{e}} = 2 \cdot r_{\text{e}} = \frac{D_{\text{V}}^3}{D_{\text{A}}^2}, \tag{6}$$

with $D_{\text{V}}$ the diameter of a spherical crystal with the same average volume as the ice crystal and $D_{\text{A}}$ the diameter of a spherical crystal with the same projected area as the ice crystal. $D_{\text{A}}$ is defined by:

$$D_{\text{A}} = 2 \cdot r_{\text{A}} = 2 \cdot \left(\frac{A}{\pi}\right)^{1/2} \tag{7}$$

and $D_{\text{V}}$ is given by:

$$D_{\text{V}} = 2 \cdot r_{\text{V}} = \left(\frac{6 \cdot V}{\pi}\right)^{1/3}, \tag{8}$$

where $V$ and $A$ are the volume and the mean projected area of the ice crystal, respectively. As demonstrated by Mitchell (2002) the definition of $D_{\text{e}}$ and $r_{\text{e}}$ of a single crystal can be applied to a PSD, when evaluated at a bulk ice density of $917 \, \text{kg} \, \text{m}^{-3}$, which finally leads to:

$$r_{\text{eff}} = \frac{3 \cdot \int_{L1}^{L2} V(L) n(L) \mathrm{d}L}{4 \cdot \int_{L1}^{L2} A(L) n(L) \mathrm{d}L}, \tag{9}$$

with $L1$ and $L2$ the minimum and maximum crystal size of the distribution.

The original ice optical properties from Yang et al. (2013) are processed by weighting the size dependent single-scattering phase function with the gamma distribution (Emde et al., 2016). For the gamma size distribution a minimum and maximum $r_{\text{eff}}$ of 5 and 90 $\mu$m are selected. Parameter $a$ in Eq. 5 is found iteratively such that the desired $r_{\text{eff}}$ of the distribution is achieved. The obtained bulk optical properties are used for RT in the solar and the TIR wavelength range. Examples of phase functions $\mathcal{P}$ for four different crystal shapes and their characteristic features are visualized in Appendix D.

– Cloud geometric thickness $\text{d}z$ is set to 1000 m. That represents a contrail after approximately 30 min lifetime (Freudenthaler et al., 1995) and an average cirrus or aged contrail as confirmed by climatologies from lidar (Noël and Haeffelin, 2007; Iwabuchi et al., 2012) and satellite observations, for example, by Sassen and Campbell (2001). During the cloud life time the ice crystals might grow due to supersaturation and WV deposition, and start to sediment. Sedimentation lowers the cloud base altitude and increases $\text{d}z$. Meerkötter et al. (1999) reported that variations in $\text{d}z$ have only a minor impact on the cloud RE. However, to estimate the effect of varying $\text{d}z$ a dedicated sensitivity study on $\text{d}z$ was performed for a sub-set of the parameter range and $\text{d}z$ of 500, 1000, and 1500 m. To investigate the effect of variations in $\text{d}z$ on solar, TIR, and net RE, a separate sensitivity study for a sub-set of the full parameter space is performed with $\text{d}z$ of 500 and 1500 m, while keeping the total ice water path (IWP) constant and, thus, the solar cloud optical thickness $\tau_{\text{ice}}$ constant. The total IWP and the scaled IWC are provided in Table 2. $\tau_{\text{ice}}$ can be approximated by:

$$\tau_{\text{ice}} = \frac{3 \cdot Q_{\text{e}} \cdot IWC \cdot \text{d}z}{4 \cdot \rho_{\text{ice}} \cdot r_{\text{eff}}} = \frac{3 \cdot Q_{\text{e}} \cdot IWP}{4 \cdot \rho_{\text{ice}} \cdot r_{\text{eff}}} = \frac{3 \cdot IWP}{2 \cdot \rho_{\text{ice}} \cdot r_{\text{eff}}} \tag{10}$$

with density of ice $\rho_{\text{ice}} = 917 \, \text{kg} \, \text{m}^{-3}$ and $Q_{\text{e}} \approx 2$ the average solar extinction efficiency factor of ice crystals (Horváth and Davies, 2007; Wang et al., 2019). It has to be noted that Eq. 10 is only applicable for the solar wavelength range.

– The parameter sensitivity study is complemented by investigating the influence of a second cloud layer. The second cloud layer is implemented as a stratiform, low-level liquid water cloud with a constant cloud top altitude $z_{\text{liq,CT}}$ at 1500 m and a geometric thickness of 500 m. The altitude of 1500 m was selected as a compromise between typical conditions of low-level stratiform clouds in the Subarctic, the mid-latitudes, and tropical regions. McFarquhar et al. (2007) and van Diedenhoven et al. (2009) found $z_{\text{liq,CT}} = 1000 \, \text{m}$ for Arctic clouds. Slightly higher $z_{\text{liq,CT}}$ between 1000 and 1500 m are found in the mid-latitudes (Rémillard et al., 2012; Muhlbauer et al., 2014). Low-level clouds in the tropics also range between 500 and 1700 m even though some cloud tops can reach up to 2000 m (Medeiros et al., 2010; Stevens et al., 2016). Fixing $z_{\text{liq,CT}}$ at 1500 m leads to liquid cloud top temperature $T_{\text{liq}}$ of 278.5 and 290.7 K for the mid-latitude and tropical profile, respectively. In the Subarctic profile however, $T_{\text{liq}}$ reaches 257.2 K ($-15.95$ K), which is below freezing and implies a super-cooled liquid water cloud. This agrees with observations from Hogan et al. (2004) and Hu et al. (2010), who found that the majority of clouds in the Arctic ($\approx 70\,\%$) are characterized by super-cooled droplets at cloud top. Furthermore, 95 % of the observed clouds that have a $T_{\text{liq}}$ between $-15$ and 0°C have super-cooled droplets at the top. The cloud optical thickness $\tau_{\text{liq}}$ at 550 nm wavelength of the liquid water cloud is varied between 0 and 20. Within

**Table 2.** Ice water path IWP (in $\mathrm{g\,m^{-2}}$) and ice water content IWC (in $\mathrm{g\,m^{-3}}$) for the reference with $\mathrm{d}z = 1000$ m and the two additional clouds with $\mathrm{d}z$ of 500 and 1500 m.

| | IWP [$\mathrm{g\,m^{-2}}$] | | | | | | |
|---|---|---|---|---|---|---|---|
| | 0.7 | 1.5 | 3 | 6 | 12 | 24 | 100 |
| IWC (dz = 500 m) [$\mathrm{g\,m^{-3}}$] | 0.0014 | 0.003 | 0.006 | 0.012 | 0.024 | 0.048 | 0.2 |
| IWC (dz = 1000 m) [$\mathrm{g\,m^{-3}}$] | 0.0007 | 0.0015 | 0.003 | 0.006 | 0.012 | 0.024 | 0.1 |
| IWC (dz = 1500 m) [$\mathrm{g\,m^{-3}}$] | 0.00045 | 0.001 | 0.002 | 0.004 | 0.008 | 0.016 | 0.0667 |

**Table 3.** Basic model configuration and selected settings.

| Model configuration | Selected value / setting |
|---|---|
| Radiative transfer solver | DISORT (Stamnes et al., 1988; Buras et al., 2011) |
| Number of streams | 16 |
| Extraterrestrial solar spectrum | Kurucz (1992) |
| Wavelength range | 0.3–3.5 $\mu$m (solar) & 3.5–75 $\mu$m (thermal-infrared) |
| Molecular absorption | REPTRAN (Gasteiger et al., 2014) |
| Ice properties | Yang et al. (2013) |
| Output altitude | 120 km = TOA |

the RT simulations the optical properties of liquid water clouds are represented by pre-calculated Mie tables (Mie, 1908; van de Hulst, 1981).

An overview of the model configuration is given in Table 3 and the input parameter space is listed in Table 4. An example libRadtran input file is provided as supplementary material.

For each of the three simulated ice crystal shapes a NetCDF file is provided (Wolf et al., 2023). The files include ice cloud optical thickness $\tau_{\mathrm{ice}}$, the simulated upward and downward irradiances $F$ at TOA with 120 km (with and without the presence of the ice cloud), and the calculated ice cloud radiative effect $\Delta F$ (solar, TIR, net). The available cloudy and cloud-free irradiances further allow to calculate the cirrus RE by scaling the 'cloudy' RE with the required cloud cover. An overview of all variables provided in the NetCDF files are given in Table 5. The data set allows the user to extract $\Delta F$ values for their parameter combinations, instead of running costly RT simulations. The look-up-table could in fact be coupled with models of any complexity, as long as they simulate the dimensions of the data set, namely: solar zenith angle, ice cloud temperature, surface albedo, ice water content, surface temperature, ice crystal effective radius, and liquid water cloud optical thickness.

The simulations base on three relative humidity profiles, which were selected to represent subarctic, mid-latitude, and tropical conditions. An estimation in RE variability due to variations in the RH profile showed an effect of less than 1 % for $\Delta F_{\mathrm{sol}}$ but can range up to 4 % for $\Delta F_{\mathrm{tir}}$ and 8 % for $\Delta F_{\mathrm{net}}$ especially for the warm and moist tropical profile. These variations have to

**Table 4.** Simulated parameter space.

| Model parameter | Symbol | Simulated values | Total number of combinations |
|---|---|---|---|
| Solar zenith angle (°) | $\theta$ | 0, 10, 30, 50, 70, 85 | 6 |
| Ice water content (g m$^{-3}$) | IWC | 0.0007, 0.0015, 0.003, 0.006, 0.012, 0.024, 0.1 | 7 |
| Crystal effective radius ($\mu$m) | $r_{\mathrm{eff}}$ | 5, 10, 15, 25, 60, 85 | 6 |
| Cirrus temperature (K) | $T_{\mathrm{cld,ice}}$ | 219, 225, 231, 237, 243 | 5 |
| Solar surface albedo | $\alpha_{\mathrm{srf}}$ | 0, 0.15, 0.3, 0.6, 1.0 | 5 |
| Surface temperature (K) | $T_{\mathrm{srf}}$ | 257.2, 288.2, 299.7 | 3 |
| Atmosphere profiles | - | US Standard atmosphere `afglus`, tropical `afglt`, subarctic winter `afglsw` | - |
| Second cloud layer optical depth | $\tau_{\mathrm{liq}}$ | 0, 1, 5, 10, 20 | 5 |
| Ice crystal shapes | - | droxtals, plates, aggregates (moderately rough aggregates of 8-element columns) | 3 |
| | | | 283,500 |

be considered, when using the data set. We further emphasizes that the simulations are performed with a 1D RT solver, i.e., plane-parallel clouds, that neglect 3D scattering and horizontal photon transport (Gounou and Hogan, 2007).

## 2.3 Relationship between effective radius, ice water content, crystal number concentration, and cloud optical thickness

The liquid water content ($LWC$) of a liquid water cloud can be obtained by:

$$\mathrm{LWC} = \frac{4}{3} \cdot \pi \cdot \rho_{\mathrm{liq}} \cdot \int_{0}^{\infty} n(r) \cdot r^3 \cdot \mathrm{d}r, \tag{11}$$

with $\rho_{\mathrm{liq}} = 1000 \ \mathrm{kg\,m^{-3}}$ the density of liquid water, $r$ the radius, and $n(r)$ the number of droplets with size $r$. Equation 11 assumes spherical ice crystals, so might be valid for droxtals, which are almost spherical ice crystals, but it is invalid for other ice crystal shapes. To obtain the particle number concentration $N_{\mathrm{ice}}$ for non-spherical crystals, appropriate power-law mass-dimension relations are needed. Here we employ Eq. 29 from Mitchell et al. (2006) but modify the notation to be consistent with the previous equations from the present study. Equation 29 from Mitchell et al. (2006) is then given by:

$$\mathrm{IWC} = \frac{\alpha \cdot \Gamma(\beta + \mu + 1) \cdot N_{\mathrm{ice}}}{\Gamma(\mu + 1) \cdot \Lambda^{\beta}}, \tag{12}$$

with $\Gamma$ the result of the numerically solved gamma function. The constants $\alpha$ and $\beta$ are the prefactor and the power in the mass–dimensional relationship, respectively. They are related by:

$$m = \alpha \cdot D^{\beta}, \tag{13}$$

**Table 5.** List of variables that are provided in the NetCDF. The output is provided at top of atmosphere located at 120 km altitude.

| Long name | Symbol | Variable name in NetCDF file | Unit |
|---|---|---|---|
| Dimensions | | | |
| Solar zenith angle | $\theta$ | *solar_zenith_angle* | $^\circ$ |
| Ice cloud temperature | $T_{\mathrm{ice}}$ | *ice_cloud_temp* | K |
| Surface albedo | $\alpha_{\mathrm{srf}}$ | *surface_albedo* | - |
| Ice water content | IWC | *ice_water_content* | $\mathrm{g\,m^{-3}}$ |
| Surface temperature | $T_{\mathrm{srf}}$ | *surface_temperature* | K |
| Ice crystal effective radius | $r_{\mathrm{eff}}$ | *crystal_effective_radius* | $\mu$m |
| Liquid water cloud optical thickness | $\tau_{\mathrm{liq}}$ | *optical_thickness_liquid_water_cloud* | - |
| Cloud fraction | - | *cloud_fraction* | - |
| Variables | | | |
| Downward solar total (direct + diffuse) irradiance | $F^{\downarrow}_{\mathrm{sol}}$ | *Fdn_sol* | $\mathrm{W\,m^{-2}}$ |
| Upward solar irradiance | $F^{\uparrow}_{\mathrm{sol}}$ | *Fup_sol* | $\mathrm{W\,m^{-2}}$ |
| Downward thermal-infrared irradiance | $F^{\downarrow}_{\mathrm{tir}}$ | *Fdn_tir* | $\mathrm{W\,m^{-2}}$ |
| Upward thermal-infrared irradiance | $F^{\uparrow}_{\mathrm{tir}}$ | *Fup_tir* | $\mathrm{W\,m^{-2}}$ |
| Solar cloud radiative effect | $\Delta F_{\mathrm{sol}}$ | *RF_sol* | $\mathrm{W\,m^{-2}}$ |
| Thermal-infrared cloud radiative effect | $\Delta F_{\mathrm{tir}}$ | *RF_tir* | $\mathrm{W\,m^{-2}}$ |
| Net radiative effect | $\Delta F_{\mathrm{net}}$ | *RF_net* | $\mathrm{W\,m^{-2}}$ |
| Ice cloud optical thickness | $\tau_{\mathrm{ice}}$ | | - |

with $m$ the mass of the ice crystal and $D$ the maximum dimension of the ice crystal. Both constants depend on the ice crystal shape and are, for example, listed in Mitchell (1996). Using Eq. 13 and assuming an exponential PSD with the special case $\mu = 0$ and $\Lambda = \frac{3}{r_{\mathrm{e}}}$ (Deirmendjian, 1962; Petty and Huang, 2011), finally leads to:

$$N_{\mathrm{ice}} = \frac{3^{\beta} \cdot \mathrm{IWC}}{\alpha \cdot \Gamma(\beta+1) \cdot r_{\mathrm{e}}^{\beta}}. \tag{14}$$

Therefore $N_{\mathrm{ice}}$ is proportional to $\frac{\mathrm{IWC}}{r_{\mathrm{e}}^{\beta}}$, with $\beta$ around 2 for aggregates, 2.4 for hexagonal-plates, and 3 for almost spherical droxtals (Mitchell, 1996).

## 2.4 Approximation of radiative transfer in the thermal-infrared

Radiation in the TIR is primarily of terrestrial origin (Glickman, 2000). Therefore, the TIR irradiance at the TOA has only an upward directed component $F^{\uparrow,\mathrm{TIR}}$, while the downward component $F^{\downarrow,\mathrm{TIR}}$ is essentially zero. The magnitude of $F^{\uparrow,\mathrm{TIR}}$ is primarily driven by the cloud absorption optical depth, the surface temperature $T_{\mathrm{srf}}$, and the (ice) cloud temperature $T_{\mathrm{cld,ice}}$ or ice cloud altitude $z_{\mathrm{ice}}$ (Corti and Peter, 2009). Assuming the Earth surface is a black body, the outgoing $F^{\uparrow}_{\mathrm{TIR}}$ at TOA could be

calculated, in a first order approximation, by the Stefan–Boltzmann-law:

$$F_{cf}^{\uparrow} = \sigma \cdot \epsilon \cdot T^4, \tag{15}$$

which is obtained by integrating the Planck function over all wavelengths and $2\pi$ of a hemispheric solid angle. In Eq. 15 the Stefan–Boltzmann-constant is represented by $\sigma = 5.67 \cdot 10^{-8}\,\mathrm{W\,m^{-2}\,K^{-4}}$ and the emissivity $\epsilon = 1$ of a blackbody. In reality however, the Earth acts as a gray-body ($\epsilon \neq 1$) and the surrounding atmosphere must be taken into account.

Absorption of radiation in the atmosphere in the TIR wavelength range depends on wavelength and atmospheric composition. The primary components that control absorption are water vapor and carbon dioxide ($CO_2$) (Liou, 1992). While $CO_2$ is well-mixed and thus approximately constant in space and time, WV is highly variable. Furthermore, the amount of WV is linked to the temperature in the AP via the Clausius–Clapeyron-equation (Corti and Peter, 2009). The lowermost values of the AP are also influenced by $T_{srf}$. Due to these interactions, Corti and Peter (2009) developed a model to estimate TIR irradiances and the resulting CRE. The model was derived by fitting RT simulations, which cover a wide range of environmental conditions, to Eq. 15, which leads to:

$$F_{cf}^{\uparrow *} \approx \sigma^* \cdot \epsilon \cdot T_{srf}^{k^*}, \tag{16}$$

with $\sigma^* = 1.607 \cdot 10^{-4}\,\mathrm{W\,m^{-2}\,K^{-2.528}}$ and $k^* = 2.528$. $F_{cf}^{\uparrow *}$ represents surface surface emission with $T_{srf}$ and atmospheric absorption.

Clouds in the atmosphere can be approximated by semi-transparent blackbodies that partly absorb and re-emit radiation according the Stefan–Boltzmann-law. The emissivity $\epsilon$ of a cloud depends on $\tau$, which in turn depends on the wavelength (Stephens et al., 1990). $F_{TIR}^{\uparrow}$ in the cloudy case can be estimated with:

$$F_{TIR}^{\uparrow} = (1 - \epsilon^*) \cdot \sigma^* \cdot T_{srf}^{k^*} + \epsilon^* \cdot \sigma^* \cdot T_{cld}^{k^*}, \tag{17}$$

with $\sigma^*$ and $k^*$ for cloud-free conditions. $\epsilon$ can be approximated by

$$\epsilon \approx 1 - \exp{-\delta \cdot \tau}, \tag{18}$$

where $\delta = D \cdot (1 = \tilde{\omega})$ and $D \approx 1.66$, relying on the zero-scattering assumption (Stephens et al., 1990), and an effective emissivity $\epsilon^*$ that is also derived from their RT simulations. Finally, the TIR RE of a cloud above a surface can be approximated with:

$$\Delta F_{tir} = F_{TIR,c} - F_{TIR,cf} \approx$$
$$\sigma^* (T_{srf}^{k^*} - T_{cld}^{k^*}) \cdot (1 - \exp{(-\delta^* \cdot \tau)}), \tag{19}$$

with $\delta^* = 0.75$. It follows from Eq. 19 that the forcing of a cloud, with constant $\tau$, is proportional to the temperature difference between cloud and surface.

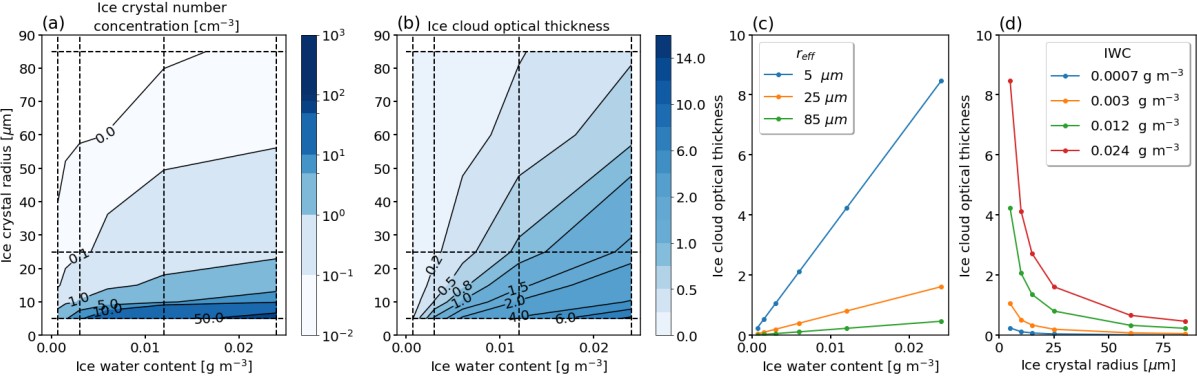

**Figure 1. (a–b)** Calculated ice crystal number concentration $N_{ice}$ (in cm$^{-3}$) and simulated cloud optical thickness $\tau_{ice}$ at 550 nm wavelength as a function of ice water content IWC (in g m$^{-3}$) and effective crystal radius $r_{eff}$ (in $\mu$m) assuming droxtals. A cloud geometric thickness d$z$ of 1000 m is selected. **(c–d)** Cross-sections along lines of constant $r_{eff}$ or IWC that are indicated as dashed lines in panel a and b, respectively.

## 3 Results

We first provide an overview of how $r_{eff}$ and IWC determine the cloud optical and microphysical properties. Figure 1a–d illustrates the dependence of $N_{ice}$ and $\tau_{ice}$ as a function of $r_{eff}$ and IWC. $N_{ice}$ is approximated by Eq. 14, assuming droxtals (almost spherical ice crystals), a mono-disperse particle size distribution, and a cloud geometric thickness d$z$ of 1000 m. The ice cloud optical thickness $\tau_{ice}$ at 550 nm wavelength is directly obtained from the libRadtran verbose output using optical properties of droxtals. The largest $N_{ice}$ values result from the smallest ice crystals sizes ($r_{eff} < 10$ $\mu$m), particularly in combination with large IWC (Fig. 1a). For combinations of small $r_{eff} < 15$ $\mu$m and large IWC, $N_{ice}$ is most sensitive to $r_{eff}$, which is indicated by the narrowing contour lines that align along the $x$-axis. For a constant $r_{eff}$ of 5 $\mu$m, the estimated $N_{ice}$ ranges from 1 to over 80 cm$^{-3}$. Such concentrations of $N_{ice} > 80$ cm$^{-3}$ are rarely observed in natural cirrus though they can occur in very young contrails and contrail-induced cirrus (Krämer et al., 2016). Generally smaller $N_{ice}$ and a reduced sensitivity to $r_{eff}$ and IWC is found for $r_{eff} > 20$ $\mu$m, where $N_{ice}$ mostly ranges below 10 cm$^{-3}$.

The inherent dependencies of $N_{ice}$ presented in Fig. 1a are also found in the distribution of the ice cloud optical thickness $\tau_{ice}$ at 550 nm shown in Fig. 1b. Following lines of constant $r_{eff}$ (Fig. 1c), the increase in IWC corresponds to a linear increase in $N_{ice}$ and, therefore, to a gain in the total scattering and absorption particle cross-sections. The absorption of radiation by liquid water and ice (as characterized by the complex refractive index) at 550 nm wavelength is weak and, therefore, scattering dominates $\tau_{ice}$. Alternatively, going along lines of constant IWC towards larger $r_{eff}$ leads to a decrease in $N_{ice}$ and a related decrease of the total scattering particle cross-section (cloud albedo effect, Fig. 1d). This effect is most effective for larger IWC (optically thick clouds) and is less pronounced for clouds with smaller IWC.

To reduce the multi-dimensionality, for each of the eight parameters a reference is defined by selecting either the minimum or maximum value from the parameter space. The reference parameters are selected to highlight the upper or lower range of each parameter and the spanned variation, and to define the reference for the fixed parameters. The reference parameters are

given by $\theta = 0°$, $T_{cld,ice} = 219$ K, $\alpha_{srf} = 0$, $T_{srf} = 299.7$ K, $r_{eff} = 85$ $\mu$m, and $\tau_{liq} = 0$ (no liquid water cloud). For IWC we use an intermediate value of 0.024 g m$^{-3}$ because together with a d$z$ of 1000 m and an $r_{eff}$ of 85 $\mu$m this leads to a $\tau_{ice}$ of 0.46 at 550 nm wavelength, which is representative for contrails and young cirrus (Iwabuchi et al., 2012). Otherwise electing the minimum or maximum IWC in combination with $r_{eff}$ of 85 $\mu$m would lead to high or low $\tau_{ice}$ that are not representative for contrails. For ice crystal shape, we select aggregates as the reference. We particularly emphasize that the defined references are not representative of any particular cloud situation, but are a useful point of comparison to assess the impact of a given parameter on the diversity of cloud RE.

Using the defined reference, Fig. 2a–c shows solar, TIR, and net $\Delta F$, respectively (similar to Meerkötter et al. (1999)). First, the influence of variations in $\theta$ is investigated in order to sample the diurnal cycle and its variation as a function of latitude. For all Sun geometries, $\Delta F_{sol}$ is negative and, therefore, the cirrus has a cooling effect in the solar spectrum on the atmosphere-surface system. $\Delta F_{sol}$ intensifies (i.e., becomes more negative) with increasing $\theta$ as the length of the optical path through the cloud, $s = \Delta z / \cos\theta$, increases, which is accompanied by enhanced scattering (and thus upward directed scattering) of the incoming radiation (Wendisch et al., 2005). In addition, a lower fraction of the incident radiation is scattered towards the surface but scattered upward to space. This is due to the strong forward peak in the ice crystal phase function $\mathcal{P}$ that decreases sharply for $\Theta > 10°$ (see in Appendix Fig. D1). An exception appears for $\theta$ of 85°, where $\Delta F_{sol}$ is smallest. Variations in $\theta$ lead to $\Delta F_{sol}$ between $-55.9$ and $-27.5$ W m$^{-2}$. As expected, $\Delta F_{tir}$ is unaffected by the Sun position with a constant $\Delta F_{tir}$ = 46.0 W m$^{-2}$. The resulting sensitivity of $\Delta F_{net}$ is driven by $\Delta F_{sol}$ with $\Delta F_{net}$ between $-9.9$ and 18.5 W m$^{-2}$. During nighttime there is no contribution from $\Delta F_{sol}$ leading to a constant, positive $\Delta F_{net}$ = 46.0 W m$^{-2}$ (leading to a warming).

As expected, variations in $r_{eff}$ have the largest effect on the solar, TIR, and net $\Delta F$, as $N_{ice}$ relates to $r_{eff}$ by the power of $-\beta$, which depends on the ice crystal shape (see Sec. 2.3 and Eq. 14). Increasing $r_{eff}$ from 5 to 85 $\mu$m leads to $\Delta F_{sol}$ between $-599.5$ and $-50.2$ W m$^{-2}$. The distribution of $\Delta F_{tir}$ has a minimum and maximum of 46.0 and 149.8 W m$^{-2}$, respectively. $\Delta F_{sol}$ dominates $\Delta F_{tir}$ and results in values of $\Delta F_{net}$ ranging from $-449.8$ to $-4.2$ W m$^{-2}$.

Variations in IWC affect solar, TIR, and net $\Delta F$. Generally, an increase in IWC (increase in $\tau_{ice}$ for fixed $r_{eff}$), enhances total scattering and absorption particle cross-sections and, therefore, intensifies the cooling in the solar (more negative $\Delta F$, cloud albedo effect) and the TIR heating (more positive $\Delta F$). $\Delta F_{sol}$ ranges from $-191.1$ to $-1.5$ W m$^{-2}$, with $\Delta F_{sol}$ = $-50.2$ W m$^{-2}$ obtained for the reference IWC. The distribution of $\Delta F_{tir}$ spans values between 1.8 and 112.7 W m$^{-2}$, leading to $\Delta F_{net}$ from $-78.4$ to 1.1 W m$^{-2}$. The $\Delta F$ given above correspond to a varying IWC and assume $r_{eff}$ = 85 $\mu$m. For smaller $r_{eff}$ $\Delta F$ increases and thus increases the range of solar, TIR and net $\Delta F$. In addition, the IWC becomes dominant over $r_{eff}$ in Fig. 2, when selecting a reference with smaller $r_{eff}$.

Variations in $\alpha_{srf}$ impact only the solar spectrum, as expected, with $\Delta F_{sol}$ between $-50.2$ and 15.4 W m$^{-2}$. The most negative RE appears over non-reflective surfaces and decreases with increasing $\alpha_{srf}$, due to the decrease in contrast between the surface and the cirrus. In cases where $\alpha_{srf}$ exceeds the cloud albedo, $\Delta F_{sol}$ becomes positive. For the optical thin reference this is the case over a fully sea ice covered area with $\alpha_{srf} \approx 1$. The TIR component remains almost unaffected with $\Delta F_{tir}$ between 39.5 and 46 W m$^{-2}$. Together with the decreasing cooling effect in the solar, the warming in the TIR mostly dominates and leads to $\Delta F_{net}$ ranging between -4.2 and 55.0 W m$^{-2}$.

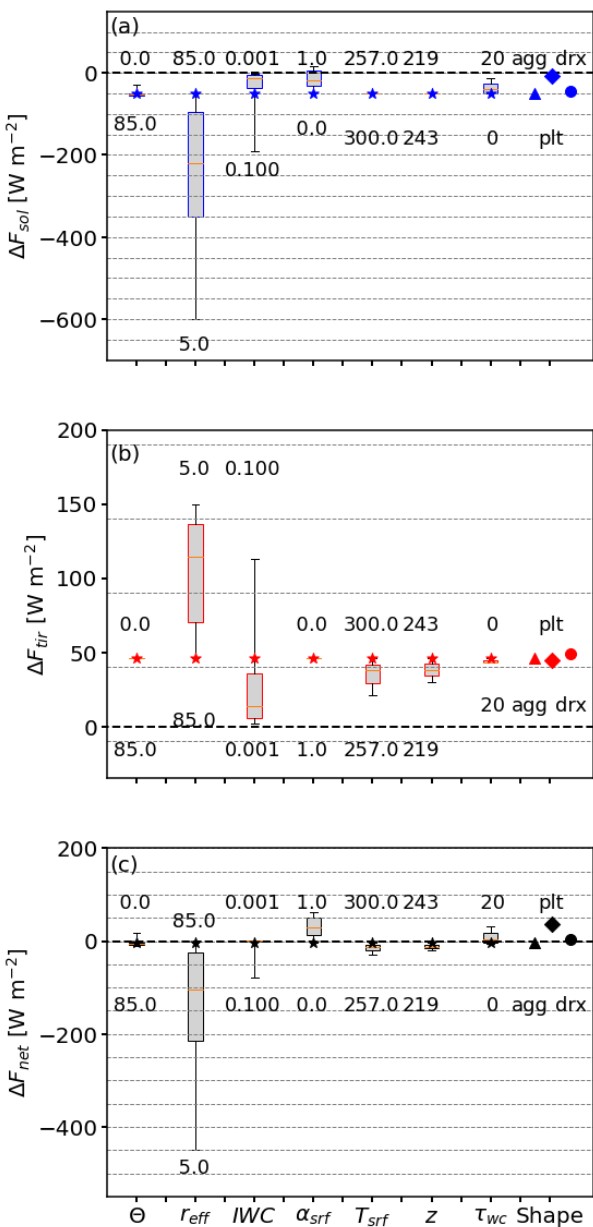

**Figure 2. (a–c)** Box and whisker plot of solar, TIR, and net $\Delta F$ (in W m$^{-2}$) due to the variation of the parameters indicated as the $x$-axis. The boxes represent the $25^{\text{th}}-$ and $75^{\text{th}}$-percentiles, while the whiskers indicate the minimum and maximum values. Median values are given in each box by horizontal, orange lines. The stars indicate the reference with solar zenith angle $\theta = 0°$, effective radius $r_{\text{eff}} = 85$ $\mu$m, ice water content IWC $= 0.024$ g m$^{-3}$, surface albedo $\alpha_{\text{srf}} = 0$, surface temperature $T_{\text{srf}} = 299.7$ K, ice cloud temperature $T_{\text{cld,ice}} = 219$ K, and liquid water cloud optical thickness $\tau_{\text{liq}} = 0$. Minimum and maximum of the parameter ranges are given by the numbers. Plot idea adapted from Meerkötter et al. (1999).

The influence of a varying surface temperature $T_{\mathrm{srf}}$ or cirrus temperature $T_{\mathrm{cld,ice}}$ (related to cloud base altitude), are inves-
tigated for a cloud scenario with a solar surface albedo $\alpha_{\mathrm{srf}}$ set to 0. Varying surface temperature $T_{\mathrm{srf}}$ or cirrus temperature
$T_{\mathrm{cld,ice}}$ (related to cloud base altitude), $\Delta F_{\mathrm{sol}}$ remains almost constant with a minimum and maximum $\Delta F_{\mathrm{sol}}$ for both param-
eters of $-50.2$ and $-49.2\,\mathrm{W\,m^{-2}}$, respectively. These small differences are due to changes in molecular absorption, which
results from the variations in the relative humidity profile as the profile depends on the selected $T_{\mathrm{srf}}$. A noticeable effect is
found for $\Delta F_{\mathrm{tir}}$, which is impacted by variations in $T_{\mathrm{cld,ice}}$ and $T_{\mathrm{srf}}$. While decreasing $T_{\mathrm{cld,ice}}$ from 243 to 219 K lowers $\Delta F_{\mathrm{tir}}$
from 46 to 29.9 $\mathrm{W\,m^{-2}}$, a decrease in $T_{\mathrm{srf}}$ from 300 to 257 K reduces $\Delta F_{\mathrm{tir}}$ from 46 to 20.8 $\mathrm{W\,m^{-2}}$. Consequently, $\Delta F_{\mathrm{tir}}$
determines the response of the resulting $\Delta F_{\mathrm{net}}$, which spans from $-4.2$ to $-19.4\,\mathrm{W\,m^{-2}}$ for $T_{\mathrm{cld,ice}}$ and $-28.7$ to $-4.2\,\mathrm{W\,m^{-2}}$
for $T_{\mathrm{srf}}$. The greater influence of $T_{\mathrm{srf}}$ on $\Delta F_{\mathrm{tir}}$ and $\Delta F_{\mathrm{net}}$ is explained simply by the greater variation of the input.

A second cloud layer is considered by inserting a liquid water cloud with a cloud top altitude $z_{\mathrm{base}} = 1500\,\mathrm{m}$ and a geometric
thickness $\mathrm{d}z = 500\,\mathrm{m}$. Figure 2 shows that this second cloud influences both components $\Delta F_{\mathrm{sol}}$ and $\Delta F_{\mathrm{tir}}$. Generally speaking,
the liquid water cloud enhances the fraction of solar, upward directed radiation compared to a dark surface. With increasing $\tau_{\mathrm{liq}}$
(increase in LWC) $\alpha_{\mathrm{cld,ice}}$ exceeds $\alpha_{\mathrm{srf}}$, which lowers the albedo contrast between the ice cloud and the surface for most of the
parameter combinations. This minimizes solar RE and leads to a minimum of $-51.1\,\mathrm{W\,m^{-2}}$ and a maximum of $-11.6\,\mathrm{W\,m^{-2}}$.
For the TIR part the increase in LWC masks the influence of the underlying surface by absorbing the upward TIR radiation from
the surface and re-emitting radiation at the liquid water cloud temperature. This leads to $\Delta F_{\mathrm{tir}}$ between 43.2 and 46.0 $\mathrm{W\,m^{-2}}$.
The resulting $\Delta F_{\mathrm{net}}$ is characterized by a minimum and maximum of $-6.5$ and $31.6\,\mathrm{W\,m^{-2}}$ primarily impacted by the solar
component.

The parameter study is complemented by investigating the effect of prescribing three different ice crystal shapes. The vari-
ation in $\Delta F_{\mathrm{sol}}$ due to the transition from almost spherical (droxtals) to non-spherical crystals (aggregates) leads to a relative
change in $\Delta F_{\mathrm{sol}}$ that is, in terms of RE, comparable to a variation in $\theta$. The strongest cooling effect (negative $\Delta F_{\mathrm{sol}}$) is found
for aggregates with $-50.2\,\mathrm{W\,m^{-2}}$ and decreases for droxtals and plates to $-44.3$ and $-8.6\,\mathrm{W\,m^{-2}}$, respectively. Ice crystal
shape also impacts $\Delta F_{\mathrm{tir}}$. Aggregates lead to $\Delta F_{\mathrm{tir}}$ of 46 $\mathrm{W\,m^{-2}}$, while plates and droxtals can cause a $\Delta F_{\mathrm{tir}}$ of 44.5 and
48.9 $\mathrm{W\,m^{-2}}$, respectively. Consequently, the largest $\Delta F_{\mathrm{net}}$ with 35.8 $\mathrm{W\,m^{-2}}$ is found for plates and followed, in decreasing
order, by droxtals and aggregates with 4.5 and $-4.2\,\mathrm{W\,m^{-2}}$, respectively. As mentioned in the introduction, the uncertainty in
the ice crystal shape causes uncertainties in the calculated $\Delta F$. Nevertheless, using three different ice crystal shapes for the ir-
radiance simulations shows that the shape-specific scattering properties are of lesser importance compared to other parameters
like the ice crystal size (distribution), the IWC, or surface properties.

The presented analysis of solar, TIR, and net $\Delta F$ sensitivity on the selected input parameters generally agrees with the
results from Meerkötter et al. (1999). We found differences in the importance of the parameters, which are explained by the
fact that our simulations span a larger and different parameter range, for example in $r_{\mathrm{eff}}$, IWC, and $T_{\mathrm{srf}}$. Selecting cloud
parameters ($\theta = 30°$, $T_{\mathrm{cld,ice}} = 231$ K, $\alpha_{\mathrm{srf}} = 0.15$, $T_{\mathrm{srf}} = 288$ K, $r_{\mathrm{eff}} = 10\,\mu\mathrm{m}$, and $\tau_{\mathrm{liq}} = 0$) whether case A in (Meerkötter
et al., 1999), we find that the IWC becomes the driving parameter, which then agrees with the results from Meerkötter et al.
(1999). However, a more quantitative comparison between Meerkötter et al. (1999) is difficult as the parameters that best match
are not identical. Even by choosing similar cloud parameters, by matching the IWP and selecting $r_{\mathrm{eff}}$ to yield $\tau_{\mathrm{ice}} \approx 0.52$, the

simulated clouds and cloud case A from Meerkötter et al. (1999) differ in d$z$, which impacts $\Delta F_{sol}$ and $\Delta F_{tir}$ with different
intensity.

It is further emphasized that the presented $\Delta F_{net}$ is representative for daytime situations only, when the Sun is above the horizon. In the absence of solar illumination during nighttime, the net effect is entirely determined by and equal to $\Delta F_{tir}$, which is positive (warming effect) in all simulation cases. Accordingly, all simulated cloud cases do have a net warming effect at night. For a more in-depth analysis, the subsequent plots focus on the impact of each individual parameter.

### 3.1   Sensitivity on ice crystal shape

One difficulty of RT simulations in ice clouds is the uncertainty about the dominating ice crystal shape, which is commonly unknown and, therefore, a general ice crystal shape has to be assumed (Kahnert et al., 2008). Scattering and absorption by an ice crystal is characterized by its orientation, complex refractive index of ice, the wavelength of the incident light, shape, size, and the resulting asymmetry parameter. The asymmetry parameter is a measure of the asymmetry of the phase function $\mathcal{P}$
between forward and backward scattering (Macke et al., 1998; Fu, 2007). $\mathcal{P}$ provides the angular distribution of the scattered direction in relation to the incident light. For example, in case of idealized hexagonal ice crystals and wavelength below 1.4 $\mu$m, the asymmetry parameter is primarily determined by the ice crystal shape / aspect ratio but for wavelength larger then 1.4 $\mu$m the asymmetry parameter also depends on the ice crystal size (Fu, 2007; Yang and Fu, 2009; van Diedenhoven et al., 2012). Consequently, the assumption of an ice crystal habit and ice crystal size, with related aspect ratio, are vital information to
estimate the ice cloud RE. Furthermore, the ice optical properties by Yang et al. (2010, 2013), which are used for the RT simulations in the present study, based on a coupling of the maximum diameter of the ice crystal and the aspect ratio, with the later one being different for each crystal shape. This impacts the RT of different ice clouds with varying IWC and $r_{eff}$.

Subsequently, the shape-effect is quantified using Eq. 20 and relative differences in $\Delta F$ are given with respect to crystals with the same $r_{eff}$ in relation to the $\Delta F$ simulated for aggregates. Figure 3a–c show $\Delta F_{sol}$ as a function of IWC, separated for
crystal shape, $r_{eff}$, and three selected $\theta$. For simplicity $\alpha_{srf}$ and $\tau_{liq}$ are set to zero in this discussion.

The strongest $\Delta F_{sol}$ is found for aggregates (green) with $r_{eff} = 5\,\mu$m with the Sun at zenith ($\theta = 0°$, Fig. 3a). A lower cooling effect in the solar spectrum is found for droxtals (orange) and plates (blue) with same $r_{eff}$. The order of $\Delta F_{sol}$ remains constant for increasing $r_{eff}$.

The spread in $\Delta F_{sol}$ across crystal shapes with the same $r_{eff}$ and IWC can be interpreted as a potential uncertainty in $\Delta F_{sol}$
due to the ice crystal shape. One has to keep in mind that the differences partially result from deviating crystal size distributions as these depend on the selected crystal shape. Macke et al. (1998) showed that, in the solar wavelength range, the crystal shape is the main driver and the actual ice PSD has only a minor effect on $\Delta F_{sol}$. Nevertheless, Mitchell (2002) and Mitchell et al. (2011) found that the PSD also has a considerable impact on $\Delta F_{tir}$, leading to differences of up to 48% in the single-scattering albedo, when switching between PSD.

To quantify the deviations resulting from the ice crystal shape, Fig. 3d–f show absolute and Fig. 3g–i present relative differences of $\Delta F_{sol}$ of droxtals and plates with respect to aggregates. For $\theta = 0°$ the largest absolute deviation is found for plates with $r_{eff}$ of 25 $\mu$m and highest IWC with an absolute range of up to 250 W m$^{-2}$ ($r_{eff} = 25$ $\mu$m, $\theta = 0°$, $\tau_{ice} = 6.6$),

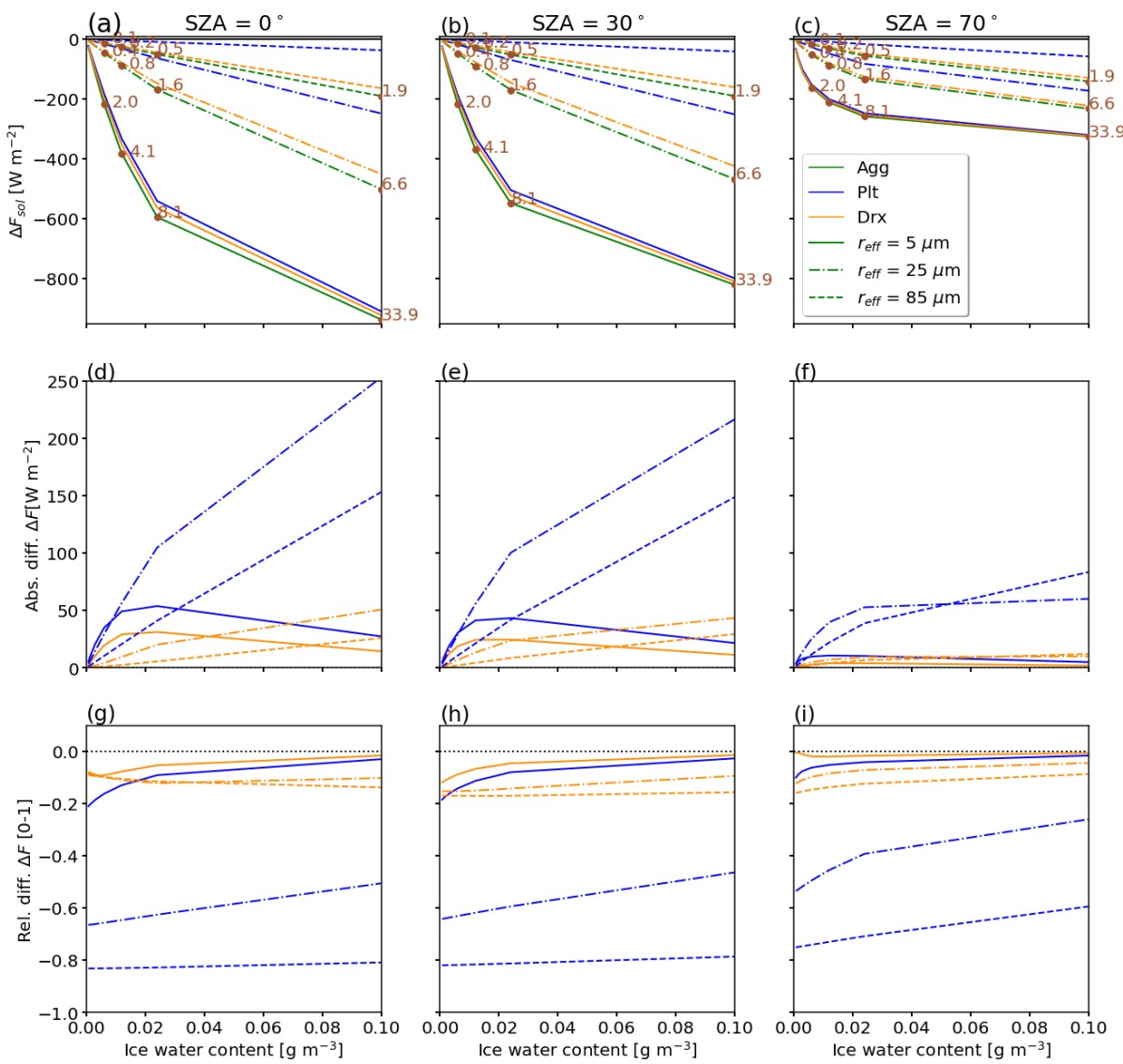

**Figure 3. (a–c)** Solar radiative effect $\Delta F_{\text{sol}}$ (in W m$^{-2}$) as a function of ice water content IWC for three values of solar zenith angle $\theta$ of $0°$, $30°$, and $70°$. Three ice crystal radii $r_{\text{eff}}$ of 5 (solid), 25 (dash-dot), and 85 $\mu$m (dashed) are indicated. The ice crystal shape is color-coded with aggregates 'Agg', plates 'Plt', and droxtals 'drx' given in green, blue, and orange, respectively. **(d–f)** show absolute difference and **(g–i)** relative difference between $\Delta F_{\text{sol}}$ of droxtals and plates with respect to aggregates with the same crystal radius. The numbers indicate the optical thickness simulated for the reference that contains ice aggregates.

corresponding to a relative difference of 58 %. Relative deviations reach even larger values, e.g., when the cloud is optically thinner and $\Delta F_{\text{sol}}$ gets smaller. In case of plates the relative deviations range from $-20\%$ ($r_{\text{eff}} = 5$ $\mu$m) to $-82\%$ ($r_{\text{eff}} =$

$\mu$m). The large absolute and relative deviations between plates and aggregates in $\Delta F_{\text{sol}}$ and later $\Delta F_{\text{net}}$ appear because plates are characterized by the smallest reflectance and absorption efficiency (Key et al., 2002; Yang et al., 2005). The absolute differences among droxtals and aggregates are smaller. With increasing IWC the absolute ranges quickly reach a maximum of 27 W m$^{-2}$ at IWC of 0.024 g m$^{-3}$ and decrease towards the largest IWC. The associated relative deviations are also smaller compared to plates, ranging between $-3\%$ ($r_{\text{eff}} = 5$ $\mu$m) and $-18\%$ ($r_{\text{eff}} = 85$ $\mu$m).

Another characteristic of the absolute range of $\Delta F_{\text{sol}}$ is the steep slope for $\theta = 0°$ over the entire range of IWC. For illumination geometries with the Sun closer to the horizon, particularly $\theta = 70°$, the behavior of absolute range in $\Delta F_{\text{sol}}$ is characterized by a rapid increase and convergence towards a maximum. At a certain IWC and related $\tau_{\text{ice}}$, the slant optical path and cloud-radiation interactions are dominated by multiple scattering that suppresses single-scattering effects of individual ice crystal shape, hence, reducing the absolute and relative difference resulting from the choice of the ice crystal shape. This is

supported by earlier observations and simulations for example by Wendisch et al. (2005), who showed that for large $\theta$ and multiple-scattering the shape effect becomes less prominent.

    Next, we consider the solar, TIR, and net $\Delta F$ at $\theta = 30°$ (Fig. 4). The left most column for $\Delta F_{\text{sol}}$ is identical to the middle column in Fig. 3. In the TIR, the largest $\Delta F_{\text{tir}}$ is generally found for smallest crystals (5 $\mu$m) and highest IWC in decreasing order from droxtals, plates, and aggregates. With increasing crystal size the order changes to droxtal, aggregates, and plates,

and the absolute values of $\Delta F_{\text{tir}}$ decrease. The largest $\Delta F_{\text{tir}}$ range of 130 W m$^{-2}$ is found for clouds with IWC between 0.024 and 0.1 g m$^{-3}$ caused by droxtals. For thin clouds with IWC $< 0.04$ g m$^{-3}$ the largest absolute range $R_{\Delta F,\text{tir}}$ of around 6.5 W m$^{-2}$ appears for $r_{\text{eff}}$ of 5 and 25 $\mu$m, which is shifting towards larger IWC with increasing $r_{\text{eff}}$ and vanishes for the largest crystals with $r_{\text{eff}}$ of 85 $\mu$m. The relative differences are largest for the optically thinnest clouds and decrease with increasing IWC. While droxtals are characterized by relative differences close to 0 % ($r_{\text{eff}} = 5$ $\mu$m; IWC $= 0.1$ g m$^{-3}$) and 18 %

($r_{\text{eff}} = 25$ $\mu$m; IWC $= 0.007$ g m$^{-3}$), plates lead to relative differences between 9 % ($r_{\text{eff}} = 5$ $\mu$m; IWC $= 0.007$ g m$^{-3}$) and $-5\%$ ($r_{\text{eff}} = 85$ $\mu$m, IWC 0.007 g m$^{-3}$). The TIR RE of the optically thickest cloud is independent on ice crystal shape, which is addressed to multiple scattering.

    For all IWC and $r_{\text{eff}}$, $\Delta F_{\text{sol}}$ is generally larger than $\Delta F_{\text{tir}}$ and, therefore, dominates resulting $\Delta F_{\text{net}}$ (Figure 4c, f). Consequently, $\Delta F_{\text{net}}$ and absolute ranges among the ice crystal shapes follow the distributions from $\Delta F_{\text{sol}}$. The largest relative

deviations are found for the optically thinnest clouds, where $\Delta F_{\text{net}}$ is generally small. In these cases of optically thin clouds consisting of the smallest crystals ($r_{\text{eff}} = 5$ $\mu$m) the relative deviations exceed the relative difference for optically thick clouds with the same crystal size by a factor of 10.

    The analysis of all simulations shows that the crystal shape assumption on the cirrus RE is small compared to other parameters particularly IWC or $r_{\text{eff}}$ (see Fig 2). However, we found a larger variability in $\Delta F_{\text{sol}}$ and the resulting $\Delta F_{\text{net}}$, i.e.,

weather a contrail has a net warming or cooling effect compared to $\Delta F_{\text{tir}}$. For the defined reference consisting of aggregates, a $\Delta F_{\text{sol}}$ of -50.2 W m$^{-2}$ was simulated, while for plates and droxtals values of $\Delta F_{\text{sol}}$ of $-8.6$ and $-44.3$ W m$^{-2}$ were obtained, respectively. The impact of the crystal shape is less pronounced in the TIR wavelength range with $\Delta F_{\text{tir}}$ of 46, 44.5, and

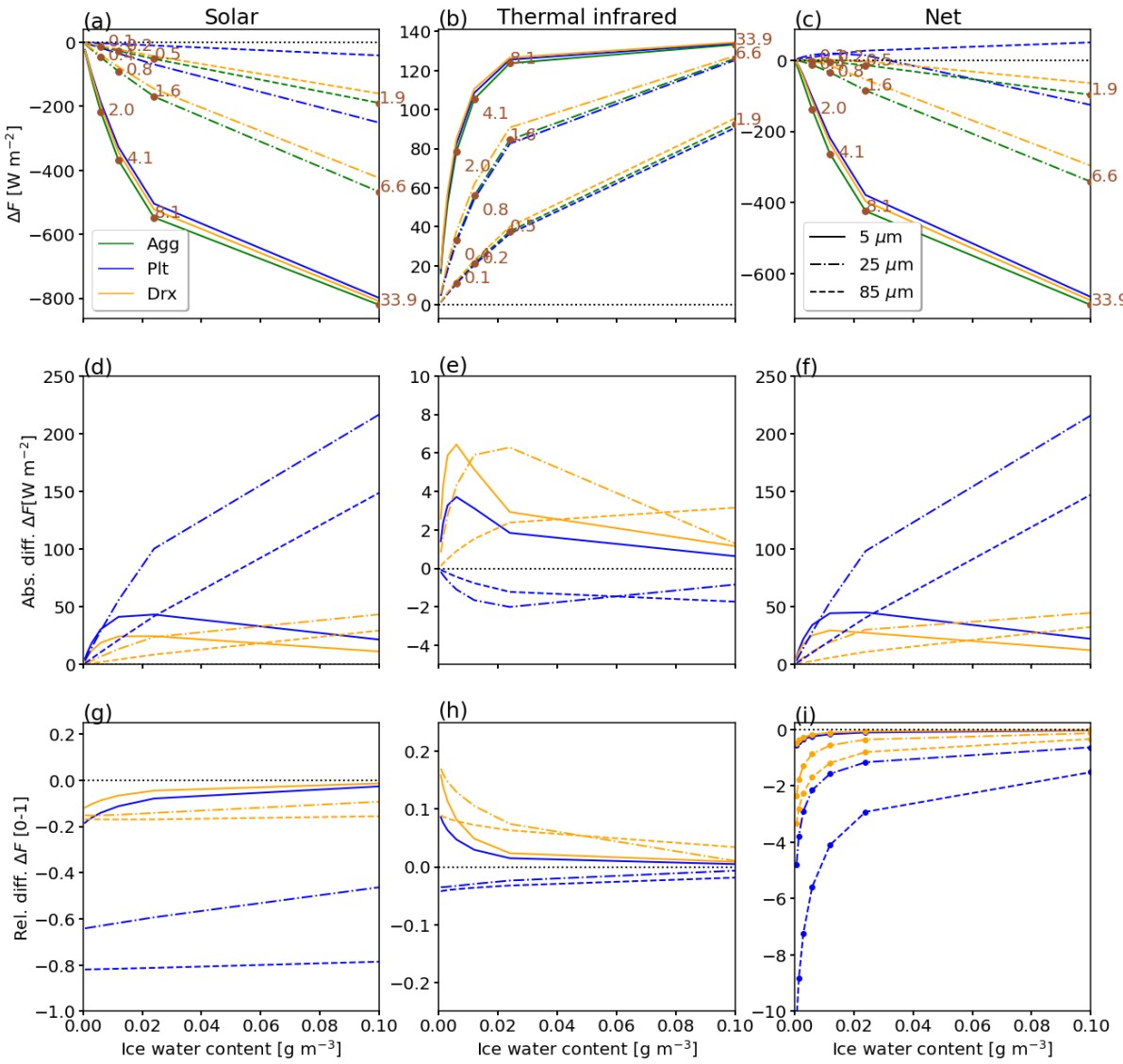

**Figure 4.** Same as Fig. 3 but for solar zenith angle $\theta = 30°$, and $\Delta F_{\mathrm{sol}}$ (left), $\Delta F_{\mathrm{tir}}$ (middle), and $\Delta F_{\mathrm{net}}$ (right).

48.9 W m$^{-2}$ for aggregates, plates, and droxtals, respectively. The variation of $\Delta F_{\text{sol}}$ propagates into $\Delta F_{\text{net}}$ with $-4.2$, 35.9 and 4.6 W m$^{-2}$ for aggregate, plates, and droxtals, respectively. Based on the presented simulations, we found larger maximum variations in $\Delta F_{\text{sol}}$, $\Delta F_{\text{tir}}$, and $\Delta F_{\text{net}}$ of 41.6, 4.4, and 40 W m$^{-2}$, respectively, compared to Meerkötter et al. (1999). They found variations in $\Delta F_{\text{sol}}$, $\Delta F_{\text{tir}}$, and $\Delta F_{\text{net}}$ of 2, 6, and 7 W m$^{-2}$, respectively. The difference are explained by the selected reference (Meerkötter et al., 1999). However, selecting cloud parameters similar to the reference cloud of Meerkötter et al. (1999), we still found larger maximum variations $\Delta F_{\text{sol}}$, $\Delta F_{\text{tir}}$, and $\Delta F_{\text{net}}$ of 17.3, 4.2, and 17.9 W m$^{-2}$, respectively. This is attributed to the remaining differences among the selected reference values.

## 3.2 Sensitivity on solar zenith angle and surface albedo

In this section the impact of each parameter is estimated by fixing one parameter at a time (represented by the $x$-axis), while the others can vary. For example, in case of $\theta$, all simulations, for steps of $\theta$ given in Table 4, are extracted from the 8-D hypercube. The extracted sub-sample, in the example for a specific $\theta$, is used to calculate and visualize the distributions of solar, TIR, and net $\Delta F$. This strategy can be interpreted as a type of sub-sampling, by averaging all unfixed parameters to project $\Delta F$ onto the one-dimensional space. The impact of each parameter is further quantified by the minimum and maximum RF. We define the full range of $\Delta F$ by:

$$R_{\Delta F} = max\{\Delta F\} - min\{\Delta F\}, \tag{20}$$

with $max\{\Delta F\}$ and $min\{\Delta F\}$ the maximum and minimum of $\Delta F$ across the sub-sampled distributions, respectively. As $R_{\Delta F}$ is susceptible to outliers, we further characterize the width of a distribution by the inter-quartile range, which is defined as the difference between the 75$^{\text{th}}$ ($Q_{75\%}$) and 25$^{\text{th}}$ ($Q_{25\%}$) percentiles of $\Delta F$:

$$Q_{\Delta F} = Q_{75\%}(\Delta F) - Q_{25\%}(\Delta F) \tag{21}$$

Variations in $\theta$ are caused by the diurnal and seasonal cycle of the Earth, or variations along the longitude at a given time. Figure 5a shows distributions of solar $\Delta F_{\text{sol}}$ for $\theta = 0°$, ranging from $-944.5$ W m$^{-2}$ (high IWC) to 78.0 W m$^{-2}$ (high $\alpha_{\text{srf}}$). For simulated $\theta < 85°$, the median values range from $-11.9$ to $-12.9$ W m$^{-2}$ with an intensification of $\Delta F_{\text{sol}}$ towards larger $\theta$. At the same time, the upper maxima of $\Delta F_{\text{sol}}$ are shifted towards zero, which is a combination of three effects: i) a decreasing downward irradiance at TOA with increasing $\theta$; ii) an increasing optical path length $s$ through the cloud with increasing $\theta$ and the corresponding increase in scattering; and iii) an increase in upward scattered radiation with increasing $\theta$ as the light rays get slanted and a larger fraction of radiation from the forward scattering range is directed upwards. Effects i) and ii) compete and are dominated by effect iii). The combination of effects i) to iii) also reduces the inter-quantile for larger $\theta$ and indicates a reduced influence of the other free parameters on $\Delta F_{\text{sol}}$. However, the smallest $\Delta F_{\text{sol}}$ is calculated for $\theta$ of 85° and is caused by the reduced side-ward scattering of ice crystals.

The value of $\theta$ where $\Delta F_{\text{sol}}$ is most intense depends on $\alpha_{\text{srf}}$ and is typically located between 50° and 70° (Markowicz and Witek, 2011). The maximum in $\Delta F_{\text{sol}}$ and the corresponding $\theta$ are explained by the strong forward scattering peak of ice crystals and the resulting weak backscattering (Haywood and Shine, 1997; Myhre and Stordal, 2001). To further elaborate on

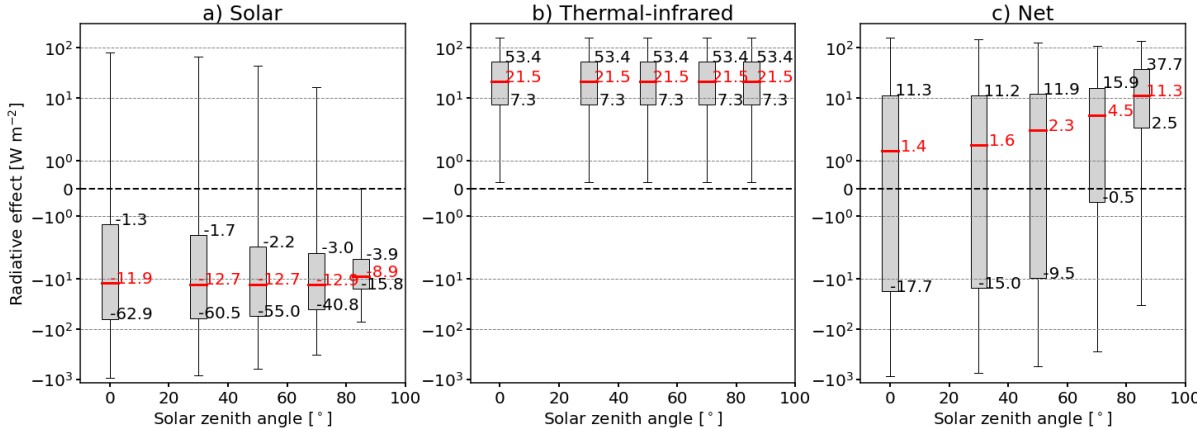

**Figure 5.** Box plots of **(a)** solar, **(b)** TIR, and **(c)** net $\Delta F$ (in W m$^{-2}$) as a function of the solar zenith angle $\theta$. Median values are indicated in red, the 25 % – 75 % range is represented by the gray boxes, and the 10 % and 90 %-percentiles are given by the whiskers. Red and black numbers indicate the 25$^{\text{th}}$- and 75$^{\text{th}}$ percentiles, and the median value, respectively. Note the logarithmic scale on the $y$-axis.

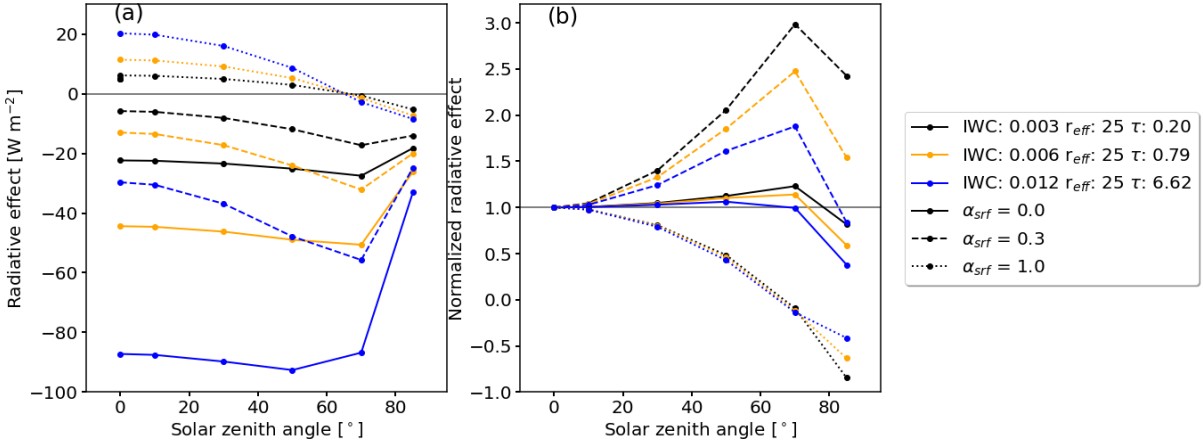

**Figure 6. (a)** Solar radiative effect $\Delta F_{\text{sol}}$ (in W m$^{-2}$) as function of solar zenith angle $\theta$ for three ice clouds with cloud optical thickness $\tau_{\text{ice}}$ of 0.1, 0.4, and 1.6. Effective radius $r_{\text{eff}}$ is given in units of $\mu$m and the ice water content IWC in units of g m$^{-3}$. The cloud is located over surfaces with a surface albedo $\alpha_{\text{srf}}$ of 0, 0.3, and 1. **(b)** Same as **(a)** but normalized with $\Delta F_{\text{sol}}$ of each case at $\theta = 0°$.

the response of $\Delta F_{\mathrm{sol}}$ on large $\theta$, Fig. 6a shows $\Delta F_{\mathrm{sol}}$ as a function of $\theta$ for selected $\tau_{\mathrm{ice}}$ and $\alpha_{\mathrm{srf}}$. For an optically thick cirrus with $\tau_{\mathrm{ice}} = 6.62$ located over a surface with $\alpha_{\mathrm{srf}} = 0$ (blue, solid curve) the maximum $\Delta F_{\mathrm{sol}}$ appears around $\theta = 50°$. For the same cloud above a more reflective surface with $\alpha_{\mathrm{srf}} = 0.3$ (blue, dashed curve) the maximum is shifted towards $\theta = 70°$. Further increasing $\alpha_{\mathrm{srf}}$ to 1 (blue, dotted curve), solar cooling turns into a heating and the strongest solar cooling is found for the largest $\theta$. Figure 6a also shows that the shift in absolute, maximum $\Delta F_{\mathrm{sol}}$ is most pronounced for optically thicker clouds.
However, the largest relative change in $\Delta F_{\mathrm{sol}}$ by varying $\theta$ appears for optically thin clouds (Coakley and Chylek, 1975).

Figure 6b shows $\Delta F_{\mathrm{sol}}$ normalized with the respective $\Delta F_{\mathrm{sol}}$ at $\theta = 0°$. The sensitivity of normalized $\Delta F_{\mathrm{sol}}$ on $\theta$ is most pronounced for optically thin clouds with $\tau_{\mathrm{ice}} = 0.2$ over a moderately reflective surface ($\alpha_{\mathrm{srf}} = 0.3$) (dashed, black). For this combination, $\Delta F_{\mathrm{sol}}$ at $\theta = 70°$ is a factor of 3 larger compared to a Sun overhead ($\theta = 0°$). The same cloud over a non-reflective surface ($\alpha_{\mathrm{srf}} = 0$) reduces the sensitivity leading to a factor of 1.2 in relation to $\Delta F_{\mathrm{sol}}$ at $\theta = 0°$ (solid, black). A similar pattern but with a generally reduced sensitivity is found for the optically thicker cloud case with $\tau_{\mathrm{ice}} = 6.62$. In this case $\Delta F_{\mathrm{sol}}$ is larger by a factor of 1.05 at $\theta = 50°$ (blue, solid) and larger by a factor of 1.7 at $\theta = 70°$ (blue dashed) with respect to a Sun at $\theta = 0°$. The large sensitivity for optically thin clouds is explained by the dominance of single-scattering, where scattering is strongly dependent on the value of the $\mathcal{P}$ at a given scattering angle. When the cloud becomes optically thicker, multiple-scattering processes start to dominate the RT and $\mathcal{P}$ is averaged over a range of scattering angles, reducing the sensitivity on $\theta$. However, while the sensitivity might be largest for optically thin clouds, the absolute $\Delta F_{\mathrm{sol}}$ of optically thin clouds is small compared to clouds with higher $\tau_{\mathrm{ice}}$.

Figure 5b shows that $\Delta F_{\mathrm{tir}}$ is unaffected by $\theta$ leading to a constant median $\Delta F_{\mathrm{tir}}$ of 21.5 W m$^{-2}$. The highest positive values of $\Delta F_{\mathrm{tir}}$ (strongest warming effect) are found for clouds with maximal IWC. The resulting $\Delta F_{\mathrm{net}}$, shown in Fig. 5c, is dominated by a warming in the TIR that leads to median $\Delta F_{\mathrm{net}}$ between 1.4 W m$^{-2}$ and 11.3 W m$^{-2}$, with a minimum of $\Delta F_{\mathrm{net}}$ of $-872.8$ W m$^{-2}$ and maximum of 160.1 W m$^{-2}$. With increasing $\theta$, $\Delta F_{\mathrm{net}}$ increases. This is caused by the shift of the lower minima of $\Delta F_{\mathrm{sol}}$ towards zero, which indicates that a larger fraction of the simulations have a reduced solar cooling effect and, thus, the fraction of simulations with a positive $\Delta F_{\mathrm{net}}$ (net warming) increase. The reduced variability of $\Delta F_{\mathrm{sol}}$ with increasing $\theta$ propagates into the distribution and variability in $\Delta F_{\mathrm{net}}$.

The influence of the underlying surface is shown in Fig. 7. For $\alpha_{\mathrm{srf}} = 0$ the surface absorbs the entire incident solar radiation creating the largest contrast between $\alpha_{\mathrm{srf}}$ and the cloud albedo $\alpha_{\mathrm{cld}}$. When the surface is fully absorbing ($\alpha_{\mathrm{srf}} = 0$), almost all simulated cloud combinations are characterized by a cooling in the solar with $\Delta F_{\mathrm{sol}}$ ranging from $-944.5$ to 80 W m$^{-2}$. The cooling is reduced when the surface becomes more reflective and the contrast between surface and cloud is reduced, which shifts the distributions and their medians towards positive $\Delta F_{\mathrm{sol}}$. With $\alpha_{\mathrm{srf}}$ approaching 0.66, around 25 % of the parameter combinations lead to a solar heating. This becomes even more pronounced towards $\alpha_{\mathrm{srf}} = 1$, where around 50 % of the simulations yield a warming effect in the solar. $\Delta F_{\mathrm{tir}}$ is unaffected by changes in $\alpha_{\mathrm{srf}}$, as expected, and remains constant for all $\alpha_{\mathrm{srf}}$ with a median at 21.5 W m$^{-2}$. The resulting $\Delta F_{\mathrm{net}}$ is dominated by a net warming effect, indicated by mostly positive median values ranging from 1.4 W m$^{-2}$ ($\alpha_{\mathrm{srf}} = 0.25$) to 18.8 W m$^{-2}$ ($\alpha_{\mathrm{srf}} = 1$). An exception is $\alpha_{\mathrm{srf}} = 0$, where more than 50 % of the simulations lead to a net cooling with a median $\Delta F_{\mathrm{net}}$ at $-2.1$ W m$^{-2}$.

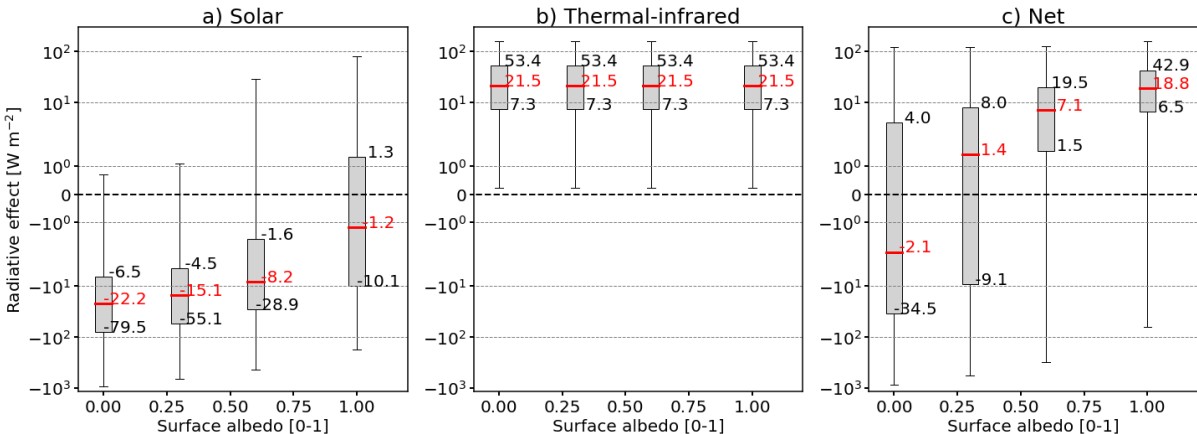

**Figure 7.** Same as Fig. 5 but as a function of the surface albedo $\alpha_{\mathrm{srf}}$.

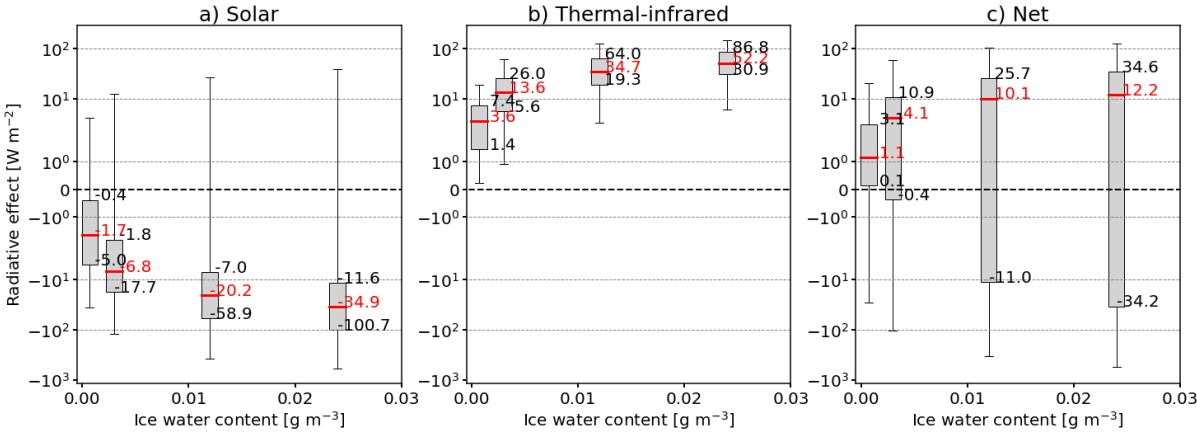

**Figure 8.** Same as Fig. 5 but as a function of ice water content IWC (in $\mathrm{g\,m^{-3}}$). For better legibility only IWC up to $0.03\ \mathrm{g\,m^{-3}}$ are plotted.

### 3.3 Sensitivity on ice water content and ice crystal radius

As presented in Fig. 2, the IWC is the second most influencing factor that controls $\Delta F$. For a constant crystal number concentration the increase in IWC leads to an increase in $r_{\mathrm{eff}}$, as well as the total particle scattering and absorption cross-sections. This enhances scattering and absorption along the optical path $s$ though the cloud. Figure 8a reveals that with increasing IWC the median of $\Delta F_{\mathrm{sol}}$ becomes more negative (intensification of the cooling effect in the solar part of the spectrum). The steepest increase is found for $\mathrm{IWC} < 0.012\ \mathrm{g\,m^{-3}}$, while for $\mathrm{IWC} \geq 0.012\ \mathrm{g\,m^{-3}}$ the solar cloud RE saturates. At the same time $Q_{\Delta F,\mathrm{sol}}$,

given by Eq. 21, increases, indicating an enhanced sensitivity of $\Delta F_{\mathrm{sol}}$ on the free parameters. The minimum and maximum of $\Delta F_{\mathrm{sol}}$ result from clouds over highly reflective surface ($\alpha_{\mathrm{srf}} = 1$) and clouds containing crystals with the smallest $r_{\mathrm{eff}} = 5\ \mu$m.

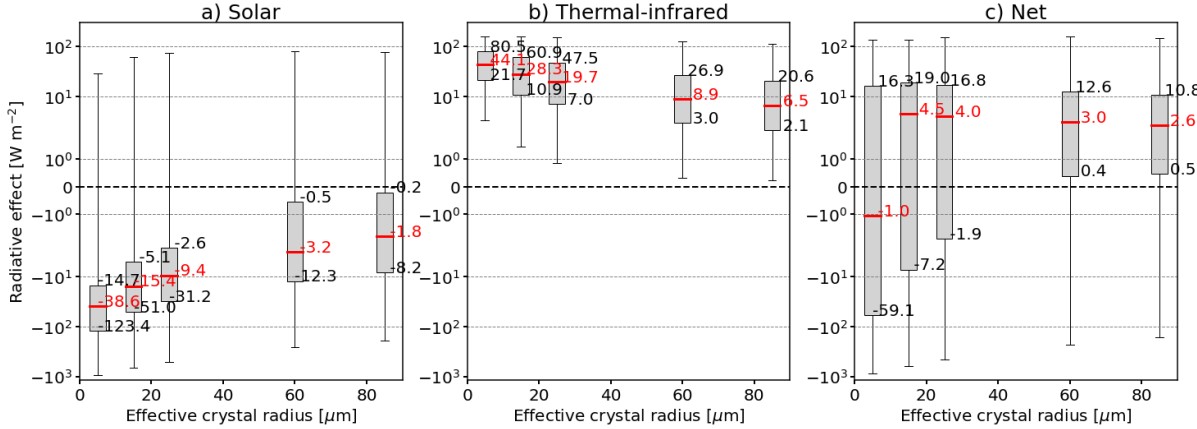

**Figure 9.** Same as Fig. 5 but as a function of effective crystal radius $r_{\mathrm{eff}}$ (in $\mu$m).

For $\Delta F_{\mathrm{tir}}$ the increase in IWC leads to an intensified warming effect (Fig. 8b). Again, this is caused by the increase in the total particle scattering and absorption cross-sections. Similarly to $\Delta F_{\mathrm{sol}}$, the steepest increase in $\Delta F_{\mathrm{tir}}$ appears for IWC $< 0.012$ g m$^{-3}$, while for larger IWC the medians approach an almost constant level and a further increase in IWC has
only a limited effect on $\Delta F_{\mathrm{tir}}$. The resulting $\Delta F_{\mathrm{net}}$ (Fig. 8c) ranges from $-543.2$ to $125.5$ W m$^{-2}$ and is skewed to positive $\Delta F_{\mathrm{net}}$ with median values between 1.1 and 12.2 W m$^{-2}$.

The size of ice crystals also influence the cloud RE, with a larger sensitivity of $\Delta F_{\mathrm{sol}}$ on $r_{\mathrm{eff}}$ than $\Delta F_{\mathrm{tir}}$ (Baum et al., 2005b). Figure 9a illustrates that cirrus with the smallest $r_{\mathrm{eff}}$ are associated with the most intense cooling effect in the solar, leading to $\Delta F_{\mathrm{sol}}$ between $-944.5$ and $80.0$ W m$^{-2}$. Small crystals and high number concentrations lead to higher $\alpha_{\mathrm{cld,ice}}$
in the solar compared to fewer and larger crystals (Stephens et al., 1990; Zhang et al., 1994). For the smallest crystals in the simulations a median $\Delta F_{\mathrm{sol}}$ of $-38.6$ W m$^{-2}$ is determined. For increasing $r_{\mathrm{eff}}$ the cooling effect in the solar range decreases and tends towards $\Delta F_{\mathrm{sol}}$ of $-1.8$ W m$^{-2}$. The intensified solar cooling (more negative $\Delta F_{\mathrm{sol}}$) with decreasing $r_{\mathrm{eff}}$ is associated with an increase in the ice crystal number concentration, while keeping IWC constant, which is also known as cloud albedo effect. In addition, ice crystals with larger $r_{\mathrm{eff}}$ are characterized by enhanced forward scattering. Hence, less
radiation is scattered to the sides or backwards into space. Figure 9a shows that clouds with larger $r_{\mathrm{eff}}$ are less sensitive to the effect of the free parameters as the inter-quartile range decreases strongly from $Q_{\Delta F,\mathrm{sol}}(r_{\mathrm{eff}} = 5\,\mu\mathrm{m}) = 108.7$ W m$^{-2}$ to $Q_{\Delta F,\mathrm{sol}}(r_{\mathrm{eff}} = 85\,\mu\mathrm{m}) = 8.0\,\mathrm{W\,m^{-2}}$. Similarly, Fig. 9b shows the strongest TIR heating for the smallest crystals / highest $N_{\mathrm{ice}}$. Such clouds have the largest total absorption cross-section and act almost as blackbodies in the TIR (Stephens et al., 1990; Zhang et al., 1994). However, an increase in $r_{\mathrm{eff}}$ while fixing IWC leads to a reduction in $\Delta F_{\mathrm{tir}}$, which is caused by the lower
total particle scattering and absorption cross-sections. $Q_{\Delta F,\mathrm{tir}}$ decreases from $58.8$ W m$^{-2}$ for $r_{\mathrm{eff}} = 5\,\mu\mathrm{m}$ to $18.5$ W m$^{-2}$ for $r_{\mathrm{eff}} = 85\,\mu\mathrm{m}$. Median values of $\Delta F_{\mathrm{net}}$, shown in Fig. 9c indicate only a net cooling for $r_{\mathrm{eff}} = 5\,\mu$m with $-1$ W m$^{-2}$, whereby elsewhere a net warming is dominant with $\Delta F_{\mathrm{net}}$ between 2.6 and 4.5 W m$^{-2}$. Simultaneously, $Q_{\Delta F,\mathrm{net}}$ slightly decreases, which indicates the reduced impact of the remaining free parameters for large crystals. The presented dependencies, especially

for small $r_{\mathrm{eff}}$, of solar, TIR, and net $\Delta F$ on $r_{\mathrm{eff}}$ and IWC agree with previous studies, e.g., from Hansen and Travis (1974), but
particularly Fu and Liou (1993) and Zhang et al. (1999).

## 3.4 Multi-dimensional dependencies on $\theta$, $\alpha_{\mathrm{srf}}$, $r_{\mathrm{eff}}$, and IWC

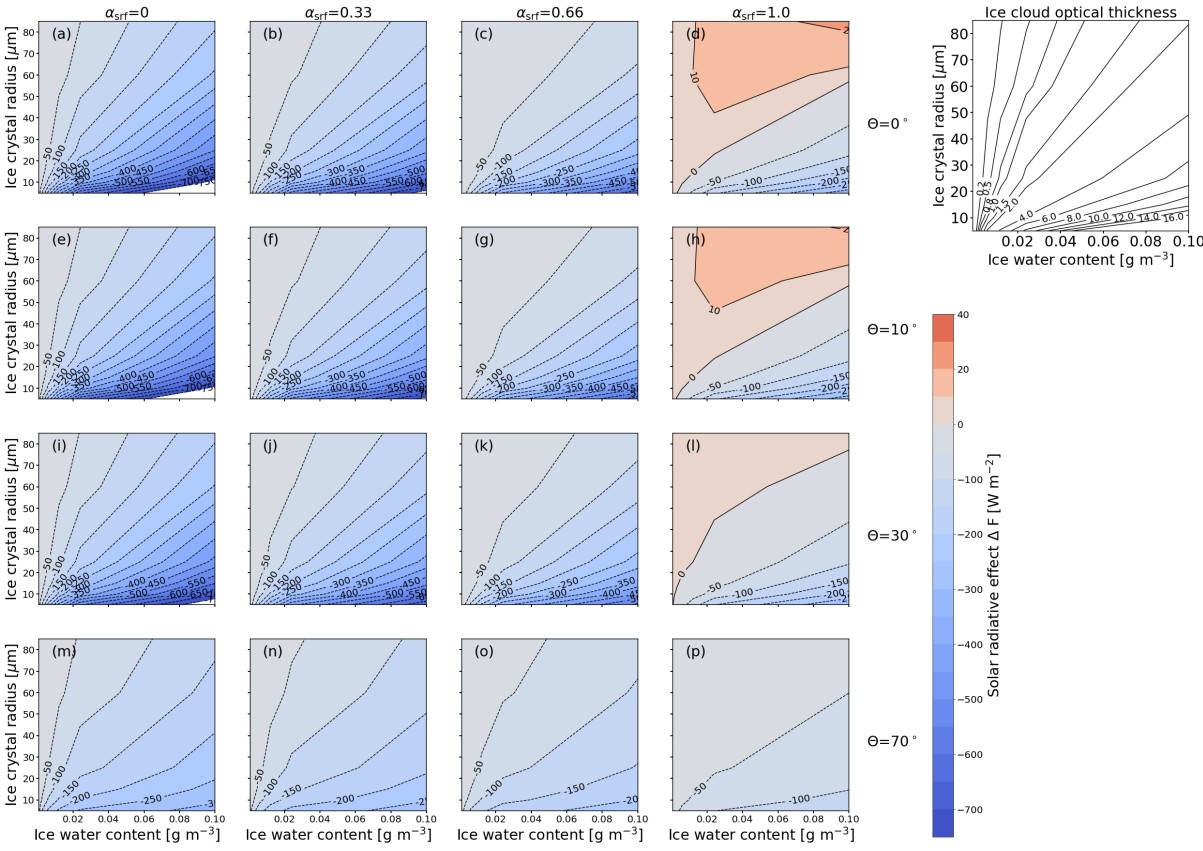

**Figure 10.** Solar cloud radiative effect $\Delta F_{\mathrm{sol}}$ in $\mathrm{W\,m^{-2}}$ sampled into two-dimensional parameter space of ice water content IWC (in $\mathrm{g\,m^{-3}}$) and effective radius $r_{\mathrm{eff}}$ (in $\mu$m). Each panel represents combinations of surface albedo $\alpha_{\mathrm{srf}}$ and solar zenith angle $\theta$. Blue values indicate negative $\Delta F_{\mathrm{sol}}$ (cooling) and red values indicate positive $\Delta F_{\mathrm{sol}}$ (warming). The contour lines provide a direct measure of the sensitivity to the indicated parameters. The top-right panel shows, for reference, the cloud optical depth $\tau$ at 550 nm wavelength that corresponds to the combinations of $r_{\mathrm{eff}}$ and IWC shown on the other panels.

The previous analysis aimed to sample the 8D–hypercube in a series of 1D–cross-sections to focus on the general distribution of $\Delta F$ that result from a single parameter. This likely masks dependencies of $\Delta F$ on specific parameter combinations that are closely interconnected. Subsequently, we focus on a detailed analysis, particularly in the solar wavelength range, to highlight
the dependencies among Sun geometry, surface albedo, and cloud properties - especially $r_{\mathrm{eff}}$ and IWC.

### 3.4.1 Solar radiative effect

Figure 10 shows $\Delta F_{\mathrm{sol}}$ as a function of IWC and $r_{\mathrm{eff}}$ for combinations of $\alpha_{\mathrm{srf}}$ (columns) and $\theta$ (rows). Moving from the left to the right column the surface becomes more reflective (increasing $\alpha_{\mathrm{srf}}$) and going from the top to the bottom row the Sun approaches the horizon (increasing $\theta$).

Figure 10a represents non-reflective surfaces and a Sun at the zenith. In these cases and focusing on ice crystals with $r_{\mathrm{eff}} > 30$ $\mu$m the contour lines are well separated. A wide spacing of the contour lines indicates a low sensitivity of $\Delta F_{\mathrm{sol}}$ on IWC and $r_{\mathrm{eff}}$. In those regions $\Delta F_{\mathrm{sol}}$ ranges from 0 to $-450$ W m$^{-2}$ (cooling), with an intensification of $\Delta F_{\mathrm{sol}}$ for decreasing $r_{\mathrm{eff}}$. The contour lines get closer for $r_{\mathrm{eff}} < 30$ $\mu$m and align with the $x$-axis, which indicates an increase in the sensitivity of $\Delta F_{\mathrm{sol}}$, particularly with respect to $r_{\mathrm{eff}}$, as it is expected from Fig. 2.

For the Sun at zenith and cirrus above reflective surfaces ($0 < \alpha_{\mathrm{srf}} < 1$), the sensitivity with respect to IWC and $r_{\mathrm{eff}}$ is generally reduced. This results from the increasing contribution of surface reflected, upward irradiance, which progressively dominates $\Delta F_{\mathrm{sol}}$ of the cirrus. $\Delta F_{\mathrm{sol}}$ is essentially a measure of the contrast between $\alpha_{\mathrm{srf}}$ and $\alpha_{\mathrm{cld,ice}}$, with $\alpha_{\mathrm{cld,ice}}$ mostly dependent on $r_{\mathrm{eff}}$ and IWC. In case of a highly reflective surface ($\alpha_{\mathrm{srf}} \geq 0.6$; Fig. 10d) the predominant cooling in the solar spectrum turns into a warming effect for most of the combinations with $\Delta F_{\mathrm{sol}}$ up to 15–20 W m$^{-2}$. Only ice clouds with

$r_{\mathrm{eff}} < 20$–30 $\mu$m and IWC $\approx 0.04$–0.1 g m$^{-3}$, i.e., high $\tau_{\mathrm{ice}} > 3$, are more reflective than the surface. Such combinations of $r_{\mathrm{eff}} < 20$ $\mu$m and IWC $\approx 0.04$–0.1 g m$^{-3}$ are associated with ice crystal number concentrations that are rarely observed in nature except for some cases of young contrails (see Fig. 1 in Krämer et al. (2016)).

For cirrus over non-reflective or slightly reflective surfaces ($\alpha_{\mathrm{srf}} \leq 0.33$) and the Sun at intermediate SZA ($\theta \geq 30°$), the contour lines separate and the sensitivity of $\Delta F_{\mathrm{sol}}$ on $r_{\mathrm{eff}}$ and IWC is reduced. However, this effect is less pronounced compared

than a change in $\alpha_{\mathrm{srf}}$. For Sun positions closest to the horizon ($\theta = 70°$) and above highly reflective surfaces ($\alpha_{\mathrm{srf}} = 1$), $\Delta F_{\mathrm{sol}}$ in Fig. 10p is characterized by a generally low sensitivity over the entire range of IWC and $r_{\mathrm{eff}}$. In spite of the warming effect for $\alpha_{\mathrm{srf}} = 1$ and $\theta \leq 30°$, the slant optical path of the incident radiation through the cloud reduces the surface influence and leads to a cooling effect with $\Delta F_{\mathrm{sol}}$ in the range of $-5$ to $-100$ W m$^{-2}$.

### 3.4.2 Thermal-infrared and net radiative effect

The TIR component of $\Delta F$ is insensitive to changes in $\theta$ and $\alpha_{\mathrm{srf}}$, and only combinations of IWC and $r_{\mathrm{eff}}$ are of relevance. In the TIR, the surface is approximated by a blackbody with a wavelength independent emissivity equal to one. The resulting distributions of $\Delta F_{\mathrm{net}}$, shown in Fig. 11, are dominated by the contribution of $\Delta F_{\mathrm{sol}}$ and, therefore, are characterized by similar sensitivities. The strongest gradient of $\Delta F_{\mathrm{net}}$ on IWC and $r_{\mathrm{eff}}$ are found for $\theta \approx 0°$ and $\alpha_{\mathrm{srf}} = 0$ (Fig. 11a). With increasing $\alpha_{\mathrm{srf}}$, $\Delta F_{\mathrm{net}}$ becomes positive for the majority of the combinations of IWC and $r_{\mathrm{eff}}$ (Fig. 11d) with the net warming being most pronounced for $\alpha_{\mathrm{srf}} = 1$, (Fig. 11d). It is further noted that for $\alpha_{\mathrm{srf}} = 1$, $\theta \leq 30°$, and $\tau_{\mathrm{liq}} < 1$, $\Delta F_{\mathrm{net}}$ is positive and almost exclusively sensitive to IWC, while for $\alpha_{\mathrm{srf}} = 1$, $\theta \leq 30°$, and $\tau_{\mathrm{liq}} > 1$, $\Delta F_{\mathrm{net}}$ also becomes sensitive to $r_{\mathrm{eff}}$. In addition, regions that have a net cooling effect, i.e., at high $N_{\mathrm{ice}}$ values, are exclusively sensitive to $r_{\mathrm{eff}}$. The cloud can have a net cooling effect, when the Sun is close to horizon (Fig. 11p), with almost no sensitivity to $r_{\mathrm{eff}}$ and IWC.

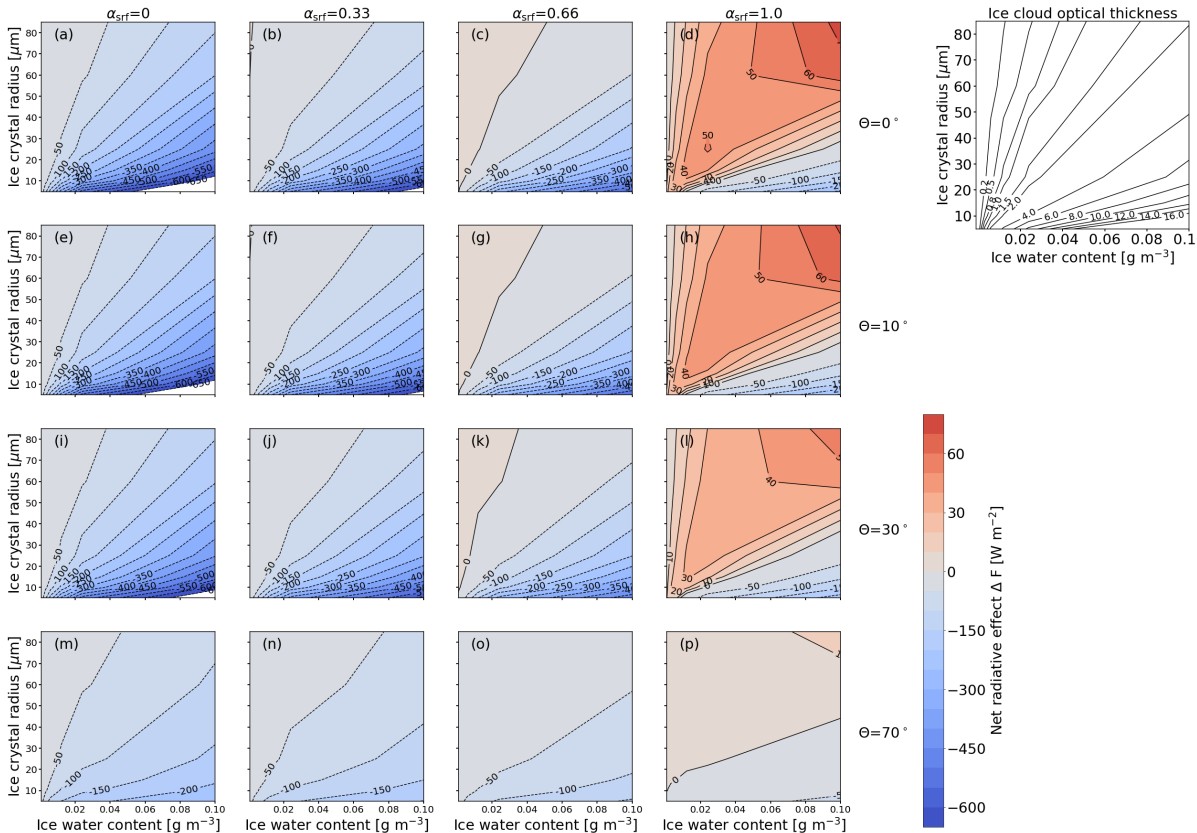

**Figure 11.** Same as Fig. 10 but for $\Delta F_{\mathrm{net}}$ (in $\mathrm{W\,m^{-2}}$).

### 3.5 Sensitivity on atmospheric profile, surface temperature, relative humidity, ice cloud altitude, and ice cloud geometric thickness

Within this study, the atmospheric profiles, the surface temperatures $T_{\mathrm{srf}}$, as well as the vertical location of the ice cloud are coupled. For example, the selection of the US standard atmosphere is directly linked to a surface temperature of $T_{\mathrm{srf}} = 288.2$ K. $T_{\mathrm{srf}}$ is equal to the lowermost temperature value in the respective AP. The vertical position of the ice cloud depends on the temperature of the AP and the selected cloud top temperature $T_{\mathrm{cld,ice}}$ (see Appendix B and Fig. B1a,b therein).

Figure 12a shows that variations in $T_{\mathrm{srf}}$ have an effect on $\Delta F_{\mathrm{sol}}$ with differences in median $\Delta F_{\mathrm{sol}}$ of up to $\pm7$ $\mathrm{W\,m^{-2}}$. Generally larger effects are found for the TIR component, where an increase in $T_{\mathrm{srf}}$ enhances the temperature difference between surface and cirrus, which leads to an intensification of the TIR heating (see Eq. 19 and Corti and Peter (2009)), shifting the median $\Delta F_{\mathrm{tir}}$ from 3.6 to 13.6 $\mathrm{W\,m^{-2}}$ (Fig. 12b). Simultaneously, the distributions broaden with increasing $T_{\mathrm{srf}}$ with $Q_{\Delta F, \mathrm{tir}}$ ranging from 1.4 to 7.4 $\mathrm{W\,m^{-2}}$ (257.2 K) and 5.6 to 26.0 $\mathrm{W\,m^{-2}}$ (299.7 K), which results from the warmer and moister tropical profile compared to the drier Subarctic profile. As a result of the almost constant $\Delta F_{\mathrm{sol}}$ and the increase in $\Delta F_{\mathrm{tir}}$, the net heating effect is enhanced with medians ranging between 0.7 and 6.6 $\mathrm{W\,m^{-2}}$.

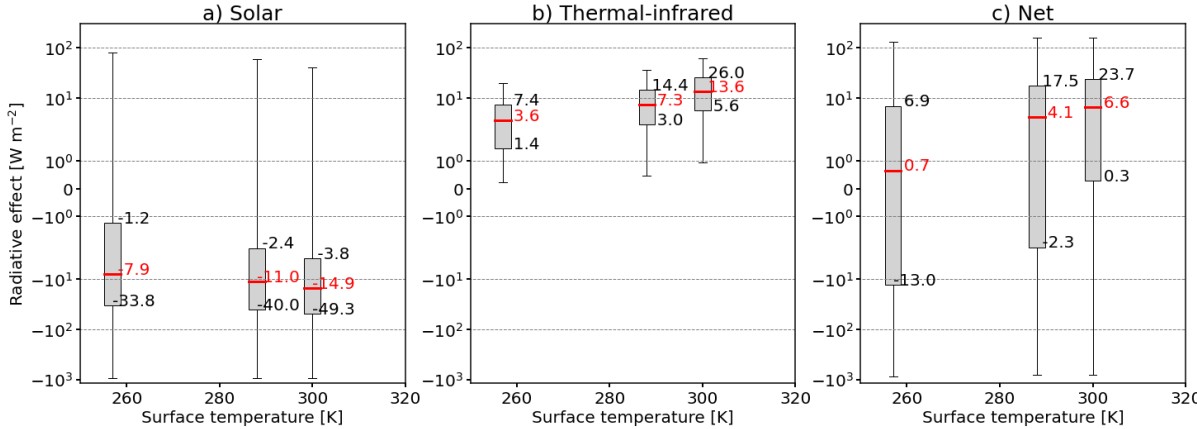

**Figure 12.** Same as Fig. 5 but for surface temperature $T_{\mathrm{srf}}$ (in K).

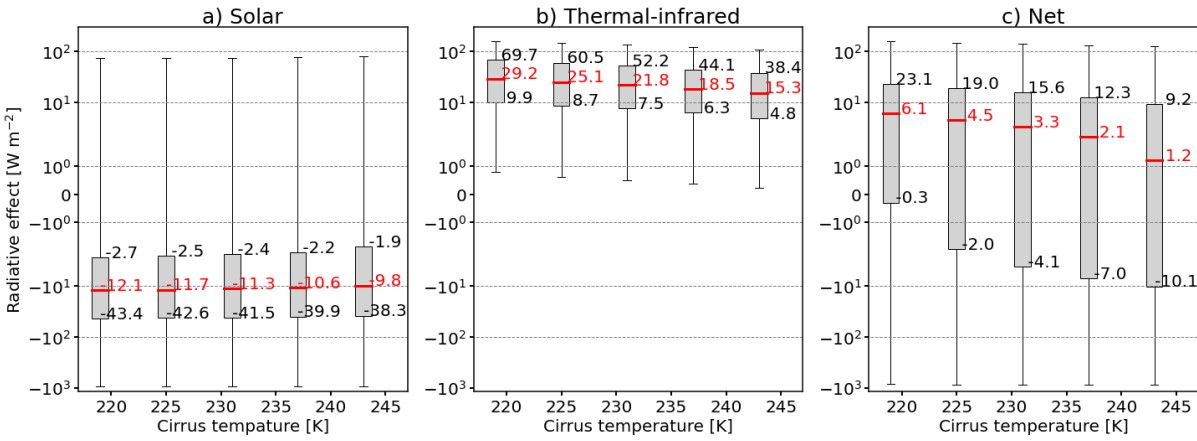

**Figure 13.** Same as Fig. 5 but for ice cloud temperature $T_{\mathrm{cld,ice}}$ (in K).

The effect of variations in $T_{\mathrm{cld,ice}}$ are shown in Fig. 13a–c. Increasing $T_{\mathrm{cld,ice}}$ reduces the temperature difference between surface and ice cloud, and therefore the TIR heating effect (Fig. 13b). Median $\Delta F_{\mathrm{tir}}$ are reduced from 29.2 to 15.3 $\mathrm{W\,m^{-2}}$, when $T_{\mathrm{cld,ice}}$ is increased from 219 to 243 K. Compared to the impact of $T_{\mathrm{srf}}$, which was varied over a range of 42.5 K, shifting the cloud in the vertical has only a minor effect on $\Delta F_{\mathrm{tir}}$ and $\Delta F_{\mathrm{net}}$, as the variation in $T_{\mathrm{cld,ice}}$ spanned only 24 K. The resulting net effect from variations in $T_{\mathrm{cld,ice}}$ leads to medians between 1.2 and 6.1 $\mathrm{W\,m^{-2}}$ for $T_{\mathrm{cld,ice}}$ of 219 and 243 K, respectively.

The previously mentioned impact of $T_{\mathrm{srf}}$ on $\Delta F_{\mathrm{sol}}$ is traced back to: a) the different optical path length through the atmosphere because of variations in cloud top altitude; and b) the different water vapor concentration due to the three applied APs. The effect of varying RH profiles were investigated by manipulating the original RH profiles by $\pm 20\,\%$ representing the variability in RH reported by Anderson et al. (1986). The RT simulations were performed for a sub-set of the parameter space

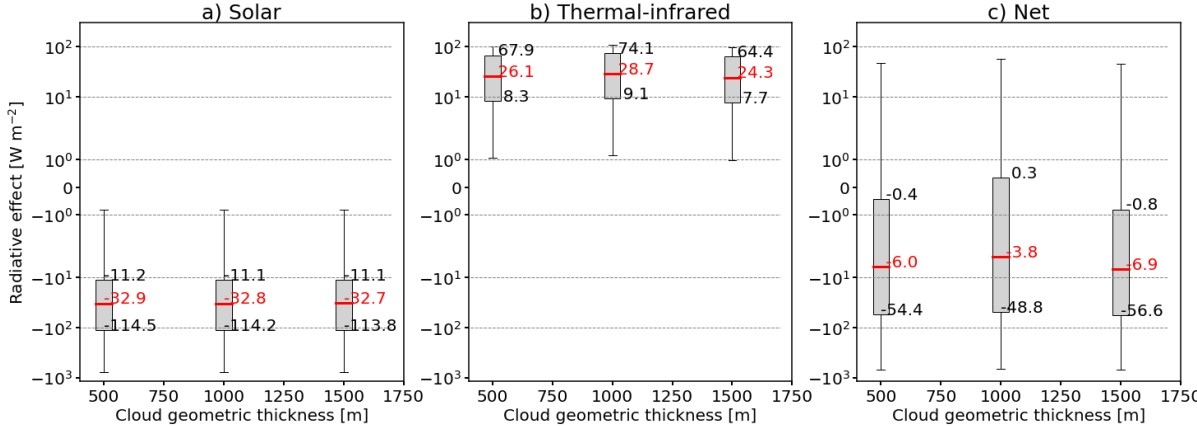

**Figure 14.** Same as Fig. 5 but for the cloud geometrical thickness d$z$ (in m) and only for a sub sample of the parameter space. Values for ice cloud temperature $T_{ice}$ = 231 K, surface temperature $T_{srf}$ = 288.2 K, surface albedo $\alpha_{srf}$ = 0.15, and liquid water cloud optical thickness $\tau_{liq}$ = 0. Values for solar zenith angle $\theta$, ice water content IWC, and effective radius $r_{eff}$ are varied.

with fixed $T_{cld,ice}$ = 231 K, $\alpha_{srf}$ = 0, and $\tau_{liq}$ = 0. The effects on solar, TIR, and net $\Delta F$ are quantified by their absolute and relative differences. Variations in RH have only a small effect on $\Delta F_{sol}$ with maximal $\pm 0.15$ W m$^{-2}$ ($\pm 0.4\,\%$) among all profiles. A slightly larger impact is found for $\Delta F_{tir}$ with up to $\pm 1.45$ W m$^{-2}$ ($\pm 4.1\,\%$) in case of the warm and moist tropical

profile (`afglt`). Less affected are the standard atmosphere (`afglus`), where $\Delta F_{tir}$ varies by $\pm 0.9$ W m$^{-2}$ ($\pm 3.2\,\%$) and the dry Subarctic profile (`afglsw`) with variations in $\Delta F_{tir}$ of $\pm 0.3$ W m$^{-2}$ ($\pm 2.4\,\%$). Consequently, `afglt` has the largest variation in $\Delta F_{net}$ of $\pm 0.8$ W m$^{-2}$ ($\pm 8\,\%$) and is followed by $\pm 0.6$ W m$^{-2}$ ($\pm 3.8\,\%$) for `afglus` and $\pm 0.2$ W m$^{-2}$ ($\pm 0.6\,\%$) for `afglsw`. Scaling the original RH profiles showed that variations on the RH profile explicitly influence the TIR wavelength range but particularly the net RE. This analysis suggests that the variations in RH have to be considered as potential source of

variability, when using this publicly available data set.

All simulations within this study were performed for a fixed cloud geometric thickness d$z$ of 1000 m. In reality however, d$z$ is likely to vary over the cirrus lifetime, for example due to sedimentation of ice crystals or vertical winds. The effect of changing d$z$ is quantified by a dedicated sensitivity analysis of $\Delta F$ for a sub-sample of the full parameter range (Table 4). A similar sub-parameter space is used as for the RH sensitivity but additionally fixing $T_{srf}$ = 288 K, i.e., using the `afglus`

profile. With $\tau_{ice}$ being proportional to the IWP of the cloud (Eq. 10), the IWP of the 1000 m reference and solar $\tau_{ice}$ are kept constant, and the IWC for the clouds with d$z$ of 500 and 1500 m clouds is scaled accordingly.

As expected from Eq. 10, the resulting effect on median $\Delta F_{sol}$, given in Fig. 14, is almost negligible with $\pm 0.1$ W m$^{-2}$ ($\pm 0.3\,\%$). Differences in median $\Delta F_{tir}$ are up to $\pm 0.6$ W m$^{-2}$ ($\pm 3.5\,\%$), which leads to differences in median $\Delta F_{net}$ of $\pm 0.6$ W m$^{-2}$ ($\pm 6.2\,\%$). The relevant relative differences in $\Delta F_{tir}$ and $\Delta F_{net}$ are explained by the varying cloud base altitude,

which modifies the vertical distribution of IWC and the temperature of the cloud base, which determines the amount of emitted radiation. In addition, geometrically thin clouds with low $\tau_{ice}$ act as gray bodies, while with an increase in d$z$ cirrus become

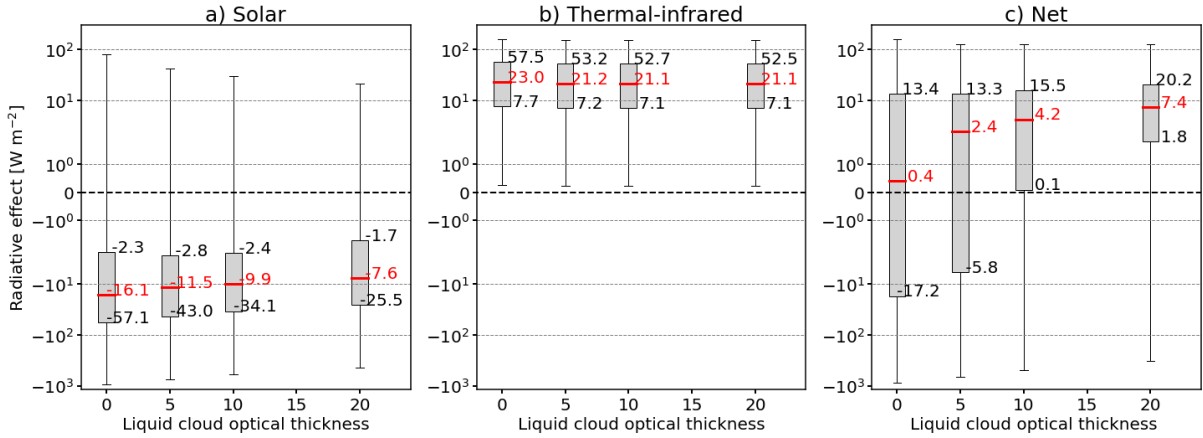

**Figure 15.** Same as Fig. 5 but for the underlying liquid water cloud optical thickness $\tau_{\mathrm{liq}}$.

opaque and act as more efficient black bodies (Corti and Peter, 2009). Fu and Liou (1993) further reported that cirrus with small $r_{\mathrm{eff}}$ reflect solar radiation at the cloud top (solar cooling) but absorb TIR radiation at the cloud base (TIR warming), which creates a temperature gradient within the cloud that depends on d$z$. From the d$z$ sensitivity analysis it is found that d$z$

can be neglected in the solar wavelength range but is of relevance for $\Delta F_{\mathrm{tir}}$ and especially $\Delta F_{\mathrm{net}}$, where absolute values are small. This partly agrees with the findings from Meerkötter et al. (1999) who showed that solar, TIR, and net $\Delta F$ are only slightly sensitive to changes in d$z$ with solar, TIR, and net $\Delta F$ below 2 W m$^{-2}$, under the premise of a constant ice water path (IWP). The presented simulations indicate $\Delta\Delta F_{\mathrm{sol}}$ of 2 W m$^{-2}$, which is comparable to Meerkötter et al. (1999), but we found slightly higher $\Delta\Delta F_{\mathrm{tir}}$ and $\Delta\Delta F_{\mathrm{net}}$ of 4.5 and 3.1 W m$^{-2}$, respectively.

**3.6  Sensitivity on underlying liquid water cloud**

The impact of an additional liquid water cloud on the cirrus $\Delta F$ is presented in Fig. 15. A liquid water cloud optical thickness $\tau_{\mathrm{liq}} = 0$ is equivalent to the absence of secondary clouds and such conditions lead to the strongest $\Delta F_{\mathrm{sol}}$ with a median of $-16.1$ W m$^{-2}$. By gradually increasing $\tau_{\mathrm{liq}}$ the reflected, upward irradiance overlays and masks the impact of the surface. In general, the response of $\Delta F_{\mathrm{sol}}$ on $\tau_{\mathrm{liq}}$ is comparable to that of an increase in $\alpha_{\mathrm{srf}}$. Introducing a cloud with $\tau_{\mathrm{liq}} = 5$ slightly

enhances the cooling in the solar spectrum $\Delta F_{\mathrm{sol}}$ from $-11.5$ to $-7.6$ W m$^{-2}$. More notable is the reduction in the variability of $\Delta F_{\mathrm{sol}}$ with the distribution becoming narrower and reducing $Q_{\Delta\mathrm{F,sol}}$ from 54.8 W m$^{-2}$ to 23.8 W m$^{-2}$.

An increase in $\tau_{\mathrm{liq}}$ from 0 to 5 shifts the median $\Delta F_{\mathrm{tir}}$ from 21.1 to 23.0 W m$^{-2}$. With a further increase in $\tau_{\mathrm{liq}}$ the medians remain almost constant, while the $Q_{\Delta\mathrm{F,tir}}$ slightly decreases. The reduction of maximum $\Delta F_{\mathrm{tir}}$ is a consequence of the attenuated temperature difference $\Delta T$ between the liquid water cloud and the ice cloud compared to the surface. The effect

on $\Delta F_{\mathrm{tir}}$ is small as the change in temperature from surface to liquid water cloud is small. In the case of the US standard atmosphere, where $\Delta T = 5$ K.

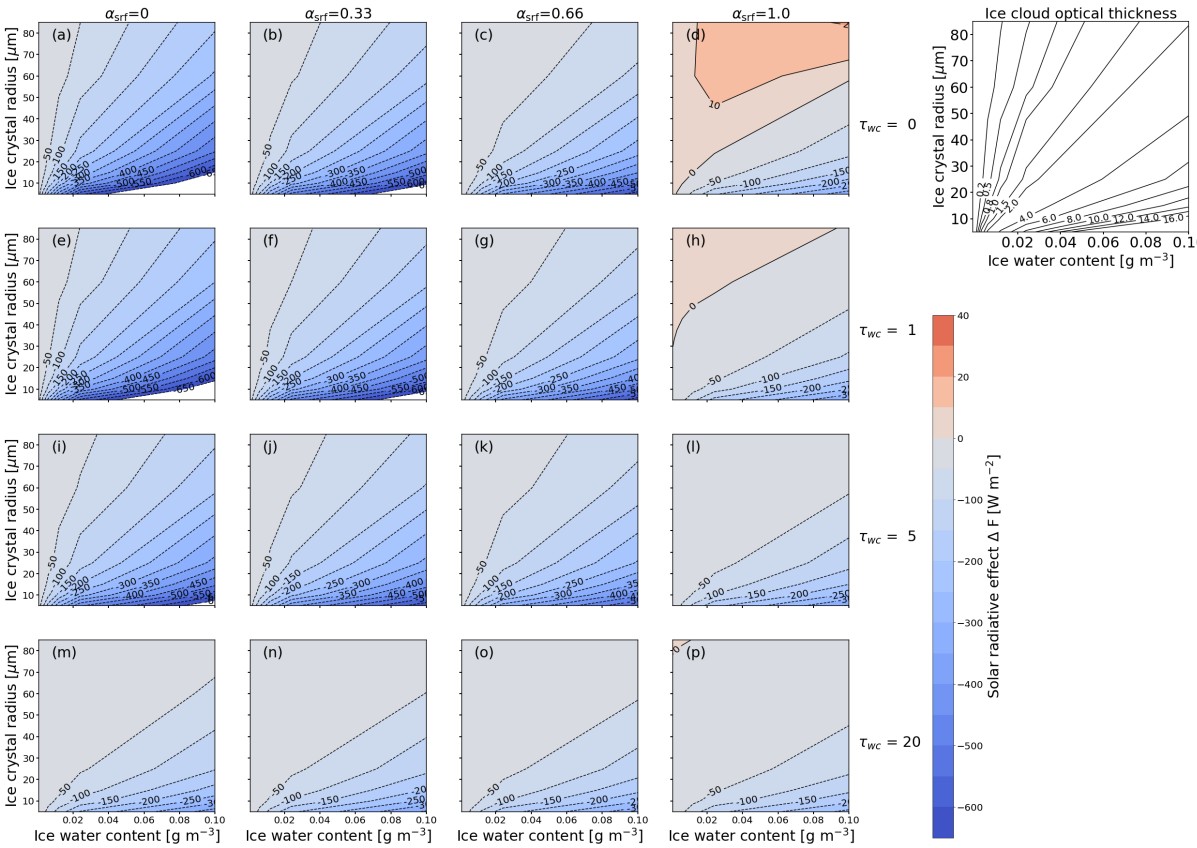

**Figure 16.** Same as Fig. 10 but $\Delta F_{\mathrm{sol}}$ (in $\mathrm{W\,m^{-2}}$) and combinations of surface albedo $\alpha_{\mathrm{srf}}$ and cloud optical thickness $\tau_{\mathrm{liq}}$ of the underlying liquid water cloud.

As a result of the reduced cooling in the solar spectrum and the stronger warming in the TIR spectrum, the net heating of the ice clouds intensifies with increasing $\tau_{\mathrm{liq}}$. The median $\Delta F_{\mathrm{net}}$ is shifted from 0.4 to 7.4 $\mathrm{W\,m^{-2}}$ with an accompanying decrease in the overall variance. While for $\tau_{\mathrm{liq}} < 5$ slightly fewer than 50 % of the combinations exert a potential net cooling by the

cirrus, positive $\Delta F_{\mathrm{net}}$ is dominating for larger $\tau_{\mathrm{liq}}$.

Figure 16 shows $\Delta F_{\mathrm{sol}}$ depending on IWC and $r_{\mathrm{eff}}$ separated for $\alpha_{\mathrm{srf}}$ (columns) and $\tau_{\mathrm{liq}}$ (rows). In the presented cases, a $\theta$ of $10°$ is selected as the influence of the surface and an additional cloud layer is of higher importance, when the Sun is close to the zenith. Due to the selection of $\theta$, the top row in Fig. 16 is the same as the second row in Fig. 10 with similar characteristic features in distribution and sensitivity: largest RE appears over dark surfaces ($\alpha_{\mathrm{srf}} = 0$) in combination with clouds containing

the largest ice number concentrations $N_{\mathrm{ice}}$ due to small $r_{\mathrm{eff}}$ and larger IWC. Increasing $r_{\mathrm{eff}}$ and / or reducing the IWC weakens $\Delta F_{\mathrm{sol}}$. Introducing the second cloud layer and gradually increasing $\tau_{\mathrm{liq}}$ generally reduces the sensitivity on the ice cloud microphysical properties and the ice cloud RE. For the special case of $\alpha_{\mathrm{srf}} = 1$, the introduction of a liquid water cloud turns

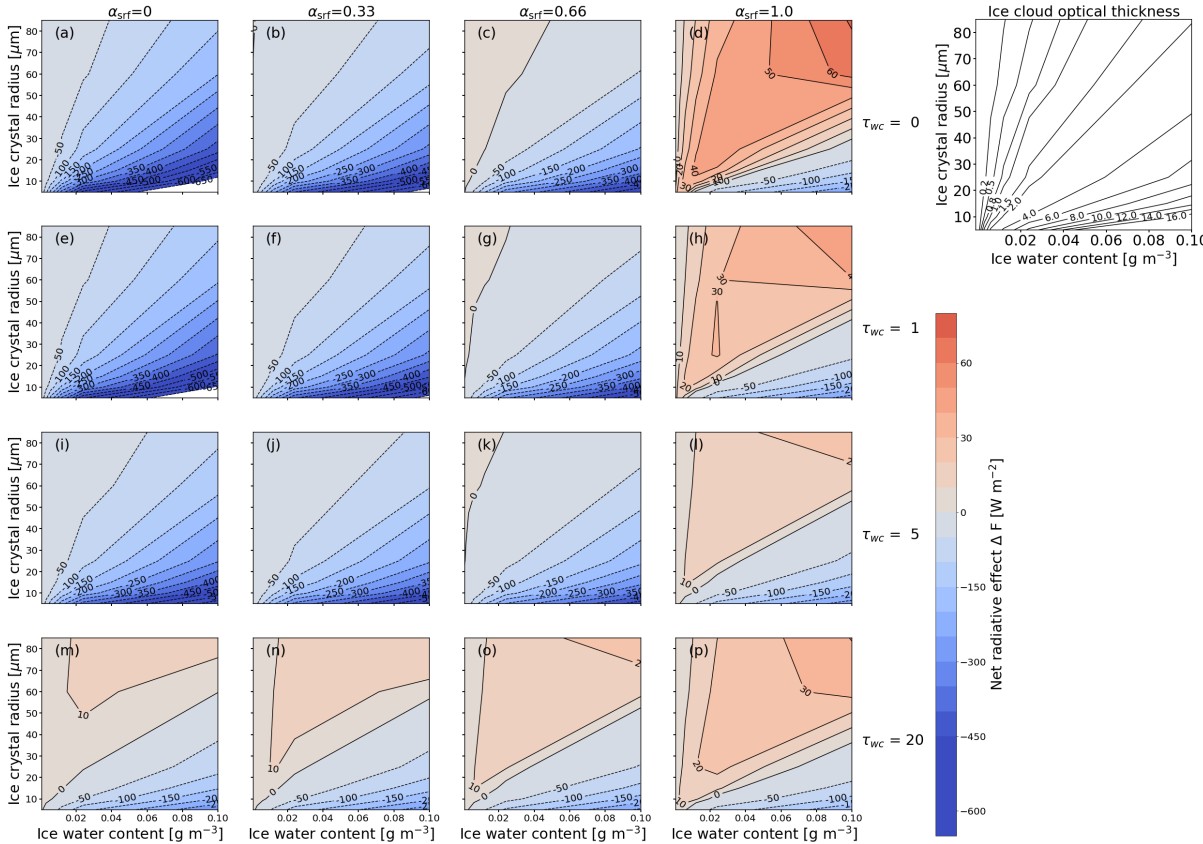

**Figure 17.** Same as Fig. 11 but for $\Delta F_{\text{net}}$ (in $\text{W m}^{-2}$) and combinations of surface albedo $\alpha_{\text{srf}}$ and cloud optical thickness $\tau_{\text{liq}}$ of the underlying liquid water cloud.

the previous solar warming ($\Delta F_{\text{sol}} \approx 10 \ \text{W m}^{-2}$) into a solar cooling effect of up to $\Delta F_{\text{sol}} = -15 \ \text{W m}^{-2}$ for typical $\tau_{\text{ice}}$ of contrails.

As shown in Fig. 15 the second cloud layer, at $z = 1500 \ \text{m}$ modifies $\Delta F_{\text{tir}}$ only slightly and multi-dimensional dependencies with respect to IWC, $r_{\text{eff}}$, $\alpha_{\text{srf}}$, and $\tau_{\text{liq}}$ are weak leading to homogeneous distributions (not shown here). Figure 17 illustrates the variations in $\Delta F_{\text{net}}$. For combinations of $\alpha_{\text{srf}} \leq 0.66$ and $\tau_{\text{ice}} \leq 5$, $\Delta F_{\text{net}}$ is determined by the solar component and its sensitivities. Special attention should be given to conditions with $\alpha_{\text{srf}} \geq 0.66$ and $\tau_{\text{ice}} > 0.8$, where $\Delta F_{\text{net}}$ turns from a cooling into a warming effect. This is due to the reduced $\Delta F_{\text{sol}}$ and the domination by $\Delta F_{\text{tir}}$. In these situations $\Delta F_{\text{net}}$ ranges between

0 and 60 $\text{W m}^{-2}$. Increasing $\tau_{\text{liq}}$ leads to a reduction in the sensitivity of $\Delta F_{\text{net}}$ with respect to $r_{\text{eff}}$ and IWC of the ice cloud. An exception are clouds with extreme $N_{\text{ice}}$, where an increased cooling effect in relation to $r_{\text{eff}}$ occurs. Furthermore, cirrus over optically thick underlying clouds (last row) has a predominantly net warming effect.

## 4 Discussion

This study focused on the cloud RE of homogeneous, horizontally infinite ice cloud layers and neglected horizontal photon
transport. The vertical and horizontal structure of ice clouds, i.e., distribution of ice water content, is typically heterogeneous,
which is one reason for differences and uncertainties between 1D-simulated and the actual RE of such clouds (Fauchez et al.,
2017, 2018). Additional differences originate from the independent pixel approximation (Cahalan et al., 1994).

For completeness and to raise awareness of potential uncertainties in the present simulations due to the effects of cloud
heterogeneity and 3D-scattering on the estimated RE, we provide a brief overview of the relevant literature. The majority
of past cirrus and contrail studies that quantified the RE sensitivity were based on one-dimensional (1D) RT simulations
(Strauss et al., 1997; Meerkötter et al., 1999; Fahey et al., 1999; Stuber et al., 2006). While aged and spread contrails might
be approximated as thin plane-parallel layers within a homogeneous atmosphere (Minnis et al., 1999), younger contrails and
cirrus are heterogeneous in their horizontal and vertical distribution of IWC. The first study that investigated 3D-radiative
effects was performed by Schulz (1998). This study was followed by Gounou and Hogan (2007) and Forster et al. (2012), who
used 3D Monte Carlo simulations and found differences in contrail solar RE between 1D and 3D simulations ranging from 5
to 40 %. The largest deviations were found for extreme cases, e.g., large solar zenith angle (Sun close to the horizon). With
the Sun illuminating the contrail or cirrus from the side, extinction and absorption within the cloud increases and scattering at
cloud sides becomes more important compared to an illumination from above. Enhanced scattering at cloud sides also increases
the likelihood that photons get scattered back into space instead of being absorbed. Such effects are not captured by 1D RT
simulations. Concerning the TIR wavelength range, Gounou and Hogan (2007) found that horizontal photons transport can
increase contrail radiative effect by around 10 %, which has to be considered in the calculation of the contrail net radiative
effect.

However, there is no systematic bias in solar, TIR, and net RE between 1D and 3D simulations and the deviations decrease
with increasing cloud homogeneity. More specifically, the differences between 1D and 3D simulations changes in magnitude
and sign depending on the cloud heterogeneity and the solar illumination geometry. We employ 1D simulations as the to-
tal number of simulations performed within this study and the computational cost for full 3D RT simulation is unpractical.
Therefore, we highlight that the provided data set can be used for situations that can be approximated by plane-parallel clouds
and solar zenith angles smaller than 70°. Results should be used carefully by considering that 3D radiative effects introduce
uncertainties.

The eight selected parameters discussed in this study were found to be the most influential on the cirrus RE. The selection is
further supported by earlier studies, e.g., Fu and Liou (1993), Zhang et al. (1999), Meerkötter et al. (1999), Yang et al. (2010),
or Mitchell et al. (2011). However, not all potential factors that impact the cirrus RE can be considered in such a parametric
study. Additional influences like aerosol layers, more complex surface albedo, or multiple overlapping cirrus and contrails
have not been investigated here and represent additional degrees of freedom. For example, previous studies found that aerosols
have only a minor influence on contrail RE (Meerkötter et al., 1999) and Sanz-Morère et al. (2021) reported that the impact of
overlap between contrails on their RE is negligible.

## 5 Summary

The net radiative effect $\Delta F$ (RE) of cirrus and contrails depends on multiple factors related to the microphysical and macrophysical cloud properties, the cloud optical properties, and radiative properties of the environment. The presented study aimed
to separate the effect of eight selected parameters: solar zenith angle $\theta$, ice water content IWC, ice crystal effective radius $r_{\text{eff}}$, cirrus temperature $T_{\text{cld,ice}}$, surface albedo $\alpha_{\text{srf}}$, surface temperature $T_{\text{srf}}$, liquid water cloud optical thickness $\tau_{\text{liq}}$ of an underlying cloud, and three ice crystal shapes on the cirrus RE. In total, 283,500 radiative transfer (RT) simulations have been performed with the libRadtran RT code by varying the 8 parameters within the ranges that are typically associated with natural cirrus and contrails. The RT simulations were performed with a 1D solver (plane-parallel clouds) and 3D scattering effects
were not considered despite the fact they are known to become relevant for large solar zenith angles ($\theta > 70°$.) Specific cases or sub-samples were selected and discussed, while the entire set of results is made available as a NetCDF file (Wolf et al., 2023).

For the presented cases the cirrus RE was discussed separately for the solar $\Delta F_{\text{sol}}$ and thermal-infrared (TIR) $\Delta F_{\text{tir}}$ part of the spectrum, but also for the combined net RE. Comparing to a chosen reference with $\theta = 0°$, $T_{\text{cld,ice}} = 219$ K, $\alpha_{\text{srf}} = 0$,
$T_{\text{srf}} = 299.7$ K, IWC $= 0.024$ g m$^{-3}$, $r_{\text{eff}} = 85$ $\mu$m, $\tau_{\text{liq}} = 0$ (no liquid water cloud), and resulting $\tau_{\text{ice}} = 0.46$ (at 550 nm) it was found that $r_{\text{eff}}$ has the largest impact on solar, TIR, and net RE. The second most important parameter is the IWC, which impacts $\Delta F_{\text{sol}}$ and $\Delta F_{\text{tir}}$ equally. In the selected case, $\Delta F_{\text{sol}}$ and $\Delta F_{\text{tir}}$ have opposite signs, meaning that the IWC has a relatively small impact on $\Delta F_{\text{net}}$. It has to be noted that the counter-balancing effect only appears during daytime, when $\Delta F_{\text{sol}} \neq 0$ W m$^{-2}$. Whether $r_{\text{eff}}$ or IWC is the most impactful parameter depends on the $r_{\text{eff}}$ chosen as a reference. However,
the dominance of $r_{\text{eff}}$ and IWC over all other parameters remains. At night, $\Delta F_{\text{net}}$ equals $\Delta F_{\text{tir}}$ and the cirrus heats the Earth-atmosphere-system. After $r_{\text{eff}}$ and IWC, the solar RE of cirrus is determined by $\theta$, $\alpha_{\text{srf}}$, $\tau_{\text{liq}}$, and the ice crystal shape in descending priority. The RE in the TIR spectrum is dominated by $T_{\text{srf}}$, $T_{\text{cld,ice}}$, $\tau_{\text{liq}}$, and the ice crystal shape. The combined net RE is controlled by $\alpha_{\text{srf}}$, $\theta$, and $T_{\text{srf}}$, sorted in decreasing importance. The relevance of selected parameters can differ for other $\tau_{\text{ice}}$ and ambient condition.

The impact of individual parameters on the solar, TIR, and net RE was further investigated and quantified by sub-sampling the entire set of simulations by fixing one parameter at a time, while the remaining parameters were allowed to vary. This can be interpreted as a type of a sub-sampling, by averaging all unfixed values of RE, to project $\Delta F$ onto the one-dimensional space.

- Variations in $\theta$ have no influence on $\Delta F_{\text{tir}}$ but only on $\Delta F_{\text{sol}}$. The majority of simulated $\Delta F_{\text{sol}}$ becomes more intense (stronger cooling) with increasing $\theta$ and reaches a maximum for $\theta$ between $50°$–$70°$. For further increasing $\theta$ the cooling effect in the solar declines. The exact location of maximum $\Delta F_{\text{sol}}$ is primarily dependent on $\alpha_{\text{srf}}$. Increasing $\theta$, the impact of the other free parameters and the resulting $\Delta F_{\text{sol}}$ are reduced. Consequently, the majority of the simulations with negative $\Delta F_{\text{sol}}$ are exceeded by positive $\Delta F_{\text{tir}}$, which leads to a positive median $\Delta F_{\text{net}}$ (warming).

- The projection of $\Delta F_{\text{net}}$ for varying $\alpha_{\text{srf}}$ showed that cirrus primarily cools in the solar, except for highly reflective surfaces with $\alpha_{\text{srf}}$ approaching 1, e.g., over ice covered regions. Contrarily, $\Delta F_{\text{tir}}$ is mostly positive and unaffected by

the variations in $\alpha_{\mathrm{srf}}$. $\Delta F_{\mathrm{tir}}$ determines the resulting $\Delta F_{\mathrm{net}}$, which leads to a net heating effect, when $\alpha_{\mathrm{srf}}$ exceeds the critical range of 0.25–0.3.

- An increase in IWC intensifies the cooling in the solar and the heating in the TIR. As both effects compete against each other and $\Delta F_{\mathrm{tir}}$ dominates $\Delta F_{\mathrm{sol}}$, the resulting net RE is a warming. An exception appears for largest IWC, where the median $\Delta F_{\mathrm{net}}$ is negative. Simultaneously, the increase in IWC causes an enhanced impact of the free parameters and associated uncertainties.

- Clouds with similar IWC but larger $r_{\mathrm{eff}}$ are comprised of fewer ice crystals, which reduces the cloud reflectivity (cloud albedo effect). Over the entire range of $r_{\mathrm{eff}}$ the sub-sampled data set is characterized by a negative $\Delta F_{\mathrm{sol}}$ that is most intense for the smallest crystals. Similarly, $\Delta F_{\mathrm{tir}}$ is largest for small crystals and decreases for large crystals. While the solar and TIR $\Delta F$ become less intense with $r_{\mathrm{eff}}$, the decrease is more pronounced for $\Delta F_{\mathrm{sol}}$ such that cirrus primarily has a positive $\Delta F_{\mathrm{net}}$. An exception are clouds with the smallest $r_{\mathrm{eff}}$ and high IWC that occur only in contrails that just formed over non-reflective surfaces.

- The surface temperature $T_{\mathrm{srf}}$ and ice cloud temperature $T_{\mathrm{cld,ice}}$ only affect the TIR component of $\Delta F$. Increasing the absolute difference between $T_{\mathrm{srf}}$ and $T_{\mathrm{cld,ice}}$ leads to an intensified TIR and resulting net heating effect.

- An underlying liquid water cloud with an increasing $\tau_{\mathrm{liq}}$ leads to a reduction in solar $\Delta F_{\mathrm{sol}}$. Simultaneously, the TIR heating remains almost constant, reducing the negative $\Delta F_{\mathrm{net}}$ (cooling) that is finally turned into positive $\Delta F_{\mathrm{net}}$ (net warming) for the majority of simulated cases.

*Data availability.* The three data-sets with all simulated irradiances, the calculated cloud radiative effect, and the ice cloud optical thickness are given in separate NetCDF-files. Each file represents an individual ice crystal shape. The data is available on the zenodo platform as Wolf et al. (2023)

## Appendix A: Overview over the multi-parameter dependencies

Figures A1 and A2 show solar $\Delta F_{\mathrm{sol}}$ and TIR $\Delta F_{\mathrm{tir}}$ (above diagonal), and net $\Delta F_{\mathrm{net}}$ (below diagonal) for combinations of parameters indicated along the $x$- and $y$-axis. Both plots are intended to provide an overview over the multi-parameter dependencies. Within each sub-panel $\Delta F$ is given as a function of the $x$- and $y$-axis, while the other parameters are set to constant values that are representative of contrails and cirrus clouds. For example, the 'IWC–SZA' panel shows $\Delta F$ as a function of IWC, with $\theta = 30°$, $T_{\mathrm{cld,ice}} = 231$ K, $r_{\mathrm{eff}} = 25$ $\mu$m, $\alpha = 0.15$, $T_{\mathrm{srf}} = 288$ K, and without a second liquid water cloud ($\tau_{\mathrm{liq}} = 0$). This can be understood as a 2D–cross-section of the 8D–hypercube. The black arrows indicate the gradient of the field. The gradient is computed with second order central differences and one-side differences at the boundaries of the field. The length of the arrow is only representative for an individual field and cannot be compared with the other fields as it

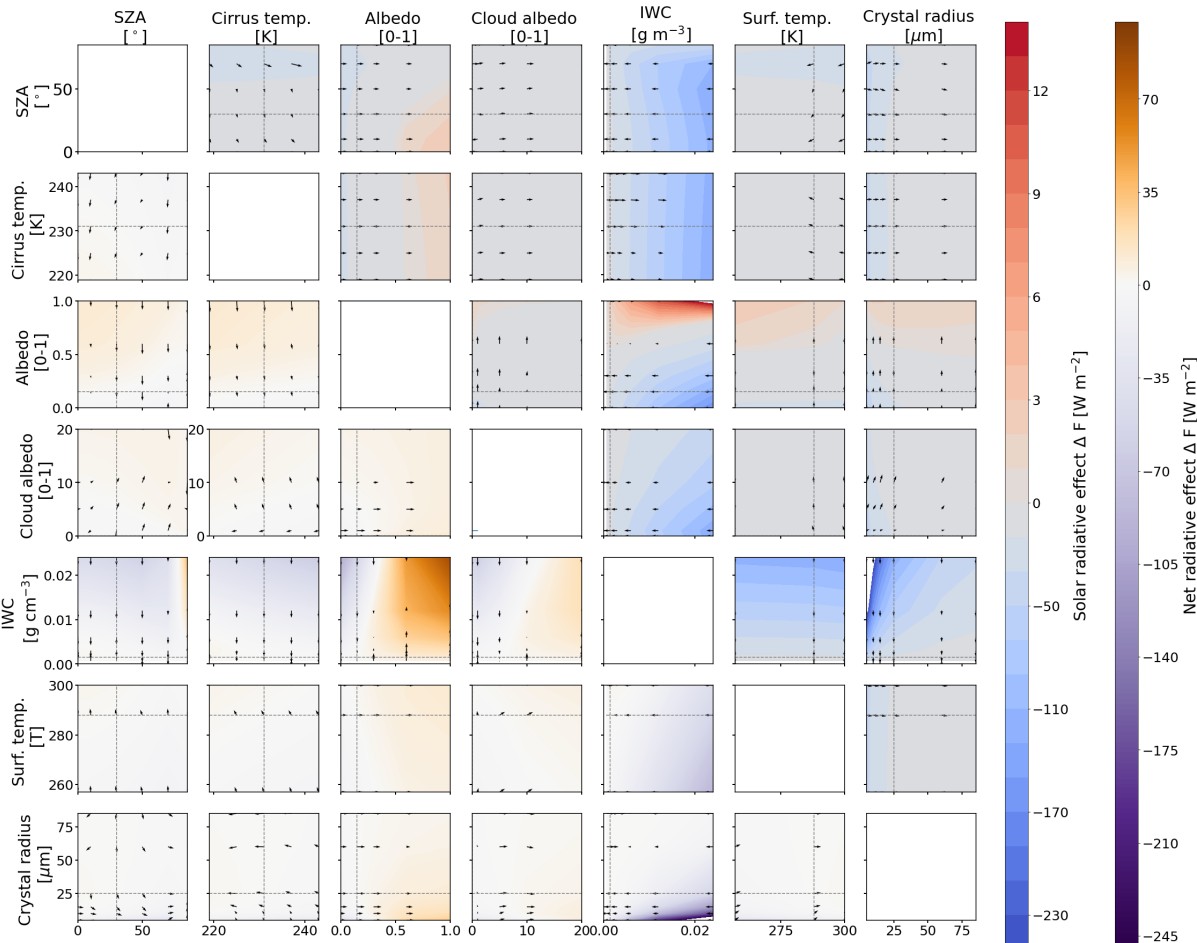

**Figure A1.** Above diagonal panels: Plot of median solar $\Delta F_{\mathrm{sol}}$ projected in two-dimensional parameter space. Blue colors indicate negative $\Delta F_{\mathrm{sol}}$ (cooling), while red colors indicate positive $\Delta F_{\mathrm{sol}}$ (warming). Below diagonal panels: Same as above diagonal but for median net $\Delta F_{\mathrm{net}}$. Purple shades indicate negative $\Delta F_{\mathrm{net}}$ (cooling), while orange shades indicate positive $\Delta F_{\mathrm{net}}$ (warming). All $\Delta F$ are given in $\mathrm{W\,m^{-2}}$. The black arrows point to the direction of the steepest slope.

depends on the units of the parameters. Therefore, the arrows are normalized and can only be interpreted for their direction and not for their length.

## Appendix B: Atmospheric profiles of temperature and relative humidity

The radiative transfer simulations within the present study use the atmospheric profiles from Anderson et al. (1986) that are provided in the libRadtran package. To cover a wide range of temperature conditions, three atmospheric profiles were 780 selected, which represent subarctic, mid-latitude, and tropical conditions given by the `afglsw`, `afglus`, and `afglt` profiles,

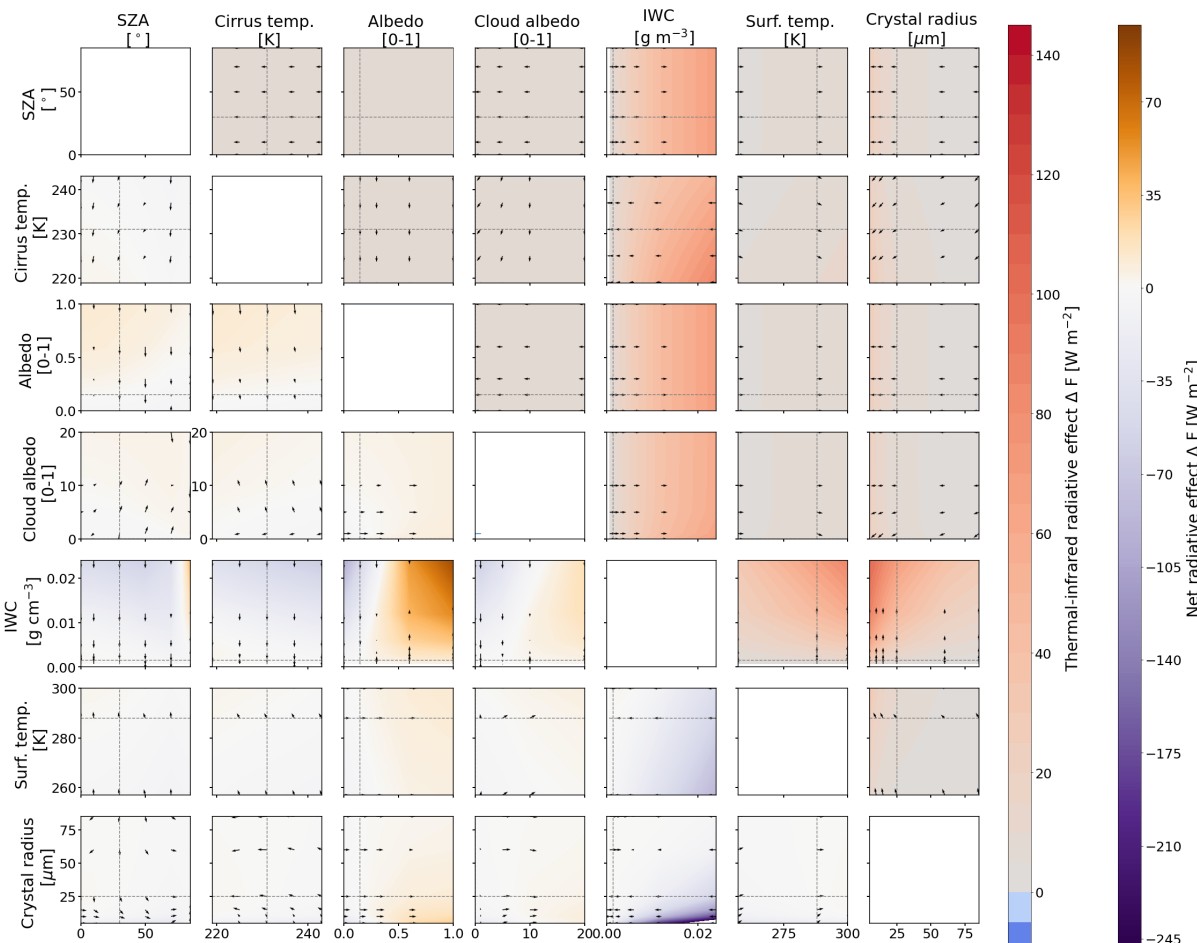

**Figure A2.** Same as Fig. A1 but above diagonal panels present median $\Delta F_{\mathrm{tir}}$. All $\Delta F$ are given in $\mathrm{W\,m}^{-2}$.

respectively. The vertical temperature profiles range from 0 to 120 km and are visualized in Fig. B1a. Figure B1b presents a close-up and Fig. B1c shows the relative humidity profile for 0 to 20 km. The position of the low-level liquid water cloud between 1000 and 1500 m is indicated by the gray shaded area. The positions of the ice cloud altitude are indicated by the colored dots.

According to Anderson et al. (1986) the presented profiles are subject to variations between 10 % and 30 %. Therefore, we multiplied the original profiles profiles by factors of 0.8 and 1.2 to: i) partly account for this variation and ii) to estimate the influence of variations in RH on the simulated solar, TIR, and net RE. The modified profiles with $\pm 20\%$ are indicated by pale colors in Fig B1c.

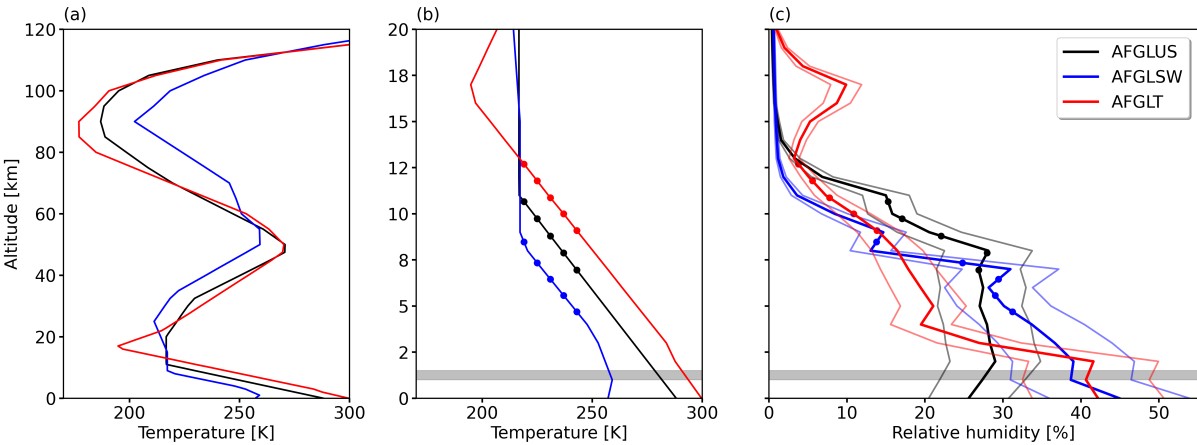

**Figure B1.** Profiles of temperature (a,b) and relative humidity (c) used for the radiative transfer simulations. The subarctic (`afglsw`), mid-latitude (`afglus`), and tropical (`afglt`) profiles are given in blue, black, and red, respectively. The modified profiles with $\pm 20\%$ are indicated by pale colors. The positions of the simulated ice water cloud are indicated by the colored dots for each profile. The position of the low-level liquid water cloud is indicated by the gray shaded area.

## Appendix C: Simulation time and accuracy

The radiative transfer solver DISORT (Stamnes et al., 1988; Buras et al., 2011) allows to select $2N$-number of streams to be used in the radiative transfer simulations. Higher number of streams increases the accuracy of the simulations but also the computational time. To obtain sufficient accuracy while keeping the computational time reasonable, the optimal trade-off was estimated by progressively increasing the number of streams from 4 to 48. The simulation with 48 streams is regarded here as the reference with the highest accuracy and computational time.

The number of streams and the timing of the RT simulations are estimated on the basis of a specific parameter combination, representing a complex cloud scene that is characterized by cloud–cloud–surface-interactions. The simulations are run for a solar zenith angle $\theta = 70°$, a cirrus temperature $T_{\mathrm{cld,ice}}$ of 233 K, a surface albedo $\alpha_{\mathrm{srf}} = 1$, an ice water content IWC = 0.0024 g m$^{-3}$, a surface temperature $T_{\mathrm{srf}} = 288$ K, an ice crystal effective radius $r_{\mathrm{eff}} = 5\,\mu$m, and an additionally underlying liquid water cloud (cloud optical thickness $\tau_{\mathrm{liq}} = 10$).

The computational time that is required for the simulations depends on the available hardware. Therefore, we provide the fraction of the computational time required for $n$ streams to a simulation with 48 streams. The accuracy is given as the relative difference between the cloud RE for a given number of streams with respect to the reference simulation.

  Figure C1 shows that the relative difference in the RE decreases with increasing number of streams (higher accuracy). A significant gain in accuracy is achieved by switching from 4 to 10 streams. For simulations with 12 to 16 streams the relative

difference remains constant at around 0.1 %. Further increasing to 24 streams provides only a slight gain in accuracy, whereas computational time increases disproportionaly. Therefore, the optimal trade-off between accuracy and computational time is obtained with 16 streams, which is the configuration used in this study.

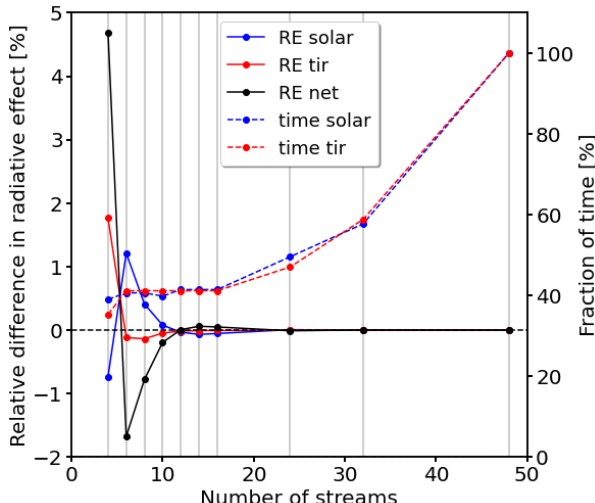

**Figure C1.** Relative deviation (in %) of solar (solid blue), TIR (solid red), and net (solid black) cloud radiative effect from the reference simulation calculated with 48 streams. The computational time is given as a fraction of the computational time needed for the solar (dashed blue) or TIR (dashed red) simulations using the maximum number of 48 streams.

We estimated the uncertainty that is associated with the REPTRAN 'coarse' parameterization instead of the 'fine' resolution by simulating one particular cloud case and running the simulation with both options. The selected simulation is characterized
by: a solar zenith angle $\theta = 70°$, for a long and slanted path through the atmosphere to maximize the impact of molecular absorption; a cirrus temperature $T_{\text{cld,ice}}$ of 233 K, as the center of the parameter space; a surface albedo $\alpha_{\text{srf}} = 0.15$, for moderate surface reflection; an ice water content IWC = 0.012 g m$^{-3}$, a surface temperature $T_{\text{srf}} = 300$ K, to select the tropical atmospheric profile with the highest water vapor concentration; and an ice crystal effective radius $r_{\text{eff}} = 25\,\mu$m. Based on the two simulations, relative differences in the solar, TIR, and net radiative forcing $\Delta F$ of 0.4 %, 0.2 %, and 1.9 % were determined,
respectively.

### Appendix D: Single-scattering phase function $\mathcal{P}$

The shape-effect is primarily caused by differences in the extinction of radiation and the asymmetry parameter. The asymmetry parameter is a measure of the asymmetry of the phase function $\mathcal{P}$ between forward and backward scattering (Macke et al., 1998; Fu, 2007). $\mathcal{P}$ provides the angular distribution of the scattered direction in relation to the incident light. As an example,
Fig. D1a–d shows $\mathcal{P}$ at 550 nm wavelength for columns; plates; droxtals; and Yang's '8–column_aggregates', which are ice crystals consisting of 8 merged columns. The phase functions are extracted from the post-processed libRadtran data set that is based on the ice optics computations from Yang et al. (2013).

All ice crystal shapes are characterized by a dominating peak in the forward direction, which drops by a factor of $10^4$ sr$^{-1}$, when the scattering angle $\Theta$ increases from 0° to 10°. For $10° < \Theta < 160°$, $\mathcal{P}$ varies between $10^{-1}$ sr$^{-1}$ and $10^1$ sr$^{-1}$.

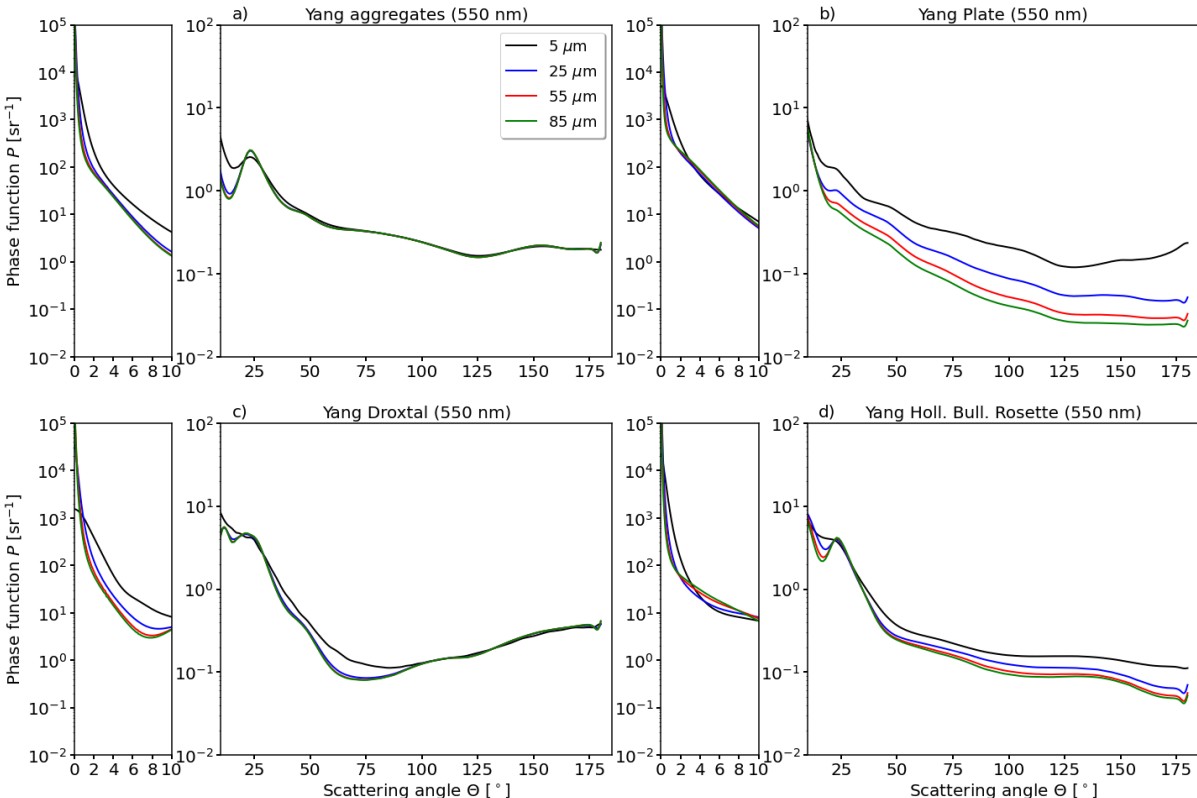

**Figure D1.** Phase function of four different ice crystal shapes with moderate surface roughness and four particle size distributions at 550 nm wavelength. Aggregates are represented by moderately rough aggregates of 8-element columns. Please note the two different $y$-scales to account for the different magnitudes in the forward scattering peak. Plotted $\mathcal{P}$ are post-processed phase functions from Emde et al. (2016) that are based on Yang et al. (2013). The phase functions from Emde et al. (2016) assume a crystal size distribution that follows a gamma function.

Towards $\Theta > 160°$ the phase function increases, showing enhanced backward scattering except for the complex shaped crystals (Fig. D1d). Further characteristics of $\mathcal{P}$ are local maxima at $22°$ scattering angles and cause halo phenomena. Additionally, non-spherical crystals (Fig. D1a,b,d) have enhanced sideward scattering compared to ice crystals with a roughly spherical shape, like droxtals (Fig. D1c) or water droplets. Another characteristic is the shift in the $\mathcal{P}$ from variations in the crystal radius $r_{\mathrm{eff}}$, which is most prominent for plates and lowest for columns.

*Author contributions.* **KW** designed the model setup, conducted the experiments and the data analysis, and prepared the manuscript. **NB** and **OB** contributed equally to the analysis and the preparation of the manuscript.

*Competing interests.* The authors declare no competing interest.

*Acknowledgements.* This research has been supported by the French Ministère de la Transition écologique et Solidaire (N° DGAC 382 N2021-39), with support from France's Plan National de Relance et de Resilience (PNRR) and the European Union's NextGenerationEU. We further would like to thank David L. Mitchell, Andreas Macke, and one anonymous reviewer for their insightful comments during the review that improved and strengthened the manuscript, and Matthias Tesche for serving as editor.

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
