# Peer review of "Sensitivity of cirrus and contrail radiative effect on cloud microphysical and environmental parameters."

_EGUsphere, 2023_

## Referee Comment (RC1)

ACP Review (preprint on EGUsphere at:
https://egusphere.copernicus.org/preprints/2023/egusphere-2023-155/ )

Title: Radiative effect by cirrus cloud and contrails – A comprehensive sensitivity study
Author(s): Kevin Wolf, Nicolas Bellouin, and Olivier Boucher
MS No.: egusphere-2023-155
MS type: Research article
Iteration: Initial submission

General Comments:

The overall concept of this study is commendable and very useful, but there are problems with this study that need to be addressed and resolved before this study can be published. In spite of these problems, the results still appear valid. For example, the authors attempt to treat cirrus cloud properties (effective radius $r_{eff}$ or diameter $D_{eff}$, IWC and $N_{ice}$) using Euclidean geometry (i.e., as spheres), and as with earlier attempts like this, at least one of these variables ends up serving as the "dust bin" (i.e., becomes corrupted, $N_{ice}$ in this case) due to this flawed approach. But since it appears that $D_{eff}$ and IWC are calculated accurately, and the radiation transfer (RT) calculations in libRadtran do not use $N_{ice}$, the results of this study still appear valid.

Another major drawback of this study is that the cirrus cloud geometrical thickness $\Delta z$ is fixed (i.e., it never varies), having a value of 0.20 km. It appears that $\Delta z$ is fixed to enable mathematical closure; otherwise Figure 1 is not possible. More importantly, $\Delta z = 0.2$ km is fine for contrails, but not for natural cirrus clouds, which are typically ~ 1.2 km on average. Since this study claims to be representative of natural cirrus clouds, the authors need a compelling argument to justify using a fixed $\Delta z$ of 0.2 km for such clouds.

The paper is well written and organized, with good quality of figures, and the results should be useful to the atmospheric radiation community. I therefore recommend publication after major revisions. Detailed comments addressing the paper's drawbacks now follow.

Major Comments:

1. Equation 1: In some conventions, $F^{\downarrow}$ is taken to be positive while $F^{\uparrow}$ is taken to be negative, in which case $\Delta F = F_c + F_{cf}$. To avoid any confusion, please mention that all flux quantities are taken to be positive.

2. Lines 127-128: Cirrus clouds are typically ~ 1 km in geometrical thickness; why was a thickness of 0.2 km selected? It is not clear how this unrealistic value impacts the

analysis under "Results"; please explain why the findings of this study are realistic in relation to this choice for geometrical thickness.

3. Equation 6: Petty and Huang (2011) was consulted for the calculation of $v_{eff}$, where it was discovered that $v_{eff}$ has no general analytical solution, making Eq. 6 here unpractical. If there is an analytical solution, it should be given here. For the special case of an exponential particle size distribution or PSD, $\mu = 0$ and $v_{eff} = 1/3$, but libRadtran has set $\mu$ to a value of 1.

4. Lines 155-157 and Eq. 7: Please mention that this $D_{eff}$ definition is the same definition derived in Mitchell (2002, JAS), provided that ice volume V is evaluated at the bulk density of ice (0.917 g/cm$^3$), as shown by the following derivation that begins with Eq. 7:

$$D_{eff} = D_V^3/D_A^2 = (6V/\pi)/(4A/\pi) = (3/2)(V/A) \tag{1}$$

where V is the ice crystal volume at bulk density and A is the mean projected area of the ice crystal, as defined on lines 159-160. But on line 164, the paper states: "where V and A are the average volume and projected area of the crystal population, respectively". It seems like a leap of faith to apply this $D_{eff}$ derived for an ice crystal to a PSD, but in Mitchell (2002) it is shown that this can be done, so please justify this leap of faith and mention the implicit ice density.

5. Equation 11: This could be done more elegantly and accurately by simply selecting appropriate power-law mass-dimension expressions for aggregates, droxtals, hex-plates. From Eq. 29 in Mitchell et al. (2006),

$$N_{ice} = \Gamma(\mu+1)\, IWC\, \Lambda^\beta / (\alpha\, \Gamma(\beta+\mu+1)) , \tag{2}$$

where $\Gamma$ denotes the gamma function, $\mu$ and $\Lambda$ are from Eq. 5 of this paper, and $\alpha$ and $\beta$ are the prefactor and exponent of the ice particle mass-dimension power law relationship (i.e., $m = \alpha D^\beta$). The $r^3$ dependence in Eq. 11 is an artifact of the Euclidean geometrical framework imposed and leads to false interpretations later in the paper, like the top of page 12. For example, from Petty and Huang (2011), $\Lambda = 3/r_e$ for exponential PSDs, giving

$$N_{ice} = 3^\beta\, IWC/(\alpha\, \Gamma(\beta+1)\, r_e^\beta ). \tag{3}$$

Thus, $N_{ice}$ has a $\beta$ dependence on ice particle size (not a cubic dependence as shown in Eq. 11), where $\beta$ tends to be ~ 2 for aggregates, ~ 2.4 for hex-plates and 3 for droxtals.

6. Lines 199-200: The cloud absorption optical depth is also very important in determining RT in the TIR; please mention this.

7. Equation 13: Is this equation used in libRadtran? If not, what is the point in mentioning it? Cloud property input to libRadtran consists of IWC and $r_e$, suggesting the zero-scattering approximation might be used for TIR hemispheric fluxes:

$$\varepsilon = 1 - \exp(-5\ \tau_{abs}/3) \tag{4}$$

where $\varepsilon$ is cloud emissivity and $\tau_{abs}$ is the cloud absorption optical depth. Please indicate whether $\varepsilon$ is calculated in libRadtran, and how it is calculated if applicable.

8. Lines 209 – 213 and Eq. 14: Eqn. (14) appears flawed since, in principle, there should be an emissivity term ($\varepsilon$) for both the surface and the ice cloud. But since typically $\varepsilon \approx 1$ at the surface, does $\varepsilon$ in (14) correspond only to the ice cloud? If so, it would be incorrect to multiply it by $T_{sfc}^4$ (which Eq. 14 does). Later, $\Delta F_{tir}$ is shown for IWC, $r_e$, and ice crystal shape, so it appears that $\varepsilon$ refers to the ice cloud and therefore $\varepsilon < 1$, but how then does $\varepsilon$ depend on IWC, $r_e$ and ice particle shape? The dependence of $\Delta F_{tir}$ on cloud properties is a complete black-box mystery and this needs to be explained.

9. Figure 1: Fixing the cloud thickness appears to be required to get closure for the system of equations producing these four figures. If so, this analysis may not be representative of natural cirrus clouds in some respects since the geometric cloud thickness $\Delta z$ is fixed at 0.2 km corresponding to extremely thin cirrus or contrails. For example, obtaining a typical range of cirrus cloud optical depth requires anomalously high IWC to compensate for the small $\Delta z$, based on the relationship: $\tau_{vis} = 3\ IWC\ \Delta z/(\rho_i\ D_{eff})$. At a minimum, the authors should explain how they obtain mathematical closure to produce these plots.

10. Figure 9a: $N_{ice}$ here has units of $cm^{-3}$ with some values exceeding 100 $cm^{-3}$. In natural cirrus clouds, $N_{ice\_ice}$ rarely exceeds $\sim 2\ cm^{-3}$. This appears to be a consequence of the $r^{-3}$ dependence of $N_{ice}$ in Eq. 11. As shown in Eq. 3 above, the dependence of $N_{ice}$ on $r_e$ is $r_e^{-\beta}$ where $\beta$ typically lies between 1.7 and 3.

11. Lines 258-259: As noted in (1) above, $N_{ice}$ is related to $r_{eff}$ by the power of $-\beta$ (not -3 as stated here).

12. Lines 295-296: How do ice particle shapes affect $\Delta F_{tir}$, given the above comments in 8?

13. Lines 307-314: The aspect ratio strongly impacts the scattering phase function and therefore the asymmetry parameter g (Fu, 2007, JAS; Van Diedenhoven et al., 2012, AMT; 2013, ACP). Please consult these studies and revise this discussion accordingly.

14. Figure 3 caption: What do the numbers refer to in Fig. 3 a-c?

15. Lines 327-329: Macke and Grosklaus (1998) addressed lidar (SW radiation). While their finding about PSDs may be true for SW radiation, Mitchell (2002, JAS) and Mitchell et al. (2011, ACP) found that PSD shape matters considerably for LW radiation.

16. Line 358: This refers to Fig. 5a, correct? Here the upper boundaries are becoming more negative with increasing $\theta$.

17. Figure 5 caption: What do the numbers next to the boxes indicate? They appear to correspond to median, 25th and 75th percentile values, but this should be called out.

18. Line 378: As far as I can tell, Fig. 2 shows that $r_{eff}$ is the primary factor controlling $\Delta F$, not IWC.

19. Lines 506-508: This could have been described more clearly under "Methods" unless I missed something.

Technical Comments:

1. Figure 2 caption: Typo where $r_{eff}$ = 5 µm; should be 45 µm?

2. Line 349: $\Delta F_{tir}$ => $\Delta F_{net}$?

---

## Community Comment (CC1)

**Comment to Wolf et al., "Radiative effect by cirrus cloud and contrails – A comprehensive sensitivity study", in review**

*Dennis Piontek\* and Ulrich Schumann\**

*\* Deutsches Zentrum für Luft- und Raumfahrt (DLR), Institut für Physik der Atmosphäre, 82234 Oberpfaffenhofen, Germany*

The study of the radiative forcing of cirrus and contrails is an important task. In particular the climate impact of contrails gets significant attention in the past years as the avoidance of contrails by next-generation aircraft engines, the rerouting of flights, and the use of sustainable aviation fuels promises to be an easily achievable climate change mitigation strategy. In that sense, we want to applaud the authors for contributing to this endeavor.

The authors present an ambitious study to evaluate the radiative forcing due to ice clouds by performing a large number of radiative transfer calculations (94,000) for different atmospheres, liquid water and ice cloud configurations (i.e., different optical depths and heights), ice crystal sizes and shapes, surface temperatures and albedos, as well as solar zenith angles. The radiative impacts in the thermal infrared and the solar spectrum are quantified. For the calculations, the established radiative transfer code libRadtran (Mayer & Kylling, 2005) was used.

As the authors pointed out, various studies already investigated the cloud radiative forcing with different foci. However, we agree to the third reviewer: the statement in lines 70-71 (most "comprehensive sensitivity study") needs further work to become fully justified. One comparable but missing study is "A Parametric Radiative Forcing Model for Contrail Cirrus" by Schumann et al. (2012a). In this study, libRadtran was used as well to simulate the thermal and solar cloud radiative forcing of contrails, covering different surface and atmospheric conditions, solar zenith angles, seven different ice particle shapes and effective particle radii up to 45 µm, different liquid and ice water configurations. In total, 36,576 calculations were performed. Based on this dataset, approximations of the long- and shortwave radiative forcing due to contrails were derived. The study also shows sensitivity studies with respect to various quantities (e.g., contrail optical depth, solar zenith angle, effective albedo).

Due to the strong similarity of the simulated datasets of Wolf et al. and Schumann et al., it appears mandatory to perform a direct comparison. Thus, we compared in a quick first study the calculations of Wolf et al. with the parameterizations developed by Schumann et al. Those are implemented in the Python package pycontrails (https://py.contrails.earth) which includes (among others) the "Contrail Cirrus Prediction Tool" (CoCiP, Schumann, 2012b).

The approximation of the longwave radiative forcing needs 5 inputs, which we estimated by data from Wolf et al. as follows:

| Input LW RE approx. of Schumann et al. (2012) | Data from Wolf et al. (2023) |
|---|---|
| Outgoing longwave radiation | Upward thermal infrared irradiance (`Fup_tir`) |
| Atmospheric temperature at contrail midlayer | Ice cloud temperature (`ice_cloud_temp`) |
| Contrail optical thickness at 550 nm | Ice cloud optical thickness at 640 nm (`tau`) |
| Optical thickness of cirrus above contrails | Set to zero |
| Contrail ice particle volume mean radius r_vol | Derived from ice crystal effective radius (crystal_effective_radius) using
        Aggregates: r_eff = 0.574 r_vol
        Droxtals: r_eff = 0.94 r_vol
(Schumann et al., 2011) |

The shortwave cloud radiative forcing needs 6 different inputs:

| Input LW RE approx. of Schumann et al. (2012) | Data from Wolf et al. (2023) |
| --- | --- |
| Solar direct radiation | Downward solar irradiance (`Fdn_sol`) |
| Reflected solar radiation | Upward solar irradiance (`Fup_sol`) |
| Solar constant | 1361 W/m² |
| Contrail optical thickness at 550 nm | Ice cloud optical thickness 640 nm (`tau`) |
| Optical thickness of cirrus above contrails | Set to zero |
| Contrail ice particle volume mean radius r_vol | Derived from ice crystal effective radius (crystal_effective_radius) using
        Aggregates: r_eff = 0.574 r_vol
        Droxtals: r_eff = 0.94 r_vol
(Schumann et al., 2011) |

The ice crystal habits are considered separately, as the habit is given as an additional parameter to the radiative forcing functions of pycontrails (here, it is mainly used to convert r_vol back to r_eff internally; the parameterization of Schumann et al., 2012a, relies solely on r_eff and is independent of the ice crystal shape). We considered rough aggregates and droxtals. Wolf et al. also performed calculations for plates.  However, the approximate conversion between r_eff and r_vol is non-linear (Schumann et al., 2011); thus, we did not consider plates for the moment.

Note that the cirrus optical depths provided by Wolf et al. and used in the approximation of Schumann et al. (2012a) are for different wavelengths (640 and 550 nm, respectively). However, we assume that the differences in the ice optical properties are in the order of few percent (Lynch & Mazuk, 2001) and, therefore, negligible.

Unfortunately, also the definitions of "top of atmosphere" differ as Wolf et al. define "top of atmosphere (TOA) at 15 km" height. As a result, the upward thermal infrared irradiance of Wolf et al. can only be considered as an approximation of the outgoing longwave radiation at top of atmosphere in the sense of Schumann et al. (2012a). This is also visible when considering the downward thermal infrared irradiance of Wolf et al., which is not zero but varies between roughly 7 and 10 W/m². The difference in the definition of top of atmosphere has also an impact on the inputs for the solar direct radiation and the reflected solar radiation, as well as the resulting cloud radiative forcings in the long- and shortwave spectrum.

Nevertheless, we find that the results of Wolf et al. and the approximations of Schumann et al. (2012a) are in reasonable agreement (see plots below), with Pearson correlation coefficients of 0.979 and higher. The longwave radiative forcing based on Schumann et al. (2012a) is slightly smaller than the results of Wolf et al. towards the lower end of considered thermal infrared radiative forcings. For the shortwave radiative forcing, we find a larger scatter between both results.

Although these results represent only a first quick look into the matter and further investigations might be necessary, the comparison already seems to show that the calculations presented by Wolf et al. (and, thus, the underlying input datasets and assumptions) agree with the work presented by Schumann et al. (2012a).

[Figure]

*Further major comments to the manuscript:*

- We appreciate that the results in Wolf et al. are close to the results in Schumann et al. (2012), but we miss a discussion of
    a) the variable humidity: It is well known that the relative humidity over ice is often close to 100 % near cirrus and contrails (see Li et al., 2023). But, what is the relative humidity in your profiles?
    b) any other absorbing gases or species (O3, CO2, aerosols)?
- Discussion of importance of large solar zenith angle SZA > 70°: The shortwave radiative forcing reaches a maximum near or above that SZA value, see Figs. 7 and 8 in Schumann et al. (2012), Fig. 12 in Markowicz & Witek (2011), Fig. 1 in Myhre & Stordal (2001); and hence this parameter range is important at sun dawn in early morning/late evening (Meerkötter et al., 1999).
- The problem with high SZA is, however, that clouds in general, and contrail cirrus clouds in particular, can only very roughly be approximated as horizontally homogenous, in particular when the sun is low over the horizon. We miss a study on the 3d-effects of contrails (depending among others on SZA, azimuth of contrail-line direction relative to the sun, on the width/thickness ratio of the contrails lines (Forster et al., 2014), besides the 3d clouds in the contrail neighborhood), besides the effects of non-spherical Earth geometry and solar radiation refraction in the atmosphere at high SZA.
- With respect to your Appendix B: In Schumann et al. (2012), Bernhard Mayer noted: "the irradiances are computed using the discrete ordinate solver by Stamnes et al. (1998), version 2.0, with six streams, which allows accurate simulations of irradiances." We wonder why you need 16 streams and cannot calculate at high SZA? Do you want to say that the former results are significantly inaccurate for methodological reasons? We expect small differences between 6 and 16 streams.
- The test example assumes a surface albedo of one and liquid water clouds below the ice clouds. Hence the solar forcing is small in this case. Is this the best test case?
- Why do you use the older Fortran version of libRadtran? The more stable C-Version is available since 2010.
- Another important issue, which is so far only approximately covered, is the effect of overlapping contrail cirrus clouds. We found (see Schumann, Poll et al., 2021) that Europe is covered frequently by very many contrails which get wide compared to the lateral distances to other contrails so that they partially overlap each other and so that contrails forming above or below the first contrails experience a changed radiation field with different effective OLR/RSR values. We used a rough approximation to account for this effect and found that it changes the computed net RF by a factor of order two over Central Europe, depending on air traffic density and humidity.
- Line 192, Eq. 11: Why do you need the factor $\beta$? The r_vol is defined with $\beta = 1$ for arbitrary habits, see Schumann et al. (2011), Eq. 18, at least for fixed ice density $\rho_{ice}$. More important (besides $\rho_{ice}$ for porous crystals), is the ratio C=r_vol/r_eff, see Eq. 1 in the same paper. Do your results change and how much if you use $\beta = 1$ consistently in your study?

*Minor comments to the manuscript:*

- Why do you use the term "Radiative Effect, RE"? We think that the term "Radiative Forcing, RF" is more often used. What is the difference between RE and RF?
- Line 32: We do not understand why you cite Jensen et al. (1994) here: "contrails are short lived and can persist...". Jensen et al. discuss tropical cirrus, not contrails. Here the paper by Schumann (1996), even if not the first (see also Schumann, 1994, and Busen & Schumann, 1995) is often cited as the most comprehensive introduction of contrails in literature at least until that time (see also Schumann & Heymsfield, 2017a, besides Kärcher, 2018).
- Line 35: Regarding the importance of cirrus cloud cover and contrails over Europe, you may also refer to Schumann, Penner et al. (2015) and Schumann, Bugliaro et al. (2021).

- Line 36: The fact that shortwave radiative forcing is mostly negative is well known. It should be mentioned that it can be positive for high surface albedo and high absorption in the atmosphere between ground and cirrus cloud as discussed in Meerkötter et al. (1999), page 1089, right column. See also Myhre & Stordal (2001), Fig. 1 (but published without explicit explanation).
- Line 137: Presumably the most comprehensive collection of aircraft in-situ and remote sensing measurements of contrail properties can be found in Schumann, Baumann et al. (2017b) and in the therein described open-access contrail library "COLI"; they cover not only young but also the more important aged contrails (partially exceeding 10,000 s).
- Line 158, Eq. 7 to 9: Very similar equations can be found in Schumann et al (2011).
- We find it strange that you cite Meerkötter et al. (1999) in the figure caption of Fig. 2, but do not discuss similarities or disagreements in the content in the text. In fact, we still have to identify any basic new information in your discussion of Fig. 2.
- The discussion of r_eff and IWC as the most important parameter is incomplete and partially misleading (at many places and in particular in section 3.3 and in the summary, line 499). Physically, the most important parameter is the optical depth $\tau$ of the contrail cirrus, which is, among others, a function of r_eff, IWC and cloud geometrical thickness D. The r_eff is a secondary factor besides crystal habit etc. Of course, IWC, r_eff, D and crystal habits are important per se and possibly easier to measure while models might primarily compute the IWC and then estimate crystal habit and optical extinction $\beta_{ext}$ for given IWC and temperature (Heymsfield et al., 2014), but $\tau \sim \beta_{ext}$ D, by definition, is the parameter which characterizes the impact of a cloud layer on radiation transfer.
- The discussion of the importance of the surface temperature is misleading. It is not the surface temperature that is important but the effective brightness temperature of the atmosphere below the contrail cirrus, which in fact depends not only on the surface temperature but also on water vapor and other IR absorber profiles and low-level clouds, besides spectral averaging. It was exactly this reason why Schumann et al. (2012a) parameterized the longwave radiative forcing not as a function of surface temperature (as also done by Corti & Peter, 2009), but as a function of OLR without contrail cirrus.

In summary, we highly appreciate that this study was performed and that we got access to the data, since this gives us the chance to test our parameterizations, but the paper needs considerable extensions and improvements before it can be published as a "comprehensive" study.

*References:*

Busen, R., & Schumann,U.: Visible contrail formation from fuels with different sulfur contents, Geophys. Res. Lett., 22 (11), 1357-1360, 1995.

Corti, T. & Peter, T.: A simple model for cloud radiative forcing, Atmos. Chem. Phys., 9, 5751–5758, 2009.

Forster, L., C. Emde, S. Unterstrasser & B. Mayer: Effects of three-dimensional photon transport on the radiative forcing of realistic contrails. J. Atmos. Sci., 69(July), 2243-2255, 2012.

Kärcher, B.: Formation and radiative forcing of contrail cirrus. Nat. Commun. 9, 1824, 2018.

Heymsfield, A., D. Winker, M. Avery, M. Vaughan, G. Diskin, M. Deng, V. Mitev & R. Matthey: Relationships between ice water content and volume extinction coefficient from in situ observations for temperatures from 0° to -86°C: Implications for spaceborne lidar retrievals. J. Appl. Meteor. Clim., 53, 479-505, 2014.

Li, Y., Mahnke, C., Rohs, S., Bundke, U., Spelten, N., Dekoutsidis, G., Groß, S., Voigt, C., Schumann, U., Petzold, A., and Krämer, M.: Upper-tropospheric slightly ice-subsaturated regions: frequency of occurrence and statistical evidence for the appearance of contrail cirrus, Atmos. Chem. Phys., 23, 2251–2271, 2023.

Lynch, D.K., & Mazuk, S.M.: Wavelength Dependence of Cirrus Optical Depth, Aerospace Report No. TR-2001(8570)-1, Space and Missile Systems Center, Air Force Space Command, Los Angeles, CA, USA, https://apps.dtic.mil/sti/pdfs/ADA399481.pdf, 2001.

Markowicz, K. M., & Witek, M. L.: Simulations of Contrail Optical Properties and Radiative Forcing for Various Crystal Shapes. J. Appl. Meteor. Climatol., 50, 1740–1755, 2011.

Mayer, B. and Kylling, A.: Technical note: The libRadtran software package for radiative transfer calculations - description and examples of use, Atmos. Chem. Phys., 5, 1855–1877, 2005.

Meerkötter, R., Schumann, U., Doelling, D.R. et al.: Radiative forcing by contrails. Annales Geophysicae 17, 1080–1094, 1999.

Myhre, G., & Stordal, F.: On the tradeoff of the solar and thermal infrared radiative impact of contrails, Geophys. Res. Lett., 28, 16, 3119-3122, 2001.

Schumann, U.: On the effect of emissions from aircraft engines on the state of the atmosphere, Ann. Geophys., 12, 365–384, 1994.

Schumann, U.: On conditions for contrail formation from aircraft exhausts, Meteorologische Zeitschrift, 5, 1, 4-23, 1996.

Schumann, U., Mayer, B., Gierens, K., Unterstrasser, S., Jessberger, P., Petzold, A., Voigt, C., & Gayet, J-F: Effective Radius of Ice Particles in Cirrus and Contrails, J. Atmos. Sci., 68(2), 300-321, 2011.

Schumann, U.: A contrail cirrus prediction model, Geosci. Model Dev., 5, 543–580, 2012a.

Schumann, U., Mayer, B., Graf, K., & Mannstein, H.: A Parametric Radiative Forcing Model for Contrail Cirrus, J. Appl. Meteorol. Climatol., 51(7), 1391-1406, 2012b.

Schumann, U., Penner, J. E., Chen, Y., Zhou, C., and Graf, K.: Dehydration effects from contrails in a coupled contrail–climate model, Atmos. Chem. Phys., 15, 11179–11199, 2015.

Schumann, U., and A. J. Heymsfield: On the Life Cycle of Individual Contrails and Contrail Cirrus. Meteor. Monogr., 58, 3.1–3.24, 2017a.

Schumann, U., Baumann, R., Baumgardner, D., Bedka, S. T., Duda, D. P., Freudenthaler, V., Gayet, J.-F., Heymsfield, A. J., Minnis, P., Quante, M., Raschke, E., Schlager, H., Vázquez-Navarro, M., Voigt, C., and Wang, Z.: Properties of individual contrails: a compilation of observations and some comparisons, Atmos. Chem. Phys., 17, 403–438, 2017b.

Schumann, U., Bugliaro, L., Dörnbrack, A., Baumann, R., & Voigt, C.: Aviation contrail cirrus and radiative forcing over Europe during 6 months of COVID-19. Geophysical Research Letters, 48, e2021GL092771, 2021.

Schumann, U., Poll, I., Teoh, R., Koelle, R., Spinielli, E., Molloy, J., Koudis, G. S., Baumann, R., Bugliaro, L., Stettler, M., and Voigt, C.: Air traffic and contrail changes over Europe during COVID-19: a model study, Atmos. Chem. Phys., 21, 7429–7450, 2021.

---

## Author Comment (AC1)

**Reply to Reviewer #1**
(Referee comment on "Radiative effect by cirrus cloud and contrails – A comprehensive sensitivity study" by Kevin Wolf et al., EGUsphere, https://doi.org/10.5194/egusphere-2023-155-RC1, 2023)

We thank the Reviewer for the time she/he spent on the manuscript. The comments helped to improve the manuscript, but more importantly spurred us into repeating our calculations with (1) a completely revised libradtran configuration to ensure that we use state-of-the-art parametrization; and (2) much extended parameter ranges to be better representative of cirrus and contrails. The discussion in the manuscript has been revised to reflect the new calculations and analyses. In the following, the Reviewer's comments and the corresponding responses are listed. The page and line references given by the Reviewer relate to the manuscript in discussion. Numbers given from our side relate to the revised manuscript.

For better legibility, the Reviewer's comments are highlighted in **bold** and changes in the manuscript are in *italic*.
* * *
General Comments:

**The overall concept of this study is commendable and very useful, but there are problems with this study that need to be addressed and resolved before this study can be published. In spite of these problems, the results still appear valid. For example, the authors attempt to treat cirrus cloud properties (effective radius reff or diameter $D_{eff}$, IWC and $N_{ice}$) using Euclidean geometry (i.e., as spheres), and as with earlier attempts like this, at least one of these variables ends up serving as the "dust bin" (i.e., becomes corrupted, Nice in this case) due to this flawed approach. But since it appears that $D_{eff}$ and IWC are calculated accurately, and the radiation transfer (RT) calculations in libRadtran do not use $N_{ice}$, the results of this study still appear valid.**

**Another major drawback of this study is that the cirrus cloud geometrical thickness Δz is fixed (i.e., it never varies), having a value of 0.20 km. It appears that Δz is fixed to enable mathematical closure; otherwise Figure 1 is not possible. More importantly, Δz = 0.2 km is fine for contrails, but not for natural cirrus clouds, which are typically ~ 1.2 km on average. Since this study claims to be representative of natural cirrus clouds, the authors need a compelling argument to justify using a fixed Δz of 0.2 km for such clouds.**

**The paper is well written and organized, with good quality of figures, and the results should be useful to the atmospheric radiation community. I therefore recommend publication after major revisions. Detailed comments addressing the paper's drawbacks now follow.**

We address these comments below.

**Major Comments:**
**1. Equation 1: In some conventions, F↓ is taken to be positive while F↑ is taken to be**

**negative, in which case ΔF = $F_c$ + $F_{cf}$. To avoid any confusion, please mention that all flux quantities are taken to be positive.**

The manuscript explains that all values are taken positive. We rephrased the sentence and made it clearer:

> *"where the upward and downward, cloudy and cloud-free irradiances are all counted positive."*

**2. Lines 127-128: Cirrus clouds are typically ~ 1 km in geometrical thickness; why was a thickness of 0.2 km selected? It is not clear how this unrealistic value impacts the analysis under "Results"; please explain why the findings of this study are realistic in relation to this choice for geometrical thickness.**

The Reviewer is right. While 0.2 km is realistic for contrails, the value is untypical for natural cirrus. Considering also the comments be the other Reviewers, the simulations have been revised. In the new simulations the cloud geometric thickness is set to 1 km to represent aged contrails and natural cirrus. Selecting a cloud geometric thickness of 1 km is supported by citing the relevant literature.

> *" […] Within the simulations, the ice cloud geometric thickness dz is set to 1000 m for all simulations, which represents an average for observed contrails as well as natural cirrus (Freudenthaler 1995, Sassen 2001, Noel 2007, Iwabuchi 2012."*

**3. Equation 6: Petty and Huang (2011) was consulted for the calculation of $\nu_{eff}$, where it was discovered that $\nu_{eff}$ has no general analytical solution, making Eq. 6 here unpractical. If there is an analytical solution, it should be given here. For the special case of an exponential particle size distribution or PSD, μ = 0 and $\nu_{eff}$ = 1/3, but libRadtran has set μ to a value of 1.**

The Reviewer is right. The analytical solution is only available for μ = 0 with $\nu_{eff}$ = 1/3 and Λ=3/$r_{eff}$. In libradtran, μ is set to 1 and $\nu_{eff}$ is set to 0.25, which is based on observations of ice particle size distributions (Evans 1998; Heymsfield 2002). The entire section and set of equations were revised and we direct the Reviewer to the provided diff-file for the new text.

**4. Lines 155-157 and Eq. 7: Please mention that this Deff definition is the same definition derived in Mitchell (2002, JAS), provided that ice volume V is evaluated at the bulk density of ice (0.917 g/cm3), as shown by the following derivation that begins with Eq. 7: Deff = DV3/DA2 = (6V/π)/(4A/π) = (3/2) (V/A) (1) where V is the ice crystal volume at bulk density and A is the mean projected area of the ice crystal, as defined on lines 159-160. But on line 164, the paper states: "where V and A are the average volume and projected area of the crystal population, respectively". It seems like a leap of faith to apply this Deff derived for an ice crystal to a PSD, but in Mitchell (2002) it is shown that this can be done, so please justify this leap of faith and mention the implicit ice density.**

This comment is linked with comment 3 above. The entire paragraph was modified, the citation to Mitchel (2002) is included, and we direct the Reviewer to the diff file for the new text.

**5. Equation 11: This could be done more elegantly and accurately by simply selecting appropriate power-law mass-dimension expressions for aggregates, droxtals, hex-plates. From Eq. 29 in Mitchell et al. (2006), Nice = Γ(μ+1) IWC Λβ / (α Γ(β+μ+1)) , (2) where Γ denotes the gamma function, μ and Λ are from Eq. 5 of this paper, and α and β are the prefactor and exponent of the ice particle mass-dimension power law relationship (i.e., m = αDβ). The r3 dependence in Eq. 11 is an artifact of the Euclidean geometrical framework imposed and leads to false interpretations later in the paper, like the top of page 12. For example, from Petty and Huang (2011), Λ = 3/re for exponential PSDs, giving Nice = 3β IWC/(α Γ(β+1) reβ ). (3) Thus, Nice has a β dependence on ice particle size (not a cubic dependence as shown in Eq. 11), where β tends to be ~ 2 for aggregates, ~ 2.4 for hex-plates and 3 for droxtals.**

The equation was intended to provide a rough guidance for the reader. Nevertheless, the Reviewer is right and the suggested relationship more accurately represents nature. The set of equations and the accompanied text have been revised. Please see the diff file.

**6. Lines 199-200: The cloud absorption optical depth is also very important in determining RT in the TIR; please mention this.**

The Reviewer is right. However, in course of the revision of the paper the section about blackbody emission has been removed from the paper.

**7. Equation 13: Is this equation used in libRadtran? If not, what is the point in mentioning it? Cloud property input to libRadtran consists of IWC and re, suggesting the zero-scattering approximation might be used for TIR hemispheric fluxes: ε = 1 - exp(-5 τabs/3) (4) where ε is cloud emissivity and τabs is the cloud absorption optical depth. Please indicate whether ε is calculated in libRadtran, and how it is calculated if applicable.**

The DISORT solver in libradtran (Buras et al 2011) calculates scattering in the TIR on basis of the bulk-scattering properties of ice crystals, analog to the solar wavelength range. Thus, the zero-scattering approximation is not used in the simulations. Equation 13 was added to the manuscript to provide guidance for the reader. To avoid misinterpretation the equation is brought into context and is expanded to section "2.4 Approximation of radiative transfer in the thermal-infrared", to incorporate suggestions from other Reviewers.

**8. Lines 209 – 213 and Eq. 14: Eqn. (14) appears flawed since, in principle, there should be an emissivity term (ε) for both the surface and the ice cloud. But since typically ε ≈ 1 at the surface, does ε in (14) correspond only to the ice cloud? If so, it would be incorrect to multiply it by Tsfc4 (which Eq. 14 does). Later, ΔFtir is shown for IWC, re, and ice crystal shape, so it appears that ε refers to the ice cloud and therefore ε < 1, but how then does ε depend on IWC, re and ice particle shape? The dependence of ΔFtir on cloud properties is a complete black-box mystery and this needs to be explained.**

As mentioned in our reply to comment 7, a dedicated section for TIR RT was added to the manuscript. It is primarily based on the TIR RT approximation given by Corti and Peter (2009). Equation 14 is now replaced by Eq. 20. Major steps to derive Eq. 20 are given in the manuscript; details can be found in Corti and Peter (2009).

**9. Figure 1: Fixing the cloud thickness appears to be required to get closure for the system of equations producing these four figures. If so, this analysis may not be representative of natural cirrus clouds in some respects since the geometric cloud thickness Δz is fixed at 0.2 km corresponding to extremely thin cirrus or contrails. For example, obtaining a typical range of cirrus cloud optical depth requires anomalously high IWC to compensate for the small Δz, based on the relationship: $\tau_{vis}$ = 3 IWC Δz/($\rho i$ Deff). At a minimum, the authors should explain how they obtain mathematical closure to produce these plots.**

All simulations have been repeated with a cloud geometric thickness of 1 km. Figure 1 has been revised accordingly. The method to calculate the concentration of ice crystals is given.

> *"[…] N_ice is approximated by Eq. 14, assuming droxtals (almost spherical ice crystals), a mono-disperse particle size distribution, and a cloud geometric thickness dz of 1 km.  [...]"*

**10. Figure 9a: Nice here has units of cm-3 with some values exceeding 100 cm-3. In natural cirrus clouds, Nice_ice rarely exceeds ~ 2 cm-3. This appears to be a consequence of the r-3 dependence of Nice in Eq. 11. As shown in Eq. 3 above, the dependence of $N_{ice}$ on re is re-β where β typically lies between 1.7 and 3.**

In line with the previous comments a sentence is given that explains the calculation. $N_{ice}$ is approximated by Eq. 15, assuming droxtals (almost spherical ice crystals), a mono-disperse particle size distribution, and a cloud geometric thickness d$z$ of 1 km. The cloud optical thickness $\tau_{ice}$ at 550 nm wavelength is directly calculated by libRadtran using optical properties from droxtals. Please see the previous comment(s) and annotations as well as the diff file.

**11. Lines 258-259: As noted in (1) above, Nice is related to reff by the power of -β (not -3 as stated here).**

The Reviewer is right and the sentence has been modified accordingly.

> *"As expected, variations in $r_{eff}$ have the largest effect on the solar, TIR, and net ΔF , as Nice relates to $r_{eff}$ by the power of −β, which depends on the particle shape (see Sec. 2.3 and Eq. 14).  [...]"*

**12. Lines 295-296: How do ice particle shapes affect ΔFtir, given the above comments in 8?**

As stated in comment 8, RT simulations with DISORT rely on the single-scattering albedo, which depends on the particle size distribution, ice water content, and selected effective radius. Keeping IWC and the effective radius constant but changing the particle shape directly

influences the particle size distribution and the related effective radius. The Reviewer points out that Mitchell (2002, JAS) and Mitchell et al. (2011, ACP) found that the shape of the PSD matters considerably for LW radiation and the ice water content and effective radius is not sufficient to describe the radiative properties of ice clouds. We added this information to the manuscript.

> *"The spread in $\Delta F_{sol}$ across crystal shapes with the same $r_{eff}$ and IWC can be interpreted as a potential uncertainty in $\Delta F_{sol}$ due to the ice crystal shape. One has to keep in mind that the differences partially result from deviating crystal size distributions as these depend on the selected crystal shape. Macke et al. (1998) showed that, in the solar wavelength range, the crystal shape is the main driver and the actual ice particle size distribution has only a minor effect on $\Delta F_{sol}$. Nevertheless, Mitchell et al. (1996) and Mitchell et al. (2011) found that the particle size distribution also has a considerable impact on $\Delta F_{tir}$, leading to differences of up to 48% in the single-scattering albedo, when switching between PSD. [...]"*

**13. Lines 307-314: The aspect ratio strongly impacts the scattering phase function and therefore the asymmetry parameter g (Fu, 2007, JAS; Van Diedenhoven et al., 2012, AMT; 2013, ACP). Please consult these studies and revise this discussion accordingly.**

The Reviewer highlights an important fact. The aspect ratio has a significant influence on the asymmetry parameter and we added this information to the manuscript. The entire section was revised. Please see the diff file for the revised version.

> *"Scattering and absorption by an ice crystal is characterized by its orientation, complex refractive index of ice, the wavelength of the incident light, shape, size, and the resulting asymmetry parameter. The asymmetry parameter is a measure of the asymmetry of the phase function P between forward and backward scattering (Macke et al., 1998; Fu, 2007). P provides the angular distribution of the scattered direction in relation to the incident light. For example, in case of idealized hexagonal ice crystals and wavelength below 1.4 µm, the asymmetry parameter is primarily determined by the ice crystal shape / aspect ratio but for wavelength larger then 1.4 µm the asymmetry parameter also depends on the ice crystal size (Fu, 2007; Yang and Fu, 2009; van Diedenhoven et al., 2012). Consequently, the assumption of an ice crystal habit and ice crystal size, with related aspect ratio, are vital information to estimate the ice cloud RE."*

**14. Figure 3 caption: What do the numbers refer to in Fig. 3 a-c?**

A sentence was added to explain the meaning of the numbers.

> *"The numbers indicate the optical thickness simulated for the reference cloud that contains ice aggregates."*

**15. Lines 327-329: Macke and Grosklaus (1998) addressed lidar (SW radiation). While their finding about PSDs may be true for SW radiation, Mitchell (2002, JAS) and Mitchell et al. (2011, ACP) found that PSD shape matters considerably for LW radiation.**

The Reviewer highlights an important point, which is now mentioned in the manuscript. The respective text is included in the modified section quoted in the reply to comment 12.

**16. Line 358: This refers to Fig. 5a, correct? Here the upper boundaries are becoming more negative with increasing θ.**

This is correct. The paragraph is introduced explicitly referring to Figure 5a.

**17. Figure 5 caption: What do the numbers next to the boxes indicate? They appear to correspond to median, 25th and 75th percentile values, but this should be called out.**

We added a sentence that explains the figures.

> *"[…] Red and black numbers indicate the 25th- and 75th percentiles, as well as the median value, respectively."*

**18. Line 378: As far as I can tell, Fig. 2 shows that reff is the primary factor controlling ΔF, not IWC.**

The Reviewer is right. The sentence has been changed.

> *"As presented in Fig. 2, the IWC is the second most influencing factor that controls ΔF. […]"*

**19. Lines 506-508: This could have been described more clearly under "Methods" unless I missed something.**

We agree with the Reviewer and added a paragraph that describes the sampling method more clearly. It is added to section "3 Results" to help to understand the results.

> *"To go beyond these basic dependencies, the impact of each parameter is estimated by fixing one parameter at a time, while the others can vary. For example, in case of $r_{eff}$, all simulations, for steps of $r_{eff}$ given in Table 4, are extracted from the 8-D hypercube. The extracted sub-sample, in the example for a specific $r_{eff}$, is used to calculate the distributions of solar, TIR, and net ΔF. These distributions are then visualized by box plots and characterized by their minimum, maximum, median, as well as the 25th- and 75th-percentiles. This strategy can be interpreted as a type of sub-sampling, by averaging all unfixed parameters to project ΔF onto the one-dimensional space [...]"*

Technical Comments:

**1. Figure 2 caption: Typo where reff = 5 μm; should be 45 μm?**

The Reviewer is right and the typo has been fixed and adapted the new upper boundary of 85 μm.

**2. Line 349: ΔFtir => ΔFnet?**

The Reviewer is right and the sentence has been changed accordingly.

---

## Author Comment (AC2)

**Reply to Reviewer #2 (Andreas Macke)**
(Referee comment on "Radiative effect by cirrus cloud and contrails – A comprehensive sensitivity study" by Kevin Wolf et al., EGUsphere, https://doi.org/10.5194/egusphere-2023-155-RC2, 2023)

We thank the Reviewer Andreas Macke for the time he spent on the manuscript. The comments helped to improve the manuscript, but more importantly spurred us into repeating our calculations with (1) a completely revised libradtran configuration to ensure that we use state-of-the-art parametrization; and (2) much extended parameter ranges to be better representative of cirrus and contrails. The discussion in the manuscript has been revised to reflect the new calculations and analyses. In the following, the Reviewer's comments and the corresponding responses are listed. The page and line references given by the Reviewer relate to the manuscript in discussion. Numbers given from our side relate to the revised manuscript.

For better legibility, the Reviewer's comments are highlighted in **bold** and changes in the manuscript are in *italic*.
* * *
General remarks:
**The manuscript describes an impressive sensitivity study (very nicely summarized in Figure 2, indeed!)) on the importance of the governing physical parameters of cirrus clouds and contrails on their radiative effects in the climate system. Numerous studies on the influence of various parameters already exist, but not on this scale presented here. The authors also largely correctly refer to the previous literature, but I would have liked to see a somewhat more quantitative presentation here. A table roughly summarizing the parameter variations and effects on the radiation effect of previous work could be helpful.**

**I understand that even 94,000 radiative transfer simulations cannot cover all cases of real-world clouds and illumination geometries. The authors should therefore make somewhat more prominent (not just at the end of the manuscript) which assumptions in their calculations constrain the phase space. This seems particularly important to me because, while the authors commendably make their data available for further radiative effect studies, there is then a danger that it will be used without further questioning. For example, ice clouds generally have a distinctive vertical structure of crystal sizes and shapes, which affects both solar reflectivity and thermal emission. Horizontal crystal orientation - as often observed - also has an effect, as does 3D radiative transfer for optically thicker ice clouds. Similarly, a crystal size distribution is always also a crystal shape distribution, so distinguishing clouds consisting of only one crystal shape is somewhat unrealistic. It is not for nothing that Baum et al. (2005) combined size and shape distributions to obtain more realistic optical properties. I realize that one cannot account for all of this in a large sensitivity study, but limitations should be clearly pointed out.**

**Some results are quite obvious, e.g. that the solar cooling effect is determined by the albedo differences of cloud and ground, and the warming effect by the temperature differences of cloud and ground. It is also not necessary to point out several times that**

**the solar parameters do not affect the terrestrial radiation effects and vice versa.**

**The study of a water cloud underlying the ice cloud seems somewhat contrived to me, see the specific references below.**

**Would it somehow be possible to reduce the number of figures, e.g. take only those whose results are referred to in the summary at the end? See also my comments below.**

These general remarks are addressed below when they are repeated in the specific remarks.

Specific remarks:

**line 54-56: I agree that liquid water clouds have simpler microphysics. However, this simplification is perhaps surpassed by the problem of 3D radiative transfer in such clouds.**
**Therefore, I would not say that radiative transfer in cirrus clouds is more complex.**

The sentence was rephrased to specify that we refer to the direct interaction of radiation and particle.

> *"To estimate the radiative impact of a cloud as well as related potential uncertainties and sensitivities, RT simulations represent a helpful tool. While the atmospheric RT in liquid water clouds composed of spherical cloud droplets can rely on geometric optics or Mie-scattering theory (Mie,1908; van de Hulst, 1981), RT simulations of ice particles are made complicated by the non-spherical shape and the interaction with the incoming radiation, i.e., through their single-scattering phase function. The single-scattering phase function, for example, has to be determined by computationally-expensive methods, like ray tracing (Bi et al., 2014), Monte Carlo simulations (Macke et al., 1996a, b), or the T-matrix method (Mishchenko, 2020)."*

**68-69: Why distinguish between sensitivity to size and to size distribution?**

The Reviewer is right. Zhang directly refers to ice crystal size distribution, while Mitchell refers to the effective radius, which is determined by the ice crystal size distribution. Therefore, both studies investigate the impact of the ice crystal size distribution on the radiative forcing / effect of cirrus clouds. The sentence has been rephrased.

> *"The effect of the ice crystal size distribution on cloud radiative forcing / effect was analyzed, for example, by Zhang et al. (1999) or Mitchell et al. (2011)."*

**104-105: ...but then you need to show/cite that 2d or 3d variability is not a driving parameter. And the present work is not even 1d (vertically resolved), but 0d (plane-parallel homogeneous).**

Thank you for this helpful comment. In line with the comments from Reviewer 3, the potential uncertainties due to 3D scattering effects and heterogeneous clouds are now mentioned and referenced in the sections "Introduction" and "Radiative transfer simulation set-up". Due to the length of the new paragraph in the Introduction we would like to direct the Reviewer to the diff file. The second paragraph is given below.

> *"The RT simulations are performed with the 1D solver DISORT (Buras et al., 2011), which is part of libRadtran. Clouds are assumed to be horizontally uniform and lateral photon transport between columns is neglected, which is called the independent pixel approximation (IPA, Stephens et al., 1991; Cahalan et al., 1994). As the main objective of this study is to map the basic dependencies of ΔF on the driving parameters, we neglect any variability in the spatial ice water content (IWC) distribution that exists in cirrus (Minnis et al., 1999). We also restrict the simulations to fully cloud covered scenes. [...]"*

**135-136: according to the title, the work is about cirrus and contrails. So, do the 3 shapes suffice for cirrus as well? Does the aspect ratio of the hexagonal particles varies with size?**

Within this study we focused on three particle shapes that represent three stages of contrail development from almost spherical particles over plates – often used in remote sensing applications - to complex aggregates. We now provide a more detailed literature overview and some references that support the selected ice crystals shapes, which confirms that the selected shapes are representative for the majority of contrails and cirrus. Due to the length of the added section we direct the Reviewer to item 4 in section 2.2 'Radiative transfer setup'.

**Table 3: I understand that some hard choices have to be made if one is to make sense of the parameter space of the physical properties of cirrus clouds. However, it seems to me that the range of only three cloud temperatures is very limited compared to the parameters that make up the optical thickness (IWC, $r_{eff}$). The cloud greenhouse effect is thus much more discretized than the albedo effect.**

Similar to the selection of three particle shapes, the step size for each parameter had to be limited. Nevertheless, we extended the number of simulations. Now, five cirrus temperatures are simulated to better capture the effect changes in ice cloud temperature and altitude. Also the number of simulated solar zenith angles and effective radius were increased.

**167: which r_min and r_max where chosen for the gamma size distribution?**

According to Emde et al (2016), the bulk optical properties are calculated for $r_{min}$ of 5 μm and $r_{max}$ of 90 μm. This information is added to the manuscript.

> *"[…] For the gamma size distribution a minimum and maximum $r_{eff}$ of 5 and 90 μm are selected. [...]"*

**173: isn't the effect of an underlying cloud not somehow accounted for already by varying surface albedo and surface temperature?**

We partly agree with the Reviewer. As similar comment concerning section 3.6 is raised below, we answer both comments below.

**190-192 and eq. (11): This rearranging only work if r^3_vol is not a function of r. But I'd think that this parameter is very much a function of r.**

The Reviewer is right. Incorporating additional Reviewer comments, the equation was modified. Equation 14 became Eq. 15 in the new manuscript. Please see the document and the latex diff file for the updated version.

**220-221: what do you mean with "diagnosed by libradtran"? For a given size and shape, the extinction coefficient should be readily available, given that the extinction efficiency = 2 for large particles.**

"Diagnosed' is an inappropriate term and the sentence was rephrased. The values of tau is directly calculated by libRadtran and extracted from the verbose output.

> *"[…] The cloud optical thickness $\tau_{ice}$ at 550 nm wavelength, given in Fig. 1d–d, is directly calculated by libRadtran using optical properties from droxtals."*

**228: The term "observed" may be misleading as this is about modeling, not observations.**

The sentence was rephrased.

> *"The inherent dependencies of […] "*

**Fig. 1: Since only theoretical relations between the dependent quantities N, IWC, $r_{eff}$, and tau are shown here, which are rather clear, one could omit this discussion and refer to a textbook on radiative transfer.**

The intention of this subsection and figure was to provide a condensed overview of the basic dependencies that will help readers, who might not be familiar with radiative transfer and the interactions of $N_{ice}$, IWC, and $r_{eff}$, to better interpret the relations and figures that follow later in the paper. Therefore, we would strive to keep this section.

**307: "To some extend" -> "For idealized hexagonal columns and plates"**

Thank you for providing the more precise formulation. It is adopted in the manuscript together with the modifications considering other Reviewers comments. Please see the diff file for the revised section.

**327-328: Macke and Großklaus is about rain drops :), you probably meant: Macke A, Francis P-N, Mc Farquhar G-M, Kinne S (1998) The role of ice particle shapes and size distributions in the single scattering properties of cirrus clouds. Journal of Atmospheric Sciences 55 (17), 2874-2883.**

The Reviewer is right and the citation was changed accordingly.

**360-361. wrt the forward peak: The forward peak (0 degree scattering angle) is never directed upward. Are you refering to the forward scattering range?**

The Reviewer is right. Here we do refer to the forward scattering range given by scattering angles <= 90 degree. The sentence was modified accordingly.

> *"[…] and iii) an increase in upward scattered radiation with increasing θ as the light rays get slanted and a larger fraction of radiation from the forward scattering range is directed upward."*

**Figs. 5b and 6b can be omitted.**

We partly agree with the Reviewer. While there is no impact of the solar zenith angle or the solar surface albedo on the TIR component, we would like to keep these plots to provide a systematic overview throughout all parameters and for symmetry with the discussion of the other parameters.

**378: "IWC is the primary factor...": Not according to Fig 2 and your previous explanation that solar and terrestrial effects of IWC cancel out each other. Do I misunderstand something here?**

The Reviewer is right the sentence is false and is now rephrased.

> *[...] IWC is the second most influencing factor [...].*

**389: "...photon path length ... has an almost negligible impact on the cloud RE in the solar and TIR.": photon path lengths in solar and thermal IR are not the same. Did you specifically calculate the mean free path length at the thermal IR? Which wavelength? Water vapor or $CO_2$ absorption might also affect the path length.**

The mean photon path length has not been calculated explicitly. We generally refer to the increased cloud optical thickness and to the fact that the cloud becomes more opaque in the solar and TIR wavelength range with increasing IWC. Following the comment, we rephrased the sentence**.**

> *"For $\Delta F_{tir}$ the increase in IWC leads to an intensified warming effect (Fig. 8b). Again, this is caused by the increase in the total particle scattering and absorption cross-sections. Similar to $\Delta F_{sol}$, the steepest increase appears for IWC < 0.012 g m$^{-3}$, while for larger IWC the medians approach an almost constant level and a further increase in IWC has only a limited effect on $\Delta F_{tir}$."*

**418 - 419: "indicates an increase in the sensitivity of ΔFsol, particularly with respect to reff": Wasn't that already obvious from Fig. 2?**

We partly agree with the Reviewer. Nevertheless, Fig. 9 provides a more nuanced overview of $F_{sol}$ on IWC and $r_{eff}$ as there is an additional separation for $\alpha_{srf}$ and the solar zenith angle. The sentence was slightly rephrased to include the Reviewers comment.

*"[…] ΔFsol, particularly with respect to $r_{eff}$ , as it is expected from Fig. 2"*

**3.4.2: The title is "Thermal IR", but the text below is about $F_{net}$**

The Reviewer is right. Now, the section explains $F_{tir}$ and $F_{net}$. The title was modified accordingly.

*"Thermal-infrared and net radiative effect"*

**3.6: Again, I would think that the radiative boundary conditions that arise from an underlying cloud are covered by the variations in surface albedo and surface temperature, already.**

We partly agree with the Reviewer that, from a radiative transfer perspective, a variation in the cloud optical thickness of the second cloud layer is already cover in the variation of the surface albedo or surface temperature. Nevertheless, we think that the discussion is beneficial for readers, who are not (yet) familiar with the topic of radiative transport and the interactions of radiation, surface, and clouds. In addition, including the second water layer in the provided data set allows to directly access simulated RF for combinations of surface properties and second cloud layer cloud optical thickness. For example, a user can directly access the cloud RF of a cloud over a surface with a certain surface albedo and second liquid water cloud (of certain COT). Without the second cloud layer, the user would be forced to transfer the COT of the liquid water cloud to an equivalent surface albedo. The transfer would have to be parameterized, which adds an additional uncertainty.

**501-502: Of course, F_sol and Delta F_sol = 0 during night. But given this obvious day-night differences in the contributions of F_sol to F_net, wouldn't it not make more sense to study F_net for 24h means?**

The Reviewer is right but this is beyond of the scope of this study in which we intended to investigate the basic dependencies of RT in ice clouds on the selected parameters.

**509: "Delta F_sol is dominated by Delta F_tir": Typo? F_sol -> F_net?**

The sentence was incorrectly phrased and is now modified.

*"For all θ and the majority of the simulations, negative $\Delta F_{sol}$ is exceeded by positive $\Delta F_{tir}$ and leads to a positive median $\Delta F_{net}$ (warming)."*

**511-512: alpha_srf = 1 is rather unrealistic on this planet. So, I don't think that solar warming ever occurs.**

The sentence was rephrased and generalized. The simulations showed a transition from a solar cooling to a solar warming between $\alpha_{srf}$ of 0.6 and 1.0. Since sea ice can have $\alpha_{srf}$ between 0.6 and 1.0, we argue that, under some circumstances, cirrus and contrails can have a solar warming and the figure is valuable.

*[…] except for $\alpha_{srf}$ approaching 1 [...]*

**515: "the resulting net RE is a warming.": -> small.**
**The competition alone does not explain a warming or cooling.**

The Reviewer is right and the sentence was rephrased. The resulting positive net RF is caused by the dominance of $\Delta F_{tir}$ over $\Delta F_{sol}$.

> *"An increase in IWC intensifies the cooling in the solar and the heating in the TIR. As both effects compete against each other and $\Delta F_{tir}$ dominates $\Delta F_{sol}$, the resulting net RE is a warming. An exception appears for largest IWC, where median $\Delta F_{net}$ is negative. Simultaneously, the increase in IWC causes an enhanced impact of the free parameters and associated uncertainties."*

**527: "Simultaneously, the TIR heating remains almost constant...": yes, because the cloud top temperature is fixed. The latter could also be subject to variations. In fact, brighter clouds often have larger vertical extend and are thus colder. I suggest to drop this "underlying cloud" study.**

The fixed temperature results from the set-up of the simulations. Here we selected $T_{cld}$ as the specified parameter and positioned the cloud depending on the atmospheric temperature profile. Alternatively, one could have fixed the altitude, e.g., 10 km, and vary $T_{cld}$. Fixing the cloud altitude would does not seem appropriate as the formation of clouds is primarily driven by temperature and the altitude varies depending on atmospheric profile and location. Similar to the statement from above, we intent to keep the underlying cloud study. The reason was given in one of the previous answers.

**528: infinite -> horizontally infinite**

The sentence was modified.

> *"[…] the cloud RE of homogeneous, horizontally infinite ice cloud [...]"*

---

## Author Comment (AC3)

**Reply to Reviewer #3**
(Referee comment on "Radiative effect by cirrus cloud and contrails – A comprehensive sensitivity study" by Kevin Wolf et al., EGUsphere, https://doi.org/10.5194/egusphere-2023-155-RC3, 2023)

We thank the Reviewer for the time she/he spent on the manuscript. The comments helped to improve the manuscript, but more importantly spurred us into repeating our calculations with (1) a completely revised libradtran configuration to ensure that we use state-of-the-art parametrization; and (2) much extended parameter ranges to be better representative of cirrus and contrails. The discussion in the manuscript has been revised to reflect the new calculations and analyses. In the following, the Reviewer's comments and the corresponding responses are listed. The page and line references given by the Reviewer relate to the manuscript in discussion. Numbers given from our side relate to the revised manuscript.

For better legibility, the Reviewer's comments are highlighted in **bold** and changes in the manuscript are in *italic*.
* * *
**This study presents a dataset of radiative transfer simulations with the goal to investigate the sensitivity of the radiative effect of cirrus and contrails. The sensitivity study comprises eight selected parameters: ice crystal effective radius, ice water content, solar zenith angle, surface albedo, liquid water cloud optical thickness of an underlying cloud, three ice crystal shapes, cirrus temperature, and surface temperature. The dataset which is submitted together with the manuscript consists of three netCDF files, one for each ice crystal shape. Results for plane-parallel radiative transfer simulations are provided as upward and downward irradiance for cloudy and clearsky scenes as well as the cloud radiative effect (CRE), integrated over the solar and thermal spectrum. While such a sensitivity study has the potential to provide interesting insights into the driving parameters on CRE of cirrus and the associated data set is useful as a reference, there are a number of major issues which have to be addressed before publication:**

**(A) The manuscript is missing a discussion of the results and comparison with previous studies which are mentioned in the introduction (Fu and Liou (1993), Yang et al. 2010, Zhang et al. 1999, Mitchell et al. 2011, and Schumann 2012). Are there new insights gained from the selected parameter space?**

We thank the Reviewer for providing these literature. During the revision of the manuscript the cited literature was consulted and compared to our results. We would like to direct the Reviewer to the diff file as the corrections have been made in multiple sections of the manuscript.

**(B) There are several major issues with the setup of the RT simulations which have to be addressed, especially since the data set is intended for public use:**

1. **Top of the atmosphere (TOA) is assumed here at 15 km (as stated e.g. in line 90 and Table 1) instead of the commonly used 120 km (Emde et al. 2016). All atmospheric profiles provided in libRadtran and used in this study are defined up to 120 km. The upward and downward irradiances computed in this study are therefore missing important contributions of molecular scattering and absorption. To allow comparison with other studies and make the data set useful for the community, irradiances should be computed at the standard TOA level.**

   We follow the suggestion of the Reviewer and set the uppermost level to 120 km. The simulations have been repeated and the manuscript has been revised accordingly. Please see the diff file.

2. **Ice cloud optical thickness values are provided for a reference wavelength of 640 nm. The standard reference wavelength, however is 550 nm. Similar as above, to allow comparison with other studies and make the data set useful for the community please use 550 nm as a reference wavelength.**

   We follow the suggestion of the Reviewer and provide the output at 550 nm wavelength. The simulations have been repeated and the manuscript has been revised. Please see the diff file.

3. **The study claims to use the "more recent ice crystal parameterizations" (line 61) but only droxtals were used from Yang et al. 2013, whereas Yang et al. 2000 was used for plates and rough aggregates. Yang et al. 2013 provides optical properties for plates and rough aggregate as well. Why not use the latest optical properties in a consistent way?**

   The Reviewer is right. For consistency and for the sake of using 'more recent ice crystal parameterizations', we have remade all simulations, now using the ice optical properties from Yang (2013).

4. **Furthermore, no explanation or discussion is provided why these specific habits were chosen. Why are e.g. columns or bullet rosettes not included? Please provide motivation to select "droxtal", "rough-aggregates" and "plates" and cite relevant literature that supports this choice as representative for cirrus, contrails, and contrail cirrus (e.g. Platnick et al. 2016, Forster et al. 2022, Järvinen et al. 2018).**

   The Reviewer highlights an important point. Item four in section 2.2 about the selected ice crystal shapes is greatly extended. Following the suggested literature it shows that rough aggregates are most commonly detected in cirrus clouds. The observations include LIDAR observations form satellite, aircraft in-situ observations, and ground-based observations. This is now mentioned in the text and supported by the suggested literature. Please see the diff file for the extended text.

5. **It is not explained why libRadtran's Fortran implementation of DISORT is used for the radiative transfer simulations instead of the faster and more robust C-version (Emde et al. 2016), when the goal is to use the "latest RT models" (line 62).**

   We thank the Reviewer for the helpful suggestion to use the DISORT solver. The solver has now been used to repeat all simulations.

6. **The results including the water cloud below the cirrus are potentially biased: "wc_modify tau set 20" in the input file will set the water cloud optical thickness to 20 at each wavelength which causes the liquid water content to vary across the spectrum. To achieve constant LWC, it has to be be scaled directly to an optical thickness of 20 at 550 nm wavelength.**

   We thank the Reviewer for this remark. All new simulations use 'wc_modify tau550 set xx' to scale the cloud optical thickness at 550 nm wavelength.

7. **The water cloud layer is fixed with cloud base at 3 km. This implies that the cloud layer is located at a different temperature for each of the 3 atmospheric profiles. As stated in the manuscript (line 174) this places the cloud even at temperatures below freezing for the subarctic winter profile. To be consistent, should the water cloud not rather be fixed at a certain temperature, the same way the altitude of the ice cloud was defined?**

   Within the subarctic winter profile all temperature values are below freezing. This implies that all potential clouds, positioned in this profile, will be below freezing and, in case of liquid clouds, contain super-cooled droplets. Nevertheless, clouds with super-cooled droplets at cloud top are frequently observed (70% of the clouds) in the arctic (e.g., Hogan 2004 and Hu 2010).
   In the simulations the liquid water cloud is positioned at a fixed cloud top altitude of 1.5 km and a geometric thickness of 0.5 km. In all three atmospheres (sub-arctic, mid latitude, and tropics) low-level clouds at this altitude occur frequently.
   Please see the diff file for the extended item 8 in section 2.2 of the manuscript that explains the positioning of the liquid water cloud.

8. **Information about the setup of the radiative transfer simulations is contradicting in several places in the manuscript, or missing:**
- **It is not explained how the surface temperature is set in the RT simulations. The stated temperatures of 273 K for afglsw and 313 K for afglus do not correspond to the surface level temperature of these atmospheric profiles as provided by Anderson et al. 1986.**

   We agree with the Reviewer. The old selection caused a discontinuity in the temperature profile at the interface between surface and atmosphere profile. The surface temperatures have been changed to agree with the lower most (0 km altitude) temperature of the atmosphere profiles.

- **Molecular absorption is stated to be Fu and Liou (1992, 1993) in Table 1, then the text states REPTRAN parameterization in "moderate" resolution (line 110), and the sample input file provided as a supplement uses REPTRAN in "coarse" resolution. Please double-check and explain the choice.**

   The REPTRAN resolution was double-checked. All new simulations have been run with a 'coarse' resolution and the manuscript has been changed accordingly.

   *The RT simulations consider molecular absorption using the 'coarse' resolution REPTRAN parameterization [...]*

- **In Table 1, and line 109 it is stated that the spectral solar irradiance according to Kurucz 1992 is used. The data provided with libRadtran has a spectral resolution of 1 nm, but the sample input file refers to a version with 5 nm resolution. How was that obtained and why did the authors choose a coarser resolution?**

  Previously, the solar irradiance from Kurucz 1992 was interpolated from the original 1 nm resolution to 5 nm resolution. In the new simulations the original 1 nm file is used.

  **(C) A clear statement of the intended use of the dataset together with assumptions made for the radiative transfer simulations and their impact on the accuracy of the results is missing. The abstract (line 21/22) states: "The data set […] can be used to compute the radiative effect of cirrus clouds, contrails, and contrail cirrus instead of full radiative transfer calculations." This is a very general statement and it is not clear what potential use cases could be. Although it is very useful to publish the results together with the paper, potential users of the data set would need more guidance: Please provide more details how the data set should be used, limitations, accuracy, possible questions that could be answered.**

  1. **Important information is missing about assumptions used for the radiative transfer simulations which have important implications for potential use cases: Plane-parallel RT instead of 3D RT, assuming TOA at 15 km, assuming randomly oriented ice crystals, parameterization of ice crystal optical properties which assumes a coupling of crystal size and aspect ratio, constant geometric thickness of the cirrus of 0.2 km, etc.**

     These assumptions are provided more prominently in item 4 in section 2.2 in the manuscript to ensure correct usage of the published data and to raise awareness of potential uncertainties.

  2. **Especially for contrails and contrail-cirrus, but also for cirrus radiative 3D effects have been shown to be non-negligible (e.g. Gounou and Hogan 2007, Kalesse 2009, Forster et al. 2011). If the presented results should be applicable to contrails the bias due to neglecting these 3D effects has to be quantified.**

     The Reviewer highlights an important point. Considering further Reviewer comments, we added a dedicated section that mentions and partly discusses the differences between 1D and 3D simulations and the associated uncertainties. However, a quantification of the differences is beyond the scope of this study. To raise awareness on that potential uncertainty, we provide numbers and citations from the suggested literature: Gounou and Hogan (2007) as well as Forster et al. (2011).

     However, we note that aged and spread contrails might be approximated as homogeneous thin plane-like clouds, which justifies the use of 1D simulations (Minnis et al., 1999).

  **More detailed comments:**

1. **Abstract line 18: Why is TIR influenced more by ice crystal shape than effective radius? In line 298 it is stated that crystal size has a stronger impact than shape. Please explain in the text.**

It is stated in the text that $r_{eff}$ and IWC are the dominating factors in the solar and TIR wavelength range. For TIR the other parameters are given in descending order.

2. **Abstract line 19: "Net RE is controlled by the surface albedo, the solar zenith angle, and the surface albedo in decreasing importance". Surface albedo is mentioned twice, please correct.**

   The Reviewer is right and the sentence has been corrected.

   > *" The combined net RE is controlled by $α_{srf}$, $θ$, and $T_{srf}$ , sorted in decreasing importance."*

3. **Line 69-72: "A comprehensive study of cirrus radiative effects was conducted by Schumann (2012), who aimed to derive an approximate model to estimate the cloud RE. While those studies are valuable, none of them presents a comprehensive sensitivity study across all relevant cloud and environmental input parameters. Therefore, we present a study that separates the effect of eight selected parameters on the cirrus RE."**
   **This is contradictory: none of the previous studies is "comprehensive", but the present study focuses on "eight selected parameters". Are the eight selected parameters of the present study enough to make it "comprehensive"? Should not the driving question be: How many and which parameters are necessary to investigate the main question / support the main statement?**

   The Reviewer is right. Claiming to provide a 'comprehensive' study is misleading. Following the suggestion of the Reviewer we rephrased the objective of this study and removed 'comprehensive' from the title and the manuscript. Nevertheless, the main objective remains, which is to identify the main drivers of the cirrus RE among the eight selected parameters.

   > *"Multiple studies that aimed to investigate the impact of a certain parameter on cloud RE have been performed in the past. Fu and Liou (1993) as well as Yang et al. (2010) focused on the effects of the selected ice crystal habit and ice water path. The effect of the ice crystal size distribution was analyzed, for example, by Zhang et al. (1999) or Mitchell et al. (2011). A comprehensive study of cirrus radiative effects was conducted by Schumann (2012), who aimed to derive a parameterization to estimate the cloud RE. While those studies are valuable, none of them investigate the effect of multiple factors, like relevant cloud and environmental input parameters. These studies have identified parameters that affect cirrus RE, but all these parameters need to be considered together, including both cloud and environmental parameters. This article is intended as a parametric sensitivity study that aims to compare the effects of major parameters. Furthermore, we identify the driving parameters of RE by sampling the input parameter range, restricted to values that are typically associated with ice clouds. Finally, we provide an open-access data set, which allows the user to extract cloud REs for user-specific combinations of the input parameters. The data set might be coupled with cloud microphysical models, e.g, the Contrail Cirrus Prediction Tool (CoCiP) from Schumann (2012), to estimate the RE of the simulated contrails"*

4. **Line 85: Please add the equation for DeltaF_net before defining DeltaF_sol and DeltaF_tir**

The equation for $\Delta F_{net}$ is now given before defining $\Delta F_{sol}$ and $\Delta F_{tir}$. Please section see section 2.1 in the manuscript or the diff file.

5. **Line 95: "The surface albedo is kept constant in this study". Which value is chosen for the solar spectrum?**

The surface albedo in the solar is set to values between 0 and 1, which are specified and discussed later in the paper. Therefore, the sentence has been moved to subsection "2.2 Radiative transfer simulation set-up". That section now reads:

> *"The Earth's surface albedo, $\alpha_{srf}$ ranges from 0 to 1, which represents the full possible range. In general, $\alpha_{srf}$ varies spectrally but here is kept constant for all solar wavelength. It is varied between 0 and 1 to include surface conditions ranging from open ocean to full sea ice or snow (Baldridge et al., 2009; Gardner and Sharp, 2010; Meerdink et al., 2019; Gueymard et al., 2019). Values of αsrf are given in Table 4. In the TIR wavelength range $\alpha_{srf}$ is assumed to be 0, which leads to an emissivity $\varepsilon = 1$ with the Earth's surface thus acting as a blackbody (Wilber, 1999)."*

6. **Line 102: "libRadtran was run as one-dimensional (1D) RT solver…" -> better: "The 1D RT solver DISORT, which is part of libRadtran, assuming horizontally uniform clouds".**

We thank the Reviewer for this suggestion and we modified the sentence accordingly.

> *"The radiative transfer solver DISORT (Buras et al., 2011) allows to select 2N -number of streams to be used in the radiative transfer simulations. [...]"*

7. **Line 119: Why would tropical and desert atmospheric profiles be interchangeable here? The different water vapor profiles affect the thermal RE as mentioned in the subsequent sentence.**

Tropical and desert atmospheric profile are not interchangeably. The amount of water vapor, especially in the lower atmosphere h< 6 km differs. Here we refereed to the surface temperature only and not to the vertical profile. Nevertheless, we follow the suggestion of the Reviewer and remove 'desert' from the sentence. This is in line with the adjusted surface temperature as the upper bound of surface temperature was changed from 16°C to 27°C (due Reviewer comment directly below). A surface temperature of 27°C are representative for tropical regions but not necessary for desert regions, which can have much higher surface temperatures. 27°C are selected as a compromise to match the lowermost temperature of the atmosphere profile (please see the comment below). In addition, we added a sensitivity study to estimate the effect of variations in the relative humidity profile. A dedicates section can be found in section 3.5. Please see the adjusted sections in the diff file.

8. **Line 121: Please double-check the surface temperatures for the subarctic winter and tropical profiles. Surface temperatures for subarctic winter is 257.2 K and 299.7 K for tropical. How is the surface temperature "set" to -40, 0, 40 degC?**

The Reviewer is right. The previously selected temperature in the atmosphere profile and the surface temperature caused a discontinuity. In the new simulations, the surface temperatures are set to the temperature of the lower most value of the selected atmosphere profile.

9. **Line 143: "Our simulations range from 5 to 45 μm for all three shapes and, therefore, focus on young contrails and cirrus." If so, aged contrails and contrail cirrus should not be mentioned in the abstract and conclusion.**

The Reviewer is right. With the new setup and repeated simulations ice particle size ranges from 5 to 85 μm, which also includes more mature contrails and cirrus clouds. (Krämer, A microphysics guide to cirrus – Part 2: Climatologies of clouds and humidity from observations, 2020, Atmos. Chem. Phys. , 20, 12569-12608, 2020).

10. **Table 3: Range does not add information here, just provide actual values. Add "total number" as last column label.**

The column 'range' has been removed and the last column is labeled 'total number of simulations'. Please see the diff file for the modifications.

11. **Line 185: "because, as 3D effects are neglected" -> "as radiative 3D effects are neglected". This is the first time 3D effects are mentioned, but this information should appear more prominently. Please cite relevant literature and add more discussion on possible biases introduced by the plane-parallel assumption and neglecting 3D RT in this study.**

Following this comment and comments from the other Reviewers, we provide a paragraph in sections "1. introduction" and "2.2 Radiative transfer simulation set-up". Please also see the response to the general comment number 2 of Reviewer 3.

12. **Results Fig. 1: it should be noted that these results do not rely on RT simulations but show basic dependencies between microphysical and optical parameters.**

The ice crystal number concentration (Fig1a) was calculated with equation 13, assuming spherical ice crystals (approximation for droxtals) and assuming a mono-disperse particle size distribution. The cloud optical thickness used in Fig. 1 b,c,d is obtained from libRadtran simulations (verbose file) using ice optical properties of droxtals. It is now detailed how Fig. 1 is created in the text and the caption.

*"We first provide an overview of how $r_{eff}$ and IWC determine the cloud optical and microphysical properties. Figure 1a–d illustrates the dependence of $N_{ice}$ and $\tau_{ice}$ as a function of $r_{eff}$ and IWC. Nice is approximated by Eq. 14, assuming droxtals (almost spherical ice crystals), a mono-disperse particle size distribution, and a cloud geometric thickness dz of 1000 m. The ice cloud optical thickness $\tau_{ice}$ at 550 nm wavelength, given in Fig. 1b–d, is directly calculated by libRadtran using optical properties from droxtals."*

13.     **Line 225: "Going beyond these dependencies…" The sensitivities discussed in the preceding paragraph do not use RT simulations. Now switch to RT results? This should be separated more clearly in the text.**

The sentence in this section was rephrased to be more clear in this aspect. Please also see the previous comment.

14.     **Fig. 1c, d: please complete legend information with "r_eff" (1c) and "IWC" (1d)**

A title was added to both legends.

15.     **Line 245: why are the parameters for the reference cloud chosen from extreme values of the parameter space? Wouldn't it be more intuitive to select mean/median values?**

Similar to Meerkötter et al. (1999), we selected the extreme values to mark either end of the simulated parameter range. Using mean values would not allow to explicitly mark the upper or lower boundary, and to investigated the effect of spanning the full range of a given parameter.

16.     **Please provide a reference from literature which states a representative cirrus optical thickness of 0.18 at 640 nm?**

Iwabuchi (2012) used CALIPSO Lidar observations and determined a mean COT of contrails of 0.19 (532nm). Nevertheless, thicker contrails may exist. The reference was added to the section. (Iwabuchi, H. / Yang, P. Liou, K. N. / Minnis, P.; Physical and optical properties of persistent contrails: Climatology and interpretation, 2012, J. Geophys. Res. Atmos. , Vol. 117, No. D6)

> *"[…] this leads to a $\tau_{ice}$ of 0.46 at 550 nm wavelength, which is representative for contrails and young cirrus (Iwabuchi et al., 2012). [...]"*

17.     **Which crystal shape is assumed for the reference cloud?**

This information was added.

> *"The reference cloud is assumed to consist of rough-aggregates."*

18.     **In Fig. 2 it looks like reff=5 um is used for the reference cloud, not 45 um.**

This was adjusted. Now a cloud with 85 µm is used.

19.     **Figure 2:**
- **The scale and grid lines of the y-axis should be comparable between the 3 subplots.**

Grid lines have different spacing to maintain clarity in the SW plot and to provide sufficient guidelines in the TIR and net plot. For better comparability we changed gridlines to an equal spacing.
However, we keep the different *y* axis otherwise the bars in the net become too small for differences among the parameters to be legible.

- **Caption: The parameter for the reference case provided here do not match the description in the text.**

  The caption and the text have been homogenized.

- **Selecting mean/median values of the parameter space would place the star closer to the mean RE, similar to the IWC case.**

  We selected either end of the parameter space to clearly show how ΔF varies, when the one of the parameters is varies to the other end of the parameter space.

- **Is a box plot representative for the 3 distinct ice crystal shape values?**

  Following this question, we separated the data of the three shapes and present them individually. In that way the discrete differences from the shape effect become clearer.

20.     **Line 249: "For the all Sun geometries…" Please double-check sentence.**

  "The" has been removed from the sentence.

    *"For all Sun geometries[…] "*

21.     **Line 274: Which values for the surface albedo were selected to investigate the sensitivity of the RE on T_srf, T_ic and tau_wc? The results should be different for alpha=0, and 1.**

  In the introduction of the reference cloud, a surface albedo value of 1 was given. To be clearer, we added a sentence at the beginning of this paragraph that explicitly states the surface albedo

    *"The influence of a varying surface temperature $T_{srf}$ or cirrus temperature $T_{cld,ice}$ (related to cloud base altitude), are investigated for a cloud scenario with a solar surface albedo $α_{srf,sol}$ set to 0. [...]"*

22.     **3.1 Sensitivity on ice crystal shape: When comparing the effect of ice crystal effective radius vs. crystal shape on the cirrus RE, it is important to mention that size and aspect ratio are coupled in the optical property parameterizations by Yang et al. 2000 and 2013. Please add this to the discussion.**

  Section 3.1 was extended an it is now mentioned that the maximum dimension of an ice crystal and the aspect ratio are coupled.

    *"[…] Furthermore, the ice optical properties by Yang et al. (2010, 2013), which are used for the RT simulations in the present paper, based on a coupling of the maximum diameter of the ice crystal and the aspect ratio, with the later one being different for each particle shape [...]"*

23.    **Figure C1: why not show the phase function for the ice crystal shapes and effective radii which are actually used?**

We follow the advice of the Reviewer and plot the phase functions for $r_{eff}$ of 5, 25,55, and 85 µm, which are the newly selected values for the simulations. Please see the revised diff file.

**Literature:**

- **Kalesse, H., 2009. *Influence of ice crystal habit and cirrus spatial inhomogeneities on the retrieval of cirrus optical thickness and effective radius* (Doctoral dissertation, Mainz, Univ., Diss., 2010).**
- **Gounou, A. and Hogan, R.J., 2007. A sensitivity study of the effect of horizontal photon transport on the radiative forcing of contrails. *Journal of the atmospheric sciences*, *64*(5), pp.1706-1716.**
- **Forster, L., Emde, C., Unterstrasser, S., and Mayer, B. 2012. Effects of three-dimensional photon transport on the radiative forcing of realistic contrails. *Journal of the atmospheric sciences*, *69*(7), pp.2243-2255.**
- **Platnick, S., Meyer, K.G., King, M.D., Wind, G., Amarasinghe, N., Marchant, B., Arnold, G.T., Zhang, Z., Hubanks, P.A., Holz, R.E. and Yang, P., 2016. The MODIS cloud optical and microphysical products: Collection 6 updates and examples from Terra and Aqua. *IEEE Transactions on Geoscience and Remote Sensing*, *55*(1), pp.502-525.**
- **Forster, L. and Mayer, B., 2022. Ice crystal characterization in cirrus clouds III: retrieval of ice crystal shape and roughness from observations of halo displays. *Atmospheric Chemistry and Physics*, *22*(23), pp.15179-15205.**
- **Järvinen, E., Jourdan, O., Neubauer, D., Yao, B., Liu, C., Andreae, M.O., Lohmann, U., Wendisch, M., McFarquhar, G.M., Leisner, T. and Schnaiter, M., 2018. Additional global climate cooling by clouds due to ice crystal complexity. *Atmospheric Chemistry and Physics*, *18*(21), pp.15767-15781.**

---

## Author Comment (AC4)

**Reply to Community Comment #1** (Dennis Piontek and Ulrich Schumann) (Community comment on "Radiative effect by cirrus cloud and contrails – A comprehensive sensitivity study" by Kevin Wolf et al., EGUsphere, https://doi.org/10.5194/egusphere-2023-155-CC1, 2023)

We thank Dennis Piontek and Ulrich Schumann for the time they spent on the manuscript. The comments helped to improve the manuscript, but more importantly spurred us into repeating our calculations with (1) a completely revised libradtran configuration to ensure that we use state-of-the-art parametrization; and (2) much extended parameter ranges to be better representative of cirrus and contrails. The discussion in the manuscript has been revised to reflect the new calculations and analyses. In the following, the Reviewer's comments and the corresponding responses are listed. The page and line references given by the Reviewer relate to the manuscript in discussion. Numbers given from our side relate to the revised manuscript.

For better legibility, the Community Comments are highlighted in bold and changes in the manuscript are in italic.
* * *
**The study of the radiative forcing of cirrus and contrails is an important task. In particular the climate impact of contrails gets significant attention in the past years as the avoidance of contrails by next- generation aircraft engines, the rerouting of flights, and the use of sustainable aviation fuels promises to be an easily achievable climate change mitigation strategy. In that sense, we want to applaud the authors for contributing to this endeavor.**

**The authors present an ambitious study to evaluate the radiative forcing due to ice clouds by performing a large number of radiative transfer calculations (94,000) for different atmospheres, liquid water and ice cloud configurations (I.e., different optical depths and heights), ice crystal sizes and shapes, surface temperatures and albedos, as well as solar zenith angles. The radiative impacts in the thermal infrared and the solar spectrum are quantified. For the calculations, the established radiative transfer code libRadtran (Mayer & Kylling, 2005) was used.**

**As the authors pointed out, various studies already investigated the cloud radiative forcing with different foci. However, we agree to the third reviewer: the statement in lines 70-71 (most "comprehensive sensitivity study") needs further work to become fully justified. One comparable but missing study is "A Parametric Radiative Forcing Model for Contrail Cirrus" by Schumann et al. (2012a). In this study, libRadtran was used as well to simulate the thermal and solar cloud radiative forcing of contrails, covering different surface and atmospheric conditions, solar zenith angles, seven different ice particle shapes and effective particle radii up to 45 μm, different liquid and ice water configurations. In total, 36,576 calculations were performed. Based on this dataset, approximations of the long- and shortwave radiative forcing due to contrails were derived. The study also shows sensitivity studies with respect to various quantities (e.g., contrail optical depth, solar zenith angle, effective albedo).**

Due to the strong similarity of the simulated datasets of Wolf et al. And Schumann et al., it appears mandatory to perform a direct comparison. Thus, we compared in a quick first study the calculations of Wolf et al. With the parameterizations developed by Schumann et al. Those are implemented in the Python package pycontrails (https://py.contrails.earth) which includes (among others) the "Contrail Cirrus Prediction Tool" (CoCiP, Schumann, 2012b).

The approximation of the longwave radiative forcing needs 5 inputs, which we estimated by data from Wolf et al. As follows:

Table (see original posting)

The ice crystal habits are considered separately, as the habit is given as an additional parameter to the radiative forcing functions of pycontrails (here, it is mainly used to convert r_vol back to r_eff internally; the parameterization of Schumann et al., 2012a, relies solely on r_eff and is independent of the ice crystal shape). We considered rough aggregates and droxtals. Wolf et al. Also performed calculations for plates. However, the approximate conversion between r_eff and r_vol is non-linear (Schumann et al., 2011); thus, we did not consider plates for the moment.

Note that the cirrus optical depths provided by Wolf et al. And used in the approximation of Schumann et al. (2012a) are for different wavelengths (640 and 550 nm, respectively). However, we assume that the differences in the ice optical properties are in the order of few percent (Lynch & Mazuk, 2001) and, therefore, negligible.

Unfortunately, also the definitions of "top of atmosphere" differ as Wolf et al. Define "top of atmosphere (TOA) at 15 km" height. As a result, the upward thermal infrared irradiance of Wolf et al. Can only be considered as an approximation of the outgoing longwave radiation at top of atmosphere in the sense of Schumann et al. (2012a). This is also visible when considering the downward thermal infrared irradiance of Wolf et al., which is not zero but varies between roughly 7 and 10 W/m². The difference in the definition of top of atmosphere has also an impact on the inputs for the solar direct radiation and the reflected solar radiation, as well as the resulting cloud radiative forcings in the long- and shortwave spectrum.

Nevertheless, we find that the results of Wolf et al. And the approximations of Schumann et al. (2012a) are in reasonable agreement (see plots below), with Pearson correlation coefficients of 0.979 and higher. The longwave radiative forcing based on Schumann et al. (2012a) is slightly smaller than the results of Wolf et al. Towards the lower end of considered thermal infrared radiative forcings. For the shortwave radiative forcing, we find a larger scatter between both results.

Although these results represent only a first quick look into the matter and further investigations might be necessary, the comparison already seems to show that the calculations presented by Wolf et al. (and, thus, the underlying input datasets and assumptions) agree with the work presented by Schumann et al. (2012a).

We thank both authors for this interesting companion. The plots they provided indicate that the parameteriziations and the simulated RE agree, which increases the confidence in our and their results. The increasing differences in ΔF towards smaller cloud optical thickness are explained by the increasing contribution of ambient conditions and the decreasing contribution

of the ice cloud itself. Consequently, parameters like the surface albedo or humidity profile, become more influential and important, and lead to deviations and the scattering. However, a detailed comparison between the simulations and the parameterization is beyond the scope of the presented study. A dedicated study, which addresses these differences in detail might be a useful contribution to the literature.

**Further major comments to the manuscript:**

**We appreciate that the results in Wolf et al. Are close to the results in Schumann et al. (2012), but we miss a discussion of a) the variable humidity: It is well known that the relative humidity over ice is often close to 100 % near cirrus and contrails (see Li et al., 2023). But, what is the relative humidity in your profiles?**

We have added plots of the temperature and relative humidity profiles used in the calculations to the appendix.

In addition, we have performed an analysis of the sensitivity of our results with respect to the RH profiles. Anderson (1986) states that standard profiles are subject to variations between 10 and 30%. So, we varied the original RH profiles by +/- 20% and repeated the simulations for a sub-set of the total range of simulations. The modified profiles are used to a) account for the potential variation in the profiles and b) to estimate the impact of different RH profiles on simulated solar, TIR, and net radiative effect. Variations in RH did not show an impact on $F_{sol}$ (+/- 0.4%) but modify $F_{tir}$ (+/- 4.1%) and $F_{net}$ (up to +/- 8),  particularly the relative values of $F_{net}$. We did not modify the RH profiles around the cloud, but this could be looked at in a follow-up study.

**b) any other absorbing gases or species (O3, CO2, aerosols)?**

Temperature and humidity profiles from libRadtran (Emde, 2016) are used, which base on the atmospheric profiles from Anderson, 1986. As already mentioned in the manuscript, molecular absorption is included but now the individual gases that contribute to the absorption are explicitly stated in the manuscript. The atmospheric composition, i.e., concentration of the gases, is also taken from the atmosphere profiles of Anderson, 1986. Absorption by aerosol is not considered in our simulations.

**Discussion of importance of large solar zenith angle SZA > 70°: The shortwave radiative forcing reaches a maximum near or above that SZA value, see Figs. 7 and 8 in Schumann et al. (2012), Fig. 12 in Markowicz & Witek (2011), Fig. 1 in Myhre & Stordal (2001); and hence this parameter range is important at sun dawn in early morning/late evening (Meerkötter et al., 1999).**

**The problem with high SZA is, however, that clouds in general, and contrail cirrus clouds in particular, can only very roughly be approximated as horizontally homogenous, in particular when the sun is low over the horizon. We miss a study on the 3d-effects of contrails (depending among others on SZA, azimuth of contrail-line direction relative to the sun, on the width/thickness ratio of the contrails lines (Forster et al., 2014), besides the 3d clouds in the contrail neighborhood), besides the effects of**

**non-spherical Earth geometry and solar radiation refraction in the atmosphere at high SZA.**

The Reviewers raise an important point. Following the suggestion and the provided literature, we extended the range of simulated SZA to 85º. Based on these additional simulations, we added a paragraph to the manuscript, in which we discusses the sensitivity of solar ΔF on SZA and link the discussion with the provided literature.

We chose a maximum SZA of 85º because the radiative transfer solver "DISORT" treats atmospheric layers as plane-parallel. Results for θ > 85º are likely nonphysical and have to be treated with caution (Stamnes, 2000; Buras, 2011). In addition, the biases between 1D and 3D simulations increase with SZA and are now highlights in the introduction of the manuscript.

**With respect to your Appendix B: In Schumann et al. (2012), Bernhard Mayer noted: "the irradiances are computed using the discrete ordinate solver by Stamnes et al. (1998), version 2.0, with six streams, which allows accurate simulations of irradiances." We wonder why you need 16 streams and cannot calculate at high SZA? Do you want to say that the former results are significantly inaccurate for methodological reasons? We expect small differences between 6 and 16 streams.**

and

**The test example assumes a surface albedo of one and liquid water clouds below the ice clouds. Hence the solar forcing is small in this case. Is this the best test case?**

The choice of 16 streams comes from a compromise between computational time accuracy. We found a small, worthwhile gain in accuracy when increasing the number of streams from 8 and 16. Adding more streams provides only negligible additional accuracy. The presented plot D1 in the appendix shows only one exemplary cloud scenario.

The case was selected to have a significant fraction of upward irradiance (contribution from the surface) plus adding the interaction of a liquid water cloud. The aim was to create a profile with cloud-surface-cloud-radiation interaction. The selected example might not be the ideal case and, therefore, the conservative approach with 16 streams is used.

The manuscript has not been changed in this regard.

**Why do you use the older Fortran version of libRadtran? The more stable C-Version is available since 2010.**

This is an important comment. We switched to the DISORT solver and repeated the simulations.

**Another important issue, which is so far only approximately covered, is the effect of overlapping contrail cirrus clouds. We found (see Schumann, Poll et al., 2021) that Europe is covered frequently by very many contrails which get wide compared to the lateral distances to other contrails so that they partially overlap each other and so that contrails forming above or below the first contrails experience a changed radiation field with different effective OLR/RSR values. We used a rough approximation to**

**account for this effect and found that it changes the computed net RF by a factor of order two over Central Europe, depending on air traffic density and humidity.**

In a recent study by Sanz-Morère et al. (2021) it is reported that contrail-contrail radiative effects can likely be neglected in estimates of the radiative effect. Furthermore, adding a second ice cloud / contrail to the simulations would add another dimension in the multi-dimensional simulation set-up. Here we wanted to focus on the basic dependencies.

> *"The parameter selection of this sensitivity study was motivated by Meerkötter et al. (1999), which was supported by previous studies, for example Fu and Liou (1993), Zhang et al. (1999), Yang et al. (2010), or Mitchell et al. (2011). Schumann et al. (2012) then parameterized the effects of the parameters identified by Meerkötter et al. (1999) on the cloud RE. Additional influences like aerosol layers, more complex surface albedo, or multiple overlapping cirrus and contrails have not been investigated here and represent additional degrees of freedom. For example, previous studies found that aerosols have only a minor influence on contrail RE (Meerkötter et al., 1999) and Sanz-Morère et al. (2021) reported that the impact of overlap between contrails on their RE is negligible. Nevertheless, the present study covers the parameters that most directly affect cirrus RE."*

**Line 192, Eq. 11: Why do you need the factor β? The r_vol is defined with β = 1 for arbitrary habits, see Schumann et al. (2011), Eq. 18, at least for fixed ice density ρice . More important (besides ρice for porous crystals), is the ratio C=r_vol/r_eff, see Eq. 1 in the same paper. Do your results change and how much if you use β = 1 consistently in your study?**

Due to this and other Reviewer comments, the set of equations have been revised in the updated manuscript. A more accurate mass-size relationship if provided in the manuscript following Mitchel (2002). We direct the authors to the diff file and see section 2.2.

**Minor comments to the manuscript:**

**Why do you use the term "Radiative Effect, RE"? We think that the term "Radiative Forcing RF" is more often used. What is the difference between RE and RF?**

Although the terms radiative forcing and radiative effect are often used interchangeably in the literature, they have different meanings. Cloud radiative effect is the contribution of clouds to the Earth's radiative budget. Radiative forcing means a change in radiative effect since pre-industrial conditions. In the case of contrails, which were not present in the atmosphere in pre-industrial conditions, radiative effect and forcing are equal. But that is not true in general, which is why we use the term "effect".

**Line 32: We do not understand why you cite Jensen et al. (1994) here: "contrails are short lived and can persist...". Jensen et al. Discuss tropical cirrus, not contrails. Here the paper by Schumann (1996), even if not the first (see also Schumann, 1994, and Busen & Schumann, 1995) is often cited as the most comprehensive introduction of**

**contrails in literature at least until that time (see also Schumann & Heymsfield, 2017a, besides Kärcher, 2018).**

Agreed. The citation from Jensen (1994) was removed and references from Schumann (1994), Schumann (2017), and Kärcher (2018) were added.

**Line 35: Regarding the importance of cirrus cloud cover and contrails over Europe, you may also refer to Schumann, Penner et al. (2015) and Schumann, Bugliaro et al. (2021).**

This is correct and we added these two references to the text in line 35.

**Line 36: The fact that shortwave radiative forcing is mostly negative is well known. It should be mentioned that it can be positive for high surface albedo and high absorption in the atmosphere between ground and cirrus cloud as discussed in Meerkötter et al. (1999), page 1089, right column. See also Myhre & Stordal (2001), Fig. 1 (but published without explicit explanation).**

This is an important point. Nevertheless, the sentence is meant as a general introduction here with the emphasis on 'most of the cases'. Nevertheless, we value the suggestion and include the two citations later in the manuscript, where the influence of a high surface on the cirrus / contrail radiative effect is discussed.

**Line 137: Presumably the most comprehensive collection of aircraft in-situ and remote sensing measurements of contrail properties can be found in Schumann, Baumann et al. (2017b) and in the therein described open-access contrail library "COLI"; they cover not only young but also the more important aged contrails (partially exceeding 10,000 s).**

We thank the Reviewers for providing this citation. It was added to the manuscript to provide guidance for the interested reader.

**Line 158, Eq. 7 to 9: Very similar equations can be found in Schumann et al (2011).**

**We find it strange that you cite Meerkötter et al. (1999) in the figure caption of Fig. 2, but do not discuss similarities or disagreements in the content in the text. In fact, we still have to identify any basic new information in your discussion of Fig. 2.**

We adopted the excellent Figure design of Meerkötter et al. (1999) for Figure 2 to provide a good introduction for the more detailed investigation of the individual parameters. The intention is not to compare to their results. We included a paragraph that describes the intention behind the Figure in the manuscript.

> *"The presented analysis of solar, TIR, and net ΔF sensitivity on the selected input parameters generally agrees with the results from Meerkötter et al. (1999). We found differences in the importance of the parameters, which are explained by the fact that*

*our simulations span a larger and different parameter range, for example in $r_{eff}$ and $T_{srf}$. In addition, the sensitivity analysis in Fig. 2 is sensitive to the selection of the reference cloud."*

**The discussion of $r_{eff}$ and IWC as the most important parameter is incomplete and partially misleading (at many places and in particular in section 3.3 and in the summary, line 499). Physically, the most important parameter is the optical depth τ of the contrail cirrus, which is, among others, a function of r_eff, IWC and cloud geometrical thickness D. The r_eff is a secondary factor besides crystal habit etc. Of course, IWC, r_eff, D and crystal habits are important per se and possibly easier to measure while models might primarily compute the IWC and then estimate crystal habit and optical extinction βext for given IWC and temperature (Heymsfield et al., 2014), but τ ~ βext D, by definition, is the parameter which characterizes the impact of a cloud layer on radiation transfer.**

We partly agree with this comment. In our opinion, clouds can be regarded from two different perspectives: microphysical properties and optical properties. In this paper we follow the microphysical perspective, based on properties like ice water content and the ice particle size distribution / $r_{eff}$. As the comment states, cloud optical thickness is then a function of IWC, $r_{eff}$, cloud geometric thickness, and particle shape.

**The discussion of the importance of the surface temperature is misleading. It is not the surface temperature that is important but the effective brightness temperature of the atmosphere below the contrail cirrus, which in fact depends not only on the surface temperature but also on water vapor and other IR absorber profiles and low level clouds, besides spectral averaging. It was exactly this reason why Schumann et al. (2012a) parameterized the longwave radiative forcing not as a function of surface temperature (as also done by Corti & Peter, 2009), but as a function of OLR without contrail cirrus.**

We acknowledge the fact that the surface temperature does not alone determines the forcing of a cirrus but the entire atmosphere between surface and the cirrus as a whole. However, we use the surface temperature as a proxy for a certain temperature- and humidity profile to represent three different regions on the Earth. In the revised version of the manuscript, particularly in Section 3.5 and Appendix B, we better highlight the coupling of surface temperature and related atmosphere profiles of temperature and humidity.

**In summary, we highly appreciate that this study was performed and that we got access to the data, since this gives us the chance to test our parameterizations, but the paper needs considerable extensions and improvements before it can be published as a "comprehensive" study.**

We would like to answer this comment similar to Reviewer 3.
Claiming to provide a 'comprehensive' study is misleading. Following the suggestion of the Reviewer we rephrased the objective of this study and removed 'comprehensive' from the title

and the manuscript. Nevertheless, the main objective remains, which is to identify the main drivers of the cirrus RE among the eight selected parameters.

The selection of the parameters primarily based on the study performed by Meerkötter et al. (1999), which was supported, e.g., by Fu and Liou (1993) as well as Yang et al. (2010), who focused on the effects ce crystal habit and the ice water path. The effect of the ice crystal size distribution was analyzed, for example, by Zhang et al. (1999) or Mitchell et al. (2011).

Later on, Schumann et al (2012) parameterized the cloud radiative effect in dependence of the parameters identified by Meerkötter et al. (1999). We take a slightly different approach compared to Schumann et al (2012) and regard the cloud radiative effect of clouds from a microphysical perspective instead of an optical perspective. In addition, we provide an update of the calculations from Meerkötter et al. (1999) by using up-to-date radiative transfer models in combination with the latest cloud optical properties.

Furthermore, we strive to identify the driving parameters of RE by sampling the input parameter range, restricted to values that are typically associated with ice clouds. Finally, we attempt to provide an open-access data set, which allows the user to extract cloud REs for user-specific combinations of the input parameters. The data set might be coupled with cloud microphysical models, e.g, the Contrail Cirrus Prediction Tool (CoCiP) from Schumann (2012), to estimate the cloud radiative effect of the simulated contrails.

---

## Referee Report (RR1)

ACP Second Review (preprint on EGUsphere at:
https://egusphere.copernicus.org/preprints/2023/egusphere-2023-155/ )

Title: Radiative effect by cirrus cloud and contrails – A comprehensive sensitivity study
Author(s): Kevin Wolf, Nicolas Bellouin, and Olivier Boucher
MS No.: egusphere-2023-155
MS type: Research article
Iteration: Second submission

**GENERAL COMMENTS:**

The manuscript has been greatly improved and, in general, the authors have done a great job in response to the review comments I have made. However, they mention Section 2.4 titled "Approximation of radiative transfer in the thermal-infrared", which somehow was not included in the revised version of this manuscript. This section is critical since it describes how radiation transfer is treated in the thermal infrared (TIR) spectrum; it is needed to properly understand the TIR results of this sensitivity study. This manuscript should not be published without it.

**SPECIFIC COMMENTS:**

1. On comment #7 from 1st review:
Equation 13: Is this equation used in libRadtran? If not, what is the point in mentioning it? Cloud property input to libRadtran consists of IWC and re, suggesting the zero-scattering approximation might be used for TIR hemispheric fluxes: $\varepsilon = 1 - \exp(-5\,\tau_{abs}/3)$ where $\varepsilon$ is cloud emissivity and $\tau_{abs}$ is the cloud absorption optical depth. Please indicate whether $\varepsilon$ is calculated in libRadtran, and how it is calculated if applicable.

Author response: The DISORT solver in libradtran (Buras et al 2011) calculates scattering in the TIR on basis of the bulk-scattering properties of ice crystals, analog to the solar wavelength range. Thus, the zero-scattering approximation is not used in the simulations. **Equation 13** was added to the manuscript to provide guidance for the reader. To avoid misinterpretation the equation **is brought into context and is expanded to section "2.4 Approximation of radiative transfer in the thermal-infrared"**, to incorporate suggestions from other Reviewers.

Referee comment for 2nd review: The author response above is puzzling since the referee is finding no section 2.4 titled "Approximation of radiative transfer in the thermal-infrared" in the revised manuscript nor in the track-changes version (the diff file) of the manuscript. In the current revised manuscript, there is no discussion of how RT in the thermal infrared (TIR) is dealt with, which is critical for a RT sensitivity study presenting results in both the solar and TIR. Since the authors mention Sect. 2.4 in their response having the title "Approximation of

radiative transfer in the thermal-infrared", it appears that this section was mistakenly omitted from the manuscript. The manuscript should not be published without this section.

2. On comment #8 from 1$^{st}$ review:
Lines 209 – 213 and Eq. 14: Eqn. (14) appears flawed since, in principle, there should be an emissivity term ($\varepsilon$) for both the surface and the ice cloud. But since typically $\varepsilon \approx 1$ at the surface, does $\varepsilon$ in (14) correspond only to the ice cloud? If so, it would be incorrect to multiply it by $T_{sfc}^4$ (which Eq. 14 does). Later, $\Delta F_{tir}$ is shown for IWC, re, and ice crystal shape, so it appears that $\varepsilon$ refers to the ice cloud and therefore $\varepsilon < 1$, but how then does $\varepsilon$ depend on IWC, re and ice particle shape? The dependence of $\Delta F_{tir}$ on cloud properties is a complete black-box mystery and this needs to be explained.

Author response: As mentioned in our reply to comment 7, a dedicated section for TIR RT was added to the manuscript. It is primarily based on the TIR RT approximation given by Corti and Peter (2009). Equation 14 is now replaced by Eq. 20. Major steps to derive Eq. 20 are given in the manuscript; details can be found in Corti and Peter (2009).

Referee comment for 2$^{nd}$ review: Same as above regarding comment #1.

3. Figure 4d: The dot-dash curve showing the absolute difference in $\Delta F$ between plates and aggregates appears flawed for IWC > 0.02 g/m$^3$, assuming Fig. 4a is correct. Perhaps I have overlooked something, but in Fig. 4a for $\theta = 30°$ and $r_e = 25$ μm (dot-dashed), $\Delta F_{sol}$ appears fairly constant between plates and aggregates for IWC > 0.02 g/m$^3$, indicating that their absolute difference in Fig. 4d should be approximately constant for IWC > 0.02 g/m$^3$ (with the dot-dash line being approximately horizontal). If there is such an error, this will affect Fig. 4f as well. The other curves look reasonable, as well as the curves in Fig. 4g.
    I now see that Fig. 4a and Fig. 3b are different, although they should be the same if I understand correctly. The curves plotted in Fig. 3b appear consistent with those in Fig. 4d, suggesting that Fig. 4a is flawed.

4. Lines 391-395: Manfred Wendish wrote a paper on this topic in JAS(?) around 2008 I'm guessing.

5. Lines 398-399: The decreasing order at $r_e = 5$ μm (droxtals, plates, aggregates) changes when $r_e$ is larger to droxtals, aggregates and plates in Fig. 4b.

6. Lines 410-11: "relative differences exceed the absolute value by a factor of 10." How is this evident from the two plots where one is unitless and the other has units?

7.  I did not have time to carefully review the sections that came after Sect. 3.1, and the authors are encouraged to do so due to the above comments pertaining to Sect. 3.1 (#s 3 – 6) and the technical comments below.

**TECHNICAL COMMENTS:**  Line numbers correspond to the revised manuscript.

1.  Line 90: The net RE given by => The net RE is given by?

2.  Lines 93-94:  Redundant portion of sentence.

3.  Line 103:  Although the meaning of TIR might be inferred from lines 91-92, it is customary to explicitly state its meaning, like "The thermal infrared radiances (TIR) include …"

4.  Line 175:  Are you sure you want $\Lambda = -1/(a \cdot b)$ since this would make the exponent in (4) positive?

5.  Equation 10:  Since you are approximating $\tau_{ice}$ for solar radiation only, this equation can be further simplified by noting $Q_e \approx 2$.

6.  Line 212:  "The altitude of 1500 k was selected" => The altitude of 1500 m was selected?

7.  Line 378-9:  Mitchell (1996) => Mitchell (2002)?

8.  Line 389:  50 W m$^{-2}$ looks reasonable for plates at $r_e = 5$ µm, but I think this discussion is relating droxtals to aggregates, in which case the number looks closer to 25 W m$^{-2}$ for IWC = 0.024 g m$^{-3}$.

---

## Referee Report (RR2)

In the following, the *reviewer's original comments are presented in italics*, the authors' responses as well as references to the new manuscript version are in standard font, and the **reviewer's new remarks are displayed in bold**.

**Major issues:**

1. *Please provide a sample input file for the thermal spectrum and cloud file as well. From the information provided in the manuscript it is not possible to reproduce the results.*
   We followed this remark and now provide an additional example input file for the thermal-infrared wavelength range. Input files for clouds can easily be created using the libRadtran manual and do not require an example. The idea of providing an example input script is to be transparent and to provide a guideline that might be used as a template by a reader, and not to provide a copy and paste ready model configuration.
   **The results provided in the look-up table, which is intended for public use, contain incorrect results for more than 50% of the cases: Varying surface albedo has been neglected for the clear-sky thermal irradiance simulations, affecting the upward, downward flux (Fup_tir, Fdn_tir) as well as the net radiative effect (RE_net) for all cases with surface albedo > 0. For a given surface temperature, the results have constant values for all albedo values in [0.15, 0.3, 0.6, 1.0], whereas they are expected to vary with surface emissivity (= 1 – albedo), see libRadtran manual for reference. Assuming the results and figures presented in the manuscript rely on the same database, they will have to be revisited as well.**

1. *How is the optical thickness computed/derived? From libRadtran directly, or using the approximation provided by Eq. 10?*
   All values of cloud optical thickness were directly extracted from the libRadtran verbose files. The ice cloud optical thickness τice at 550~nm wavelength is directly obtained from the libRadtran verbose output using optical properties of droxtals.
   **The optical thickness for a given IWC and effective crystal radius will vary across ice crystal habits. If the optical thickness is determined only for droxtals and assumed constant for plates and column-aggregates, this approach will lead to incorrect results for these other habits.**

**Other technical issues to be addressed:**

2. *Line 112: Please explain why REPTRAN "coarse" mode is justified for this application and show that it provides sufficient resolution compared to "medium" and "fine" mode (if needed, consult with libRadtran team). REPTRAN 'coarse' provides a spectral resolution of 15 cm-1, which corresponds to Δλ =0.41nm (at 550 nm) and Δλ =3.5 μm at (50 μm).*
   We would like to direct the Referee to Fig. 3.7 on page 47 of the libRadtran Documentation (version 2.04). Figure 3.7 shows the different spectral resolutions of REPTRAN coarse, medium, and fine. As given in these examples, the resolution 'coarse' resolves the major features of the spectrum and, therefore, we argue that coarse is sufficient for broadband irradiance simulations. We further argue that the 'coarse' resolution is sufficient for broadband irradiance applications while acknowledging that higher spectral resolutions are

required for spectrally resolving radiance simulations. Furthermore, when calculating solar, TIR, and net radiative effect as differences between cloudy and cloud-free simulations, effects and potential errors from molecular absorption and due to the choice of spectral resolution from libRadtran partially compensate.

http://www.libradtran.org/doc/libRadtran.pdf (last access: June 28th, 2023)

**Please provide quantitative info about the bias introduced by choosing REPTRAN coarse vs. fine, as well as by limiting the thermal spectrum to 75,000 nm instead of 100,000 nm – both on the solar, thermal, and net radiative effect. It would be sufficient to run one simulation based on an extreme case of parameters (for which the largest effect is expected). This information is important for potential future users of the look-up table results.**

**Minor issues:**

3. *Please use the official description as provided in Yang et al. 2013 when referring to these ice crystal properties. It is not clear which ice crystal shapes and roughness levels are used in this study. Yang et al. 2013 provide each habit in 3 different roughness levels. The sample input file hints at the choice of "moderately rough aggregates of 8-element columns". Please double-check throughout the manuscript.*

   Aggregates consisting of '8-element-columns' were used from Yang et al. (2013). This has been clarified in the manuscript in Sec. 2.2 Radiative transfer simulation set-up. Later in the text the term 'aggregates' is used synonymously for '8-element-columns'.

   **Please use the full term "moderately rough aggregates of 8-element columns", "8-element columns" is not specific enough in this case, since there are three different roughness levels provided by Yang et al. 2013 (see original comment above).**
   **This applies in a similar way to Figure D1: Please change "8--column aggregates (called 'aggregates' thereafter)" to "aggregates of 8-element columns with moderate surface roughness (called 'aggregates' thereafter)".**
   **Please double-check throughout the manuscript.**

4. Lines 146-152: Several airborne in situ measurement campaigns that targeted cirrus and contrails imply that aggregates are the dominating ice crystal habit (Liu et al., 2014; Holz et al., 2016; J.rvinen et al., 2018). For example, Järvinen et al. (2018) found that 61 to 81 % of the sampled ice crystals were aggregates with a rough surface. Such ice crystals are also assumed in current remote sensing applications of ice cloud, e.g., in the re-defined ice optical properties used by the Moderate Resolution Imaging Spectroradiometer (MODIS) Collection 6 product (Yang et al., 2013; Holz et al., 2016;Platnick et al., 2017; Forster and Mayer, 2022). Therefore, we selected 8–column–aggregates as the primary ice crystal habit.
   **Please double-check the literature, the references here are still not correct: Järvinen et al. 2018 report that 61 to 81% of the sampled ice crystals were found to be *complex* [meaning they had featureless phase functions; they do not mention *aggregates* here]. Later, they state that "severely roughened column aggregates" are found to best represent their observations. MODIS Collection 6 assumes severely roughened 8-element column aggregates as well. Forster and Mayer (2022) found mixtures of severely roughened (~60%) and smooth (~40%) 8-column aggregates to best match observations of (thin) cirrus. In fact, the latter more closely motivates the use of**

**moderately rough 8-element columns in this study. Please note that the optical properties of aggregates closely resemble those of their components (e.g. the asymmetry factor of aggregates of columns is similar to that of individual columns), so it is important to be specific here about the type of aggregates, as well as the degree of surface roughness (cf. comment #4 above).**

5. *Table 3 states "Molecular absorption: Fu and Liou (1992, 1993)". This is inconsistent with the earlier statement of REPTRAN. Please correct/clarify.*
   The table has been corrected according to the Referee's comment and we have replaced the former citation with the Gasteiger et al. (2014) reference.
   **Please add REPTRAN (Gasteiger et al, 2014) to the table.**

6. **The sample libRadtran input file for the thermal-infrared specifies the solar zenith angle, which does not have any meaning in this spectral range. Even though this won't have any impact on the simulation results, please remove this line as it potentially confuses future readers/users.**

7. *Compared to Meerk.tter et al, the visualization here is dominated by the parameter range for reff, making it almost impossible to visually resolve variations in F_net, tir, sol for the remaining variables.*
   We partly agree with the Referee and elongated the figure to improve the legibility of the figure. However, the large bar from $R_{eff}$ in relation to the other bars is also a direct indicator of the relevance of each parameter considering the typical parameter range. Consequently, the importance of each individual parameter, in relation to the others, is directly visible in the figure.
   **A log scale would help here, or interrupting the y-axis at -400 W/m2 (solar) -200 W/m2 (net).**

---

## Author Response (AR2)

**Report #2 Referee#1 (David Mitchell)**

**Checklist for reviewers**
**1) Scientific significance**
Does the manuscript represent a substantial contribution to scientific progress within the scope of this journal (substantial new concepts, ideas, methods, or data)?
Outstanding **Excellent** Good Fair Low
**2) Scientific quality**
Are the scientific approach and applied methods valid? Are the results discussed in an appropriate and balanced way (consideration of related work, including appropriate references)?
Outstanding Excellent **Good** Fair Low
**3) Presentation quality**
Are the scientific results and conclusions presented in a clear, concise, and well structured way (number and quality of figures/tables, appropriate use of English language)?
Outstanding Excellent Good **Fair** Low
**For final publication, the manuscript should be**
accepted as is
accepted subject to technical corrections
**accepted subject to minor revisions**
reconsidered after major revisions
rejected

**Were a revised manuscript to be sent for another round of reviews:**
**I would be willing to review the revised manuscript.**
I would not be willing to review the revised manuscript.

**Suggestions for revision or reasons for rejection**
(visible to the public if the article is accepted and published)
**GENERAL COMMENTS:**

The manuscript has been greatly improved and, in general, the authors have done a great job in response to the review comments I have made. However, they mention Section 2.4 titled "Approximation of radiative transfer in the thermal infrared", which somehow was not included in the revised version of this manuscript. This section is critical since it describes how radiation transfer is treated in the thermal infrared (TIR) spectrum; it is needed to properly understand the TIR results of this sensitivity study. This manuscript should not be published without it.

**SPECIFIC COMMENTS:**

**1. On comment #7 from 1st review:**
**Equation 13: Is this equation used in libRadtran? If not, what is the point in mentioning it? Cloud property input to libRadtran consists of IWC and re, suggesting the zero-scattering approximation might be used for TIR hemispheric fluxes: $\varepsilon = 1 - \exp(-5\,\tau_{abs}/3)$ where $\varepsilon$ is cloud emissivity and $\tau_{abs}$ is the cloud absorption optical depth. Please indicate whether $\varepsilon$ is calculated in libRadtran, and how it is calculated if applicable.**

**Author response: The DISORT solver in libradtran (Buras et al 2011) calculates scattering in the TIR on basis of the bulk-scattering properties of ice crystals, analog to the solar wavelength range. Thus, the zero-scattering approximation is not used in the simulations. Equation 13 was added to the manuscript to provide guidance for the reader. To avoid misinterpretation the equation is brought into context and is expanded to section "2.4 Approximation of radiative transfer in the thermal-infrared", to incorporate suggestions from other Reviewers.**

**Referee comment for 2nd review: The author response above is puzzling since the Referee is finding no section 2.4 titled "Approximation of radiative transfer in the thermal-infrared" in the revised manuscript nor in the track-changes version (the diff file) of the manuscript. In the current revised manuscript, there is no discussion of how RT in the thermal infrared (TIR) is dealt with, which is critical for a RT sensitivity study presenting results in both the solar and TIR. Since the authors mention Sect. 2.4 in their response having the title "Approximation of radiative transfer in the thermal-infrared", it appears that this section was mistakenly omitted from the manuscript. The manuscript should not be published without this section.**

Please see the answer to the following comment.

**2. On comment #8 from 1st review:**
**Lines 209 – 213 and Eq. 14: Eqn. (14) appears flawed since, in principle, there should be an emissivity term (ε) for both the surface and the ice cloud. But since typically ε ≈ 1 at the surface, does ε in (14) correspond only to the ice cloud? If so, it would be incorrect to multiply it by Tsfc4 (which Eq. 14 does). Later, ΔFtir is shown for IWC, re, and ice crystal shape, so it appears that ε refers to the ice cloud and therefore ε < 1, but how then does ε depend on IWC, re and ice particle shape? The dependence of ΔFtir on cloud properties is a complete black-box mystery and this needs to be explained.**

**Author response: As mentioned in our reply to comment 7, a dedicated section for TIR RT was added to the manuscript. It is primarily based on the TIR RT approximation given by Corti and Peter (2009). Equation 14 is now replaced by Eq. 20. Major steps to derive Eq. 20 are given in the manuscript; details can be found in Corti and Peter (2009).**

**Referee comment for 2nd review: Same as above regarding comment #1.**

The section "Approximation of radiative transfer in the thermal-infrared" is re-introduced. The section provides a brief outline of TIR radiative transfer that helps to understand and interpret the results related to the TIR radiative effect. We would like to direct the Referee to the diff file to avoid copying the entire section into the point-by-point response.

**3. Figure 4d: The dot-dash curve showing the absolute difference in ΔF between plates and aggregates appears flawed for IWC > 0.02 g/m3, assuming Fig. 4a is correct. Perhaps I have overlooked something, but in Fig. 4a for θ = 30° and re = 25 μm (dot-dashed), ΔFsol appears fairly constant between plates and aggregates for IWC > 0.02 g/m3, indicating that their absolute difference in Fig. 4d should be approximately**

**constant for IWC > 0.02 g/m3 (with the dot-dash line being approximately horizontal). If there is such an error, this will affect Fig. 4f as well. The other curves look reasonable, as well as the curves in Fig. 4g.**
**I now see that Fig. 4a and Fig. 3b are different, although they should be the same if I understand correctly. The curves plotted in Fig. 3b appear consistent with those in Fig. 4d, suggesting that Fig. 4a is flawed.**

The Referee correctly noted that Fig.4a was flawed. The plot has been corrected and replaced, and is now consistent with Fig. 3.

**4. Lines 391-395: Manfred Wendish wrote a paper on this topic in JAS(?) around 2008 I'm guessing.**

The Referee is right. An explanation and the following reference were added: Wendisch, M. / Pilewskie, P. / Pommier, J. / Howard, S. / Yang, P. / Heymsfield, A. J. / Schmitt, C. G. / Baumgardner, D. / Mayer, B., Impact of cirrus crystal shape on solar spectral irradiance: A case study for subtropical cirrus, 2005, J. Geophys. Res. Atmos. , Vol. 110, No. D3

*This is supported by earlier observations and simulations for example by Wendisch et al (2005).*

**5. Lines 398-399: The decreasing order at re = 5 μm (droxtals, plates, aggregates) changes when re is larger to droxtals, aggregates and plates in Fig. 4b.**

The Referee is correct. The sentence has been corrected.

*With increasing crystal size the order changes to droxtal, aggregates, and plates, and the absolute values of $\Delta F_{TIR}$ decrease.*

**6. Lines 410-11: "relative differences exceed the absolute value by a factor of 10." How is this evident from the two plots where one is unitless and the other has units?**

The Referee is right. The sentence has been phrased incorrectly and has been modified. The new sentence reads:

*The largest relative deviations are found for the optically thinnest clouds, where $\Delta F_{net}$ is generally small. In these cases of optically thin clouds consisting of the smallest crystals ($r_{eff} = 5$ μm), the relative deviations exceed the relative difference for optically thick clouds with the same crystal size by a factor of 10.*

**7. I did not have time to carefully review the sections that came after Sect. 3.1, and the authors are encouraged to do so due to the above comments pertaining to Sect. 3.1 (#s 3 – 6) and the technical comments below.**

We have taken note of the comment and have thoroughly reviewed the remaining part of the manuscript for errors. We would like to direct the Referee to the pdf diff file.

**TECHNICAL COMMENTS: Line numbers correspond to the revised manuscript.**

**1. Line 90: The net RE given by => The net RE is given by?**
The Referee is right and the sentence has been corrected.

**2. Lines 93-94: Redundant portion of sentence.**
The redundant section has been removed.

**3. Line 103: Although the meaning of TIR might be inferred from lines 91-92, it is customary to explicitly state its meaning, like "The thermal infrared radiances (TIR) include …"**
We agree with the Referee and now explain the abbreviation.

**4. Line 175: Are you sure you want Λ = − 1/(a·b) since this would make the exponent in (4) positive?**
The minus sign has been removed.

**5. Equation 10: Since you are approximating τice for solar radiation only, this equation can be further simplified by noting Qe ≈ 2.**
The Referee is right. We added  Qe ≈ 2 and simply the equation further.

**6. Line 212: "The altitude of 1500 k was selected" => The altitude of 1500 m was selected?**
The typo was corrected and the unit was changed to: 1500 m

**7. Line 378-9: Mitchell (1996) => Mitchell (2002)?**
The Referee is right. The citation has been corrected.

**8. Line 389: 50 W m-2 looks reasonable for plates at re = 5 μm, but I think this discussion is relating droxtals to aggregates, in which case the number looks closer to 25 W m-2 for IWC = 0.024 g m-3.**
The Referee is right. The values has been correct to 27 W m$^{-2}$.

**Report #2  Referee#2  (Andreas Macke)**

Checklist for reviewers
1) Scientific significance
Does the manuscript represent a substantial contribution to scientific progress within the scope of this journal (substantial new concepts, ideas, methods, or data)?
Outstanding  Excellent    **Good** Fair    Low
2) Scientific quality
Are the scientific approach and applied methods valid? Are the results discussed in an appropriate and balanced way (consideration of related work, including appropriate references)?
Outstanding  Excellent    **Good** Fair    Low
3) Presentation quality
Are the scientific results and conclusions presented in a clear, concise, and well structured way (number and quality of figures/tables, appropriate use of English language)?
Outstanding  **Excellent**    Good  Fair    Low
For final publication, the manuscript should be
accepted as is
accepted subject to technical corrections
**accepted subject to minor revisions**
reconsidered after major revisions
rejected

**Were a revised manuscript to be sent for another round of reviews:**
I would be willing to review the revised manuscript.
**I would not be willing to review the revised manuscript.**

**Suggestions for revision or reasons for rejection**
(visible to the public if the article is accepted and published)
I still think that the consideration of a water cloud under the cirrus cloud is rather arbitrary and basically physically accounted for by the variation of the radiation quantities surface albedo and temperature. I found the arguments for keeping this cloud configuration not fully convincing. I leave the decision on this to the editor.

We would like to comment on this advice with the following arguments.

One argument for an additional cloud layer is that the cloud albedo is spectrally different from the surface albedo as we assume a constant surface albedo for all wavelength. By including the liquid water cloud layer, we achieve consistency between the surface, the liquid cloud and the ice cloud temperatures, which would not be the case by simply varying the surface temperature.

**Report #2 Referee #3 (anonymous)**

**Checklist for reviewers**
**1) Scientific significance**
**Does the manuscript represent a substantial contribution to scientific progress within the scope of this journal (substantial new concepts, ideas, methods, or data)?**
Outstanding  Excellent     Good  **Fair**   Low
**2) Scientific quality**
**Are the scientific approach and applied methods valid? Are the results discussed in an appropriate and balanced way (consideration of related work, including appropriate references)?**
Outstanding  Excellent     Good  **Fair**   Low
**3) Presentation quality**
**Are the scientific results and conclusions presented in a clear, concise, and well structured way (number and quality of figures/tables, appropriate use of English language)?**
Outstanding  Excellent     Good  **Fair**   Low
**For final publication, the manuscript should be**
accepted as is
accepted subject to technical corrections
accepted subject to minor revisions
**reconsidered after major revisions**
rejected

**Were a revised manuscript to be sent for another round of reviews:**
**I would be willing to review the revised manuscript.**
I would not be willing to review the revised manuscript.

**Suggestions for revision or reasons for rejection**
(visible to the public if the article is accepted and published)
**The authors have addressed several points raised in the previous review. Yet, two key issues that were previously raised have not been appropriately addressed. Additionally, there are several other remaining issues that need to be resolved before the publication can proceed. In light of this, I strongly recommend consulting with both the libRadtran team and the authors of the reference studies (cf. (A)) in order to effectively address and resolve these outstanding concerns.**

**(A) The manuscript is missing a (quantitative and more detailed) discussion of the results with previous studies which are mentioned in the introduction (Fu and Liou (1993), Yang et al. 2010, Zhang et al. 1999, Mitchell et al. 2011, Schumann 2012, Meerkötter et al. 1999). It is stated that the presented study builds upon and is motivated by Meerkötter et al. 1999. Even the central Figure 2 is adapted from this study as stated in the figure caption. References to this study and the other central publication from Schumann et al. 2012 have been added in response to the previous review, but remain all qualitative (cf. lines 349, 412, 582). A clear motivation is missing why a new study is needed and what the improvements or new insights are compared to previous results.**

General comment on the more quantitative comparison:

The requested more quantitative comparison with the cited literature is complicated or impossible as the data sets used in these studies are not publicly available. Only the parameterizations by Schumann et al (2012) might be used to re-create the fir from Schumann et al (2012) that could be compare it with our results. This was partly done in the Community comment (https://doi.org/10.5194/egusphere-2023-155-CC1) by Schumann et al., which showed good agreement between our simulations and the parameterization. However, I was not our intention to replicate or recreate previous studies. The motivation for this study was to provide a parameter-based sensitivity study and an publicly available data set that can be used for various potential applications.

We followed the suggestion of the Referee and provided a more quantitative comparison where possible. This comparison is limited to a few sections as our simulated clouds deviate from the cloud setup in Meerkoetter et al. (1999), particularly the geometric thickness. Even Meerkoetter et al. (1999) states that:
"Our results depend also strongly on the assumed IWC, the geometrical thickness of the contrails, and the particle sizes of the contrails, all together controlling the optical depth of the contrails. Contrails could have a factor of 3 smaller IWC than assumed in this study (see Fig. 1), may be geometrically thicker by a factor of 3 (Freudenthaler et al., 1996; Sassen, 1997), and the particle size is known, at best, to a factor of 2, implying an uncertainty in optical depth of about factor 4 (based on the square of the sum of the individual uncertainty factors). The contrail cover value is estimated to be uncertain by a factor of about 2. Hence, the given amount of radiative forcing by contrails is uncertain by a factor of about 5, mainly because of uncertain contrail cover and optical depth values."

**1. Line 349: "The presented analysis of solar, TIR, and net ΔF sensitivity on the selected input parameters generally agrees with the results from Meerkötter et al. (1999). We found differences in the importance of the parameters, which are explained by the fact that our simulations span a larger and different parameter range, for example in reff and Tsrf. In addition, the sensitivity analysis in Fig. 2 is sensitive to the selection of the reference cloud."**

Concerning Line 349:
Selecting parameters from our parameter space that are most similar to the cloud case (A) from Meerkoetter et al (1999) yield a change in the importance of the parameter. While we found $R_{eff}$ to be more important than IWC for $dF_{sol}$ in our study, selected parameter ranges similar to Meerkoetter et al (1999) changed the order of importance of the parameters, with IWC becoming more important than $R_{eff}$.

A major difference between Meerkoetter et al (1999) and our study is the cloud geometric thickness. Meerkoetter et al (1999) used an IWP of 4.4 g m$^{-2}$ and an IWC of 0.021 g m$^{-3}$. Assuming a homogeneous distribution of IWC, this leads to an approximate cloud geometric thickness of 210 m. In our case the simulations have been performed for a cloud geometric thickness of 1000 m. Matching IWC and reff of our simulations to yield a similar cloud optical thickness of 0.52 is only valid for the solar wavelength range. The cloud geometric thickness will influence the results in the TIR wavelength range and subsequently in the net cirrus radiative effect.

**2. Line 412: "The analysis of all simulations shows that the shape assumption has only second-order implications on the RE compared to other parameters like IWC or reff (see Fig 2), which agrees with Meerkötter et al. (1999)."**

The qualitative description is extended by a more quantitative one. The section has been extended as follows:

*The analysis of all simulations shows that the crystal shape assumption on the cirrus RE is small compared to other parameters particularly IWC or $r_{eff}$ (see Fig 2). However, we found a larger variability in $\Delta F_{sol}$ and the resulting $\Delta F_{net}$, i.e., whether a contrail has a net warming or cooling effect compared to $\Delta F_{tir}$. For the defined reference consisting of aggregates, a $\Delta F_{sol}$ of -50.2 W $m^{-2}$ was simulated, while for plates and droxtals values of $\Delta F_{sol}$ of −8.6 and −44.3 W $m^{-2}$ were obtained, respectively. The impact of the crystal shape is less pronounced in the TIR wavelength range with $\Delta Ft_{ir}$ of 46, 44.5, and 48.9 W $m^{-2}$ for aggregates, plates, and droxtals, respectively. The variation of $\Delta F_{sol}$ propagates into $\Delta F_{net}$ with −4.2, 35.9 and 4.6 W $m^{-2}$ for aggregate, plates, and droxtals, respectively. Based on the presented simulations, we found larger maximum variations in $\Delta F_{sol}$, $\Delta F_{tir}$, and $\Delta F_{net}$ of 41.6, 4.4, and 40 W $m^{-2}$, respectively, compared to Meerkötter et al. (1999). They found variations of $\Delta F_{sol}$, $\Delta F_{tir}$, and $\Delta F_{net}$ of 2, 6, and 7 W $m^{-2}$, respectively. The difference are explained by the selected reference (Meerkötter et al., 1999). However, selecting cloud parameters similar to the reference cloud of Meerkötter et al. (1999), we still found larger maximum variations $\Delta F_{sol}$, $\Delta F_{tir}$, and $\Delta F_{net}$ of 17.3, 4.2, and 17.9 W $m^{-2}$, respectively. This is attributed to the remaining differences among the selected reference values.*

**3. Line 582: "While Meerkötter et al. (1999) showed that solar, TIR, and net ΔF are only slightly sensitive to changes in dz – under the premise of a constant ice water path (IWP) – the present simulations indicate that the effects on ΔFtir and particularly ΔFnet have to be considered."**

Concerning line 582:
The section was extended and we quantify the effects from varying d*z*. The following sentences were modified and added:

*This partly agrees with the findings from Meerkötter et al. (1999) who showed that solar, TIR, and net ΔF are only slightly sensitive to changes in dz with solar, TIR, and net ΔF below 2 W $m^{-2}$, under the premise of a constant ice water path (IWP). The presented simulations indicate ΔFsol of 2 W $m^{-2}$, which are comparable to Meerkötter et al. (1999), but we found slightly higher ΔFtir and ΔFnet of 4.5 and 3.1 W $m^{-2}$, respectively.*

**(B) A clear statement of the intended use of the dataset together with assumptions made for the radiative transfer simulations and their impact on the accuracy of the results is missing. What is the intended use case of this dataset beyond the presented sensitivity study? What are possible applications or scientific questions that can be answered with the data?**
**The following statement was added in response to the previous review (line 76): "The data set might be coupled with cloud microphysical models, e.g, the Contrail Cirrus Prediction Tool (CoCiP) from Schumann (2012), to estimate the RE of the simulated**

**contrails." If this is the intended use case, please provide more details on how CoCiP works and how it can be coupled with this dataset.**

The look-up-table is intended to estimate the contrail radiative effect based on the eight-dimensional parameter space. Therefore, CoCiP was mentioned as a potential user-case example. However, to avoid to be too prescriptive, we rephrased the manuscript and provide a more generic example as we hesitate to imply or determine the usage of the data to the users. Quite the opposite. We are interested in new applications that we cannot think of. In the light of this we argue that the look-up-table can be coupled together with models of any complexity as long as the model output agrees with the dimensions of the data set.

The text is rephrased as following:

*The look-up-table could in fact be coupled with models of any complexity, as long as they simulate the dimensions of the data set, namely: solar zenith angle, ice cloud temperature, surface albedo, ice water content, surface temperature, ice crystal effective radius, and liquid water cloud optical thickness.*

**(C) Technical issues regarding radiative transfer setup and results:**
**1. Effect of neglecting 3D radiative transfer:**
**a. While a discussion about 3D effects has been added, it is important to mention in the abstract and summary that this study is based on 1D radiative transfer.**

We agree with the Referee and now mention that the simulations were performed with a 1D radiative transfer solver.

*[...] In total, 283,500 plane-parallel radiative transfer simulations have been performed, not including three-dimensional scattering effects. Parameter ranges are select that are typically associated with natural cirrus and contrails. In addition, the effect of variations in the relative humidity profile and the ice cloud geometric thickness have been investigated for a sub-set of the simulations. The multi-dimensionality and complexity of the 8-dimensional parameter space makes it impractical to discuss all potential configurations in detail. Therefore, specific cases are selected and discussed.  [...]*

*[...] The RT simulations were performed with a 1D solver (plane-parallel clouds) and 3D scattering effects were not considered despite the fact they become relevant*
*become relevant for large solar zenith angles ($\theta > 70\circ$.) [...]*

**b. The study by Gounou and Hogan was not cited correctly: Line 624: "…found differences in contrail solar RE between 1D and 3D simulations of up to 40%. These values were found for extreme cases, e.g., large solar zenith angle (Sun close to the horizon)." Significant differences were found not only for extreme cases:**
**--> The summary states: "The horizontal photon transport increases the longwave RF of contrails by around 10%". In the shortwave "the horizontal photon transport weakly decreases the magnitude of the RF by around 5%" and up to 30% for solar zenith angles >70deg. "When we consider the net RF […] the effect of the horizontal photon transport becomes important." Please correct accordingly.**

The section has been rephrased to the following:
*[... ] This study was followed by Gounou and Hogan (2007) and Forster et al. (2012), who used 3D Monte Carlo simulations and found differences in contrail solar RE between 1D and 3D simulations ranging from 5 to 40 %. The largest deviations were found for extreme cases, e.g., large solar zenith angle (Sun close to the horizon). With the Sun illuminating the contrail or cirrus from the side, extinction and absorption within the cloud increases and scattering at cloud sides becomes more important compared to an illumination from above. Enhanced scattering at cloud sides also increases the likelihood that photons get scattered back into space instead of being absorbed. Such effects are not captured by 1D RT simulations. Concerning the TIR wavelength range, Gounou and Hogan (2007) found that horizontal photons transport can increase contrail radiative effect by around 10 %, which has to be considered in the calculation of the contrail net radiative effect.*

**c. Line 645: "Only few studies are available on cirrus 3D effects, e.g., Hogan and Kew (2005); Gounou and Hogan (2007)." Should this sentence be deleted?**

This sentence was accidentally kept in the manuscript and is now removed from the text.

**2. Radiative transfer simulations with DISORT:**
**a. Line 105: DISORT (Buras et al. 2011) is not original author, please cite original author in addition:**
**Stamnes, K., Tsay, S.-C., Wiscombe, W., and Jayaweera, K.: Numerically stable algorithm for discrete–ordinate–method radiative transfer in multiple scattering and emitting layered media, Appl. Opt., 27, 2502–2509, 1988.**

We thank the Referee for providing this reference and have added it to the manuscript.

**b. The use of the solar and thermal spectral range is inconsistent throughout the manuscript. The introduction (line 38) mentions a range from 0.2-3.5 µm and 3.5-100 µm, respectively. Table 3: 0.3–3.5 µm (solar) & 3.5–75 µm (thermal-infrared). Why do the simulations not cover the full range up to 100 µm?**

The exact wavelength range that is covered by the term 'thermal-infrared' is not exactly defined in the literature and can be debated. For example, Petty (2006)* describes the thermal infrared band as located between 4 and 50 µm, and the far IR band between 50 and 1000 µm. However, we acknowledged the commonly used agreement of the terrestrial radiation reaching up to 100 µm. But, as outlined in the manuscript, the wavelength range from 3.5 to 75 µm covers 99.3% of the 3.5 to 100 µm range and, in our opinion, can be regarded as equivalent.

(*) Petty, G. W.; A first course in Atmospheric Radiation, 2006, Sundoc Publishing

**c. Line 112: Please explain why REPTRAN "coarse" mode is justified for this application and show that it provides sufficient resolution compared to "medium" and "fine" mode (if needed, consult with libRadtran team).**
REPTRAN 'coarse' provides a spectral resolution of 15 cm$^{-1}$, which corresponds to $\Delta\lambda$ =0.41 nm (at 550 nm) and $\Delta\lambda$ =3.5 µm at (50 µm).

We would like to direct the Referee to Fig. 3.7 on page 47 of the libRadtran Documentation (version 2.04). Figure 3.7 shows the different spectral resolutions of REPTRAN coarse, medium, and fine. As given in these examples, the resolution 'coarse' resolves the major features of the spectrum and,  therefore, we argue that coarse is sufficient for broadband irradiance simulations.

We further argue that the 'coarse' resolution is sufficient for broadband irradiance applications while acknowledging that higher spectral resolutions are required for spectrally resolving radiance simulations. Furthermore, when calculating solar, TIR, and net radiative effect as differences between cloudy and cloud-free simulations, effects and potential errors from molecular absorption and due to the choice of spectral resolution
from libRadtran partially compensate.

http://www.libradtran.org/doc/libRadtran.pdf (last access: June 28[th], 2023)

**d. Table 3 states "Molecular absorption: Fu and Liou (1992, 1993)". This is inconsistent with the earlier statement of REPTRAN. Please correct/clarify.**

The table has been corrected according to the Referee's comment and we have replaced the former citation with the Gasteiger et al. (2014) reference.

**e. Line 130: Surface temperatures Tsrf are set to −15.8∘C (subarctic winter), 15.05∘C (US standard), and 26.5∘C (tropical), respectively, to match the lower most temperature in the atmospheric profiles. Why is the additional selection of surface temperature necessary? The lowest temperature should coincide with surface level?**

The sentence was rephrased for better clarity to the following:
[...] *Surface temperatures Tsrf of −15.95∘C (subarctic winter), 14.85∘C (US standard), and 26.55∘C (tropical) are defined in libRadtran by the lower most temperature in the APs.* [...]

**f. Sample input file: why 257.2 K for surface temperature? This is not consistent with the parameters provided in the manuscript (e.g. Tab. 4).**

The value in the table has been corrected to 257.2 K.

**g. Line 134: "The cirrus cloud top temperatures T_cld,ice are selected to span the temperature range in which contrails and cirrus typically form (Krämer et al., 2020). Here we cover a range from 219 to 243 K. The resulting ice cloud top altitudes z_ice,CT are set to the altitude, where the temperature in the APs equals the desired T_cld,ice". How are the cloud top altitudes computed? Linear interpolation? Please clarify.**

Following the suggestion of the Referee a sentence was added to better explain how the altitude was determined.

[...] *Here we cover a range from 219 to 243 K. The resulting ice cloud top altitudes zice,CT are set to the altitude, where the temperature in the APs equals the desired Tcld,ice. zice,CT was found by linear interpolation between the altitude and temperature levels.* [...]

**h. Table 1: Cirrus pressure/altitude do not match the values. The labels seem to be swapped. Consider reducing the information to the essential numbers to make the table easier to understand, e.g. provide temperatures in [K] only.**

The labels have been swapped and moved in the table to appear directly over the respective values. The table should be more clear now. We would like to keep both units of K and ºC, as degree Celsius is frequently used in literature about cirrus observations, while libRadtran uses K.

**i. Why does the provided uvspec input file compute results at several altitudes? It is unclear why this is necessary and might even slow down the computation.**

For purposes not relevant for the study, the output was specified for multiple layers. To not confuse the reader the additional layers have been removed from the example script.

**j. Please provide a sample input file for the thermal spectrum and cloud file as well. From the information provided in the manuscript it is not possible to reproduce the results.**

We followed this remark and now provide an additional example input file for the thermal-infrared wavelength range. Input files for clouds can easily be created using the libRadtran manual and do not require an example. The idea of providing an example input script is to be transparent and to provide a guideline that might be used as a template by a reader, and not to provide a copy and paste ready model configuration.

**3. Choice of ice crystal properties:**
**a. Please use the official description as provided in Yang et al. 2013 when referring to these ice crystal properties. It is not clear which ice crystal shapes and roughness levels are used in this study. Yang et al. 2013 provide each habit in 3 different roughness levels. The sample input file hints at the choice of "moderately rough aggregates of 8-element columns". Please double-check throughout the manuscript.**

Aggregates consisting of '8-element-columns' were used from Yang et al. (2013). This has been clarified in the manuscript in Sec. 2.2 Radiative transfer simulation set-up. Later in the text the term 'aggregates' is used synonymously for '8-element-columns'.

**b. Line 151: Forster and Mayer (2022): "…found that cirrus are frequently comprised of mixtures of rough-aggregates and plates". This reference is not correct, the abstract states: "mixtures of smooth and rough column aggregates". Please correct.**

The sentence and this reference have been removed. The reference Forster et al (2022) was added to the previous sentence.

*Three different ice crystal shapes, namely: i) '8–column–aggregates' (called 'aggregates' thereafter), agglomerations of 8–columnar ice crystals; ii) 'droxtals', almost spherical ice crystals; and iii) 'plates' are used. These three shapes are selected to represent different stages in the temporal evolution of contrails. Several airborne in situ measurement*

*campaigns that targeted cirrus and contrails imply that aggregates are the dominating ice crystal habit (Liu et al., 2014; Holz et al., 2016; Järvinen et al., 2018). For example, Järvinen et al. (2018) found that 61 to 81 % of the sampled ice crystals were aggregates with a rough surface. Such ice crystals are also assumed in current remote sensing applications of ice cloud, e.g., in the re-defined ice optical properties used by the Moderate Resolution Imaging Spectroradiometer (MODIS) Collection 6 product (Yang et al., 2013; Holz et al., 2016; Platnick et al., 2017; Forster and Mayer, 2022). Therefore, we selected 8–column–aggregates as the primary ice crystal habit. The second most observed habit are plate-like ice crystals (Holz et al., 2016; Forster et al., 2017; Järvinen et al., 2018), which are included in the simulations as a second shape. The 'droxtal' parameterization is selected to estimate ΔF of young contrails, which primarily consist of near-spherical ice crystals (Goodman et al., 1998; Lawson et al., 1998; Gayet et al., 2012). We emphasize that contrails can be comprised of other ice crystal shapes, like single columns, hollow columns, 3D bullet rosettes, or mixtures of these (Lawson et al., 1998; Baum et al., 2005a), but the simulated shapes cover the majority of observed cirrus situations. The utilized ice optical properties of the three selected shapes are based on the parameterization from Yang et al. (2013) that assume randomly oriented ice crystals with a 'moderate' surface roughness.*

**c. Figure D1 shows phase functions for smooth ice crystals but in the text, "rough" crystals are mentioned. Please use the official description from Yang et al. 2013 and show the phase functions which are actually used to compute the look-up table.**

As described in the previous comment, we used 8-column-aggregates, droxtals, and plates with moderate surface roughness in the simulations. This has been checked throughout the manuscript.

*Three different ice crystal shapes, namely: 8--column--aggregates (called 'aggregates' thereafter), agglomerations of 8--columnar ice crystals; 'droxtals', almost spherical ice crystals; and 'plates' are used. [...]*

**4. Figure 2:**
**a. It is not clear how the variation of F_net, tir, sol in Figure 2 are achieved: are the remaining parameters fixed to the "reference case" or are they averaged as stated in line 281: "This strategy can be interpreted as a type of sub-sampling, by averaging all unfixed parameters to project ΔF onto the one-dimensional space."? Please clarify in the manuscript.**

Please see the next response to comment b)

**b. If the variation is investigated by "averaging all unfixed parameters", it is not clear what the role of the reference cloud is.**

The Referee is right. The phrase "averaging all unfixed parameters" was incorrectly positioned, which caused the misinterpretation. The section was shifted to the beginning of subsection "Sensitivity on solar zenith angle and surface albedo", as the averaging of all unfixed parameters is applied to create Figure 5 and the following ones. The reference is only used to create and discuss Fig.2.

We have also refrained from the term 'reference cloud', but call it reference. The reference is defined by selecting the minimum or maximum of the parameter range, except for IWC. An intermediate IWC was selected to obtain a cloud optical thickness of 0.48 (at 550 nm) that is representative for contrails. Selecting an intermediate IWC was required as using either the minimum or maximum IWC would lead to almost no cloud (smallest IWC) or a cloud with an optical thickness well exceeding 30 being unrepresentative for contrail or cirrus. In contrast, all other parameters, like solar zenith angle, $R_{eff}$, or surface albedo, from their minimum to their maximum can occur for cirrus clouds.

The text, describing the reference, was rephrased in the following way:

*[...] To reduce the multi-dimensionality, for each of the eight parameters a reference is defined by selecting either the minimum or maximum value from the parameter space. The reference parameters are selected to highlight the upper or lower range of each parameter and the spanned variation, and to define the reference for the fixed parameters. The reference parameters are given by $\theta = 0°$, $T_{cld,ice} = 219$ K, $\alpha_{srf} = 0$, $T_{srf} = 299.7$ K, $r_{eff} = 85$ µm, and $\tau l_{iq} = 0$ (no liquid water cloud). For IWC we use an intermediate value of 0.024 g m$^{-3}$ because together with a dz of 1000 m and an reff of 85 µm this leads to a $\tau_{ice}$ of 0.46 at 550 nm wavelength, which is representative for contrails and young cirrus (Iwabuchi et al., 2012). Otherwise selecting the minimum or maximum IWC in combination with reff of 85 µm would lead to high or low $\tau_{ice}$ that are not representative for contrails. For ice crystal shape, we select aggregates as the reference. We particularly emphasize that the defined references are not representative for any particular cloud situation, but are a useful point of comparison to assess the impact of a given parameter on the diversity of cloud RE.*

**c. The choice of parameters for the reference cloud is inconsistent: while for the IWC, a value "representative for contrails and young cirrus" (line 292) is selected, an effective radius of 85 µm seems quite extreme. A more representative choice would be 20 µm, for example.**

85 µm is the maximal $R_{eff}$ value that can be simulated with the default configuration of libRadtran. However, contrails and cirrus can be composed of ice crystals that are much larger. Sizes of up to several hundred µm have been observed. Therefore, 85 µm cannot be regarded as extreme.

**d. Similar, a surface albedo of 0 (completely black surface) is not very realistic. For ocean a value of 0.2 would be more representative.**

A value of 0.2 as the lower minimum and for open ocean is debatable. The albedo of open ocean, depending in wind conditions, can reach values of 0.08 or even lower. As pointed out, we wanted to cover the full (theoretical) albedo range from 0 to 1.

Literature that supports low sea surface albedo (from libRadtran user manual):
Cox, C. and Munk, W.: Measurement of the roughness of the sea surface from photographs of the sun's glitter, Journal of the Optical Society of America, 44, 838–850, 1954a.

Cox, C. and Munk, W.: Statistics of the sea surface derived from sun glitter, Journal of Marine Research, 13, 198–227, 1954b.

Nakajima, T. and Tanaka, M.: Effect of wind-generated waves on the transfer of solar radiation in the atmosphere-ocean system, J. Quant. Spectrosc. Radiat. Transfer, 29, 521–537, 1983.

**e. Fig. 2c reff 85 and 5 µm labels seems to be swapped. Please double-check.**

The Referee is right. The labels have been corrected and the figure was replaced.

**f. Compared to Meerkötter et al, the visualization here is dominated by the parameter range for reff, making it almost impossible to visually resolve variations in F_net, tir, sol for the remaining variables.**

We partly agree with the Referee and elongated the figure to improve the legibility of the figure. However, the large bar from $R_{eff}$ in relation to the other bars is also a direct indicator of the relevance of each parameter considering the typical parameter range. Consequently, the importance of each individual parameter, in relation to the others, is directly visible in the figure.

**5. Provided dataset**
**a. Cloudy_cloudfree variable is defaulted to 9.96921e+36 for both values. A more intuitive variable name would be cloud_fraction with values 0 and 1, referring to clearsky and cloudy, respectively. The meaning of these values can be added to the attributes.**

We follow the suggestion of the Referee and renamed the variable to cloud_fraction. The variable cloud_fraction is now correctly implemented in the data-set.

**b. Tau: not specified at which wavelength, this should be added to the attributes.**

It is not clear whether the Referee is concerned with the labeling of tau for the liquid or ice water cloud. However, the labeling of the data set was revised and the wavelength information (550 nm) was added to all ice and liquid water cloud optical thickness fields.

**c. Why does tau have dimensions ('solar_zenith_angle', 'ice_cloud_temp', 'surface_albedo', 'ice_water_content', 'surface_temperature', 'crystal_effective_radius', 'optical_thickness_liquid_water_cloud')? It should only depend on IWC and effective radius, that way saving storage space.**

For consistency with the size of the other data fields and for convenience in reading the data, the dimensions of the tau array have been kept.  The full data set, containing three files with several thousands of entries, and a total size of 15 MB, is surely manageable in terms of storage space.

**d. How is the optical thickness computed/derived? From libRadtran directly, or using the approximation provided by Eq. 10?**

All values of cloud optical thickness were directly extracted from the libRadtran verbose files.

*The ice cloud optical thickness τice at 550~nm wavelength is directly obtained from the libRadtran verbose output using optical properties of droxtals.*

---

## Author Response (AR3)

**Reply to Reviewer #3  Report #3**
(Referee comment on "Radiative effect by cirrus cloud and contrails – A comprehensive sensitivity study" by Kevin Wolf et al., EGUsphere, 2023)

In the following, the Reviewer's comments are highlighted in **bold**, comments from our side are given in standard font, and changes in the manuscript are given in *italic*.

**Major issues:**

*1.* **The results provided in the look-up table, which is intended for public use, contain incorrect results for more than 50% of the cases: Varying surface albedo has been neglected for the clear-sky thermal irradiance simulaCons, affecCng the upward, downward flux (Fup_Cr, Fdn_Cr) as well as the net radiaCve effect (RE_net) for all cases with surface albedo > 0. For a given surface temperature, the results have constant values for all albedo values in [0.15, 0.3, 0.6, 1.0], whereas they are expected to vary with surface emissivity (= 1 – albedo), see libRadtran manual for reference. Assuming the results and figures presented in the manuscript rely on the same database, they will have to be revisited as well.**

Surface albedo, as is customary in atmospheric sciences, refers to the albedo of the surface in the solar part of the electromagnetic spectrum. In the infrared we assume an emissivity of 1 (see line 130 in the last version of the manuscript), which is a reasonable assumption for most surfaces. Of course, the radiative quantities in the terrestrial infrared do not depend on surface albedo in the solar spectrum and solar zenith angle. For ease of use, all the radiative quantities in the database are provided with the same dimensions (incl. surface albedo and solar zenith angle). In this way the fluxes or RE in the solar and terrestrial parts of the spectrum can be summed easily to produce net quantities. The comment by the Reviewer that the calculations are incorrect for > 50% of the cases is unfounded.
For the sake of clarity, we now specifically mention that surface albedo is for the solar spectrum in a revised version of Table 4.

**2.The optical thickness for a given IWC and effective crystal radius will vary across ice crystal habits. If the optical thickness is determined only for droxtals and assumed constant for plates and column-aggregates, this approach will lead to incorrect results for these other habits.**

The sentence is taken out of context. The first paragraph of section 3 describes the relation of ice crystal radius, ice water content, ice crystal number concentration, and ice cloud optical thickness for droxtals (only). We already specified that this „overview' is for droxtals, hence, we do use the verbose output of libRadtran for droxtals.

Concerning the look-up-table: One of the dimensions in the database is the ice water content. The ice cloud optical depth for a given ice water content differs with the different crystal habits. This is why we provide the ice cloud optical depth in the database, where it can be checked that ice cloud optical depth varies across crystal habits as it indeed should. Furthermore, for ease of use, ice cloud optical depth is provided with the same dimensions as the other quantities but only varies with ice water content for a given crystal habit.

We further clarified caption of Fig.1:

*Calculated ice crystal number concentration Nice (in cm$^{-3}$) and simulated cloud optical thickness $\tau_{ice}$ at 550 nm wavelength as a function of ice water content IWC (in g m$^{-3}$) and effective crystal radius $r_{eff}$ (in µm) assuming droxtals [...]*

**Other technical issues to be addressed:**

**3.Please provide quantitative info about the bias introduced by choosing REPTRAN coarse vs. fine, as well as by limiting the thermal spectrum to 75,000 nm instead of 100,000 nm – both on the solar, thermal, and net radiative effect. It would be sufficient to run one simulation based on an extreme case of parameters (for which the largest effect is expected). This information is important for potential future users of the look-up table results.**

We estimated the uncertainty in the simulations when using REPTRAN 'coarse' instead of 'fine' by simulating one particular cloud case. The selected simulation is defined by: a SZA of 70º, for a long and slanted path through the atmosphere to maximize the impact of molecular absorption; an ice cloud temperature of 231 K, as the center of the parameter space; a surface albedo of 0.15, for moderate surface reflection; an IWC of 0.012 g m$^{-3}$; a surface temperature of 300 K, to select the tropical atmospheric profile with the highest water vapor concentration; and an intermediate effective radius of 25 µm. No underlying liquid water cloud is simulated.
This set up has been simulated with REPTRAN 'coarse' and 'fine' leading to relative differences in the solar, TIR, and net radiative forcing of 0.4%, 0.2%, and 1.9%, respectively. The following section is added to the manuscript in the Appendix C.

*[...] We estimated the uncertainty that is associated with the REPTRAN 'coarse' parameterization instead of the 'fine' resolution by simulating one particular cloud case and running the simulation with both options. The selected simulation is characterized by: a solar zenith angle θ = 70◦, for a long and slanted path through the atmosphere to maximize the impact of molecular absorption; a cirrus temperature $T_{cld,ice}$ of 233 K, as the center of the parameter space; a surface albedo $\alpha_{srf}$ = 0.15, for moderate surface reflection; an ice water content IWC = 0.012 g m$^{-3}$, a surface temperature $T_{srf}$ = 300 K, to select the tropical atmospheric profile with the highest water vapor concentration; and an ice crystal effective radius $r_{eff}$ = 25 µm. Based on the two simulations, relative differences in the solar, TIR, and net radiative forcing ΔF of 0.4 %, 0.2 %, and 1.9 % were determined, respectively.*

The bias introduced by considering a limit of 75 µm instead of 100 µm for terrestrial quantities is 0.7 %, as we already discussed in line 105 of the last version of the manuscript.

**Minor issues:**

**4. Please use the full term "moderately rough aggregates of 8-element columns", "8-element columns" is not specific enough in this case, since there are three different roughness levels provided by Yang et al. 2013 (see original comment above). This**

**applies in a similar way to Figure D1: Please change "8--column aggregates (called 'aggregates' thereafter)" to "aggregates of 8-element columns with moderate surface roughness (called 'aggregates' thereafter)". Please double-check throughout the manuscript.**

The abbreviation 'Aggregates' is now introduced with the full description 'moderately rough aggregates of 8-element columns' provided by the Reviewer. The title of Fig. D1 was changed to 'aggregates' only as the full name is too long and the abbreviation 'aggregates' is explained in the caption:

*[...]Aggregates are represented by moderately rough aggregates of 8-element columns[...]*

**5. Please double-check the literature, the references here are still not correct: Järvinen et al. 2018 report that 61 to 81% of the sampled ice crystals were found to be complex [meaning they had featureless phase functions; they do not mention aggregates here]. Later, they state that "severely roughened column aggregates" are found to best represent their observations. MODIS Collection 6 assumes severely roughened 8-element column aggregates as well. Forster and Mayer (2022) found mixtures of severely roughened (~60%) and smooth (~40%) 8-column aggregates to best match observations of (thin) cirrus. In fact, the latter more closely motivates the use of moderately rough 8-element columns in this study. Please note that the optical properties of aggregates closely resemble those of their components (e.g. the asymmetry factor of aggregates of columns is similar to that of individual columns), so it is important to be specific here about the type of aggregates, as well as the degree of surface roughness (cf. comment #4 above).**

The citations have been corrected by following the Reviewers comments. The section has been rephrased as following:

*[...] Several airborne in situ measurement campaigns that targeted cirrus and contrails imply that aggregates are the dominating ice crystal habit (Liu et al., 2014; Holz et al., 2016; Järvinen et al., 2018). For example, Järvinen et al. (2018) found that 61 to 81 % of the sampled ice crystals had complex shapes. They further noted that severely roughened column aggregates resemble their observations best. Such ice crystals are also assumed in current remote sensing applications of ice cloud, e.g., in the re-defined ice optical properties used by the Moderate Resolution Imaging Spectroradiometer (MODIS) Collection 6 product (Yang et al., 2013; Holz et al., 2016; Platnick et al., 2017; Forster and Mayer, 2022). Furthermore, Forster and Mayer, 2022 found mixtures of severely roughened (~60 %) and smooth (~40 %) 8-column-aggregates to best match observations of (thin) cirrus. As a compromise, we selected moderately rough 8–column–aggregates as the primary ice crystal habit. [...]*

**6. Please add REPTRAN (Gasteiger et al, 2014) to the table.**

Table 3 has been updated and REPTRAN is added.

**7.The sample libRadtran input file for the thermal-infrared specifies the solar zenith angle, which does not have any meaning in this spectral range. Even though this won't have any impact on the simulation results, please remove this line as it potentially confuses future readers/users.**

To avoid confusion for the reader, the line for the solar zenith angle is removed from the TIR input file.

**8. A log scale would help here, or interrupting the y-axis at -400 W/m2 (solar) -200 W/m2 (net).**

We prefer keeping the full, linear scale to show the full extent of the forcing, without any modification of the *y*-axis.